# Conformal Prediction for Hierarchical Data

**Guillaume Principato**                                    *guillaume.principato@universite-paris-saclay.fr*
*EDF R&D, Palaiseau, France*
*Université Paris-Saclay, CNRS, Inria, Laboratoire de mathématiques d'Orsay, Orsay, France*

**Gilles Stoltz**                                          *gilles.stoltz@universite-paris-saclay.fr*
*Université Paris-Saclay, CNRS, Inria, Laboratoire de mathématiques d'Orsay, Orsay, France*
*HEC Paris, Jouy-en-Josas, France*

**Yvenn Amara-Ouali**[*]                                    *yvenn.amara-ouali@edf.fr*
*EDF R&D, Palaiseau, France*
*Université Paris-Saclay, CNRS, Laboratoire de mathématiques d'Orsay, Orsay, France*

**Yannig Goude**[*]                                         *yannig.goude@edf.fr*
*EDF R&D, Palaiseau, France*
*Université Paris-Saclay, CNRS, Laboratoire de mathématiques d'Orsay, Orsay, France*

**Bachir Hamrouche**[*]                                     *bachir.hamrouche@edf.fr*
*EDF R&D, Palaiseau, France*

**Jean-Michel Poggi**[*]                                    *jean-michel.poggi@universite-paris-saclay.fr*
*Université Paris-Saclay, CNRS, Inria, Laboratoire de mathématiques d'Orsay, Orsay, France*
*Université Paris Cité, Paris, France*

[*]Equal contribution

**Reviewed on OpenReview:** *https://openreview.net/forum?id=lBDEZiW7MX*

## Abstract

We consider conformal prediction for multivariate data and focus on hierarchical data, where some components are linear combinations of others. Intuitively, the hierarchical structure can be leveraged to reduce the size of prediction regions for the same coverage level. We implement this intuition by including a projection step (also called a reconciliation step) in the split conformal prediction [SCP] procedure, and prove that the resulting prediction regions are indeed globally smaller. We do so both under the classic objective of joint coverage and under a new and challenging task: component-wise coverage, for which efficiency results are more difficult to obtain. The associated strategies and their analyses are based both on the literature of SCP and of forecast reconciliation, which we connect. We also illustrate the theoretical findings, for different scales of hierarchies on simulated data. The code to reproduce the experiments is available on GitHub.

## 1 Introduction

This article combines two post-hoc procedures (two procedures that are applied after initial forecasts were computed): conformal prediction and forecast reconciliation for hierarchical data, both in a regression setting.

### 1.1 Motivation

Hierarchical data arise in many domains of applications, where values to be predicted are organized into basic categories that sum up to or may be aggregated into higher-level categories. Two examples include household

expenditure surveys, where spending on food, housing, taxes, etc., add up to the total household expenditure, or time series such as energy consumption data (Brégère & Huard, 2022), which are recorded at different geographic granularities. Such hierarchical structures are actually ubiquitous in various domains (see the review article by Athanasopoulos et al., 2024), including economics, demography, retail sales and supply chain, energy forecasting. Forecasting methods were introduced to take into account the hierarchical structure directly in the training phase, which could be computationally extensive and reduce the flexibility in the choice of the model (Doumèche et al., 2025). Therefore, the present study is based on an alternative line of research known as *forecast reconciliation.*

The aim of forecast reconciliation is to exploit the linear relationships defining the hierarchy in a post-hoc manner, after some initial forecasts were produced, so as to ensure that the final forecasts are *coherent* across levels; this is often achieved via projections (not necessarily orthogonal ones). This idea was shown to be effective for improving the accuracy of point forecasts. However, extending these techniques to probabilistic forecasting remains a major challenge, despite the increasing importance of predictive uncertainty in modern decision-making (Gneiting & Katzfuss, 2014).

In parallel, *conformal prediction* provides a general and model-agnostic framework for constructing finite-sample valid prediction regions from any underlying forecasting method. Both forecast reconciliation and conformal prediction are post-hoc procedures applied after point forecasts are obtained. This conceptual similarity naturally suggests combining them to produce improved, hierarchy-aware prediction regions.

The goal of this work is to investigate this combination from a theoretical standpoint. While the setting considered here encompasses a wide range of applications, it should be noted that many practical instances of hierarchical data take the form of time series. Such cases typically lead to violations of the i.i.d. assumption on non-conformity scores adopted in this article. Hence, the present analysis, developed under a favorable i.i.d. framework, establishes a theoretical basis that can guide future developments in more general settings.

## 1.2 Related work

**Forecast reconciliation.** Forecast reconciliation aims to exploit the hierarchical structure so as to improve the quality and coherence of forecasts. The guiding intuition is that aggregate quantities, located higher in the hierarchy, are often easier to predict, and that these forecasts can be used to refine those at lower levels (Athanasopoulos et al., 2024). Conversely, local forecasts may convey valuable disaggregated information that can improve higher-level predictions. A central line of work (Hyndman et al., 2011; Wickramasuriya et al., 2019; Panagiotelis et al., 2021) approaches reconciliation through the scope of projections onto the subspace of so-called coherent forecasts; see Appendix C for additional background. More recently, probabilistic extensions of forecast reconciliation have been developed. In particular, Wickramasuriya (2024) studied reconciliation methods for Gaussian predictive distributions and Panagiotelis et al. (2023) proposed a general optimization-based framework, where reconciled forecasts are obtained by minimizing a proper scoring rule through gradient descent. However, we did not leverage results from these probabilistic extensions to build our own approach.

**Conformal prediction.** Conformal prediction is a general framework for constructing prediction sets with finite-sample coverage guarantees, based on any underlying forecasting method and under mild assumptions– typically, exchangeability of the data. It was first formalized by Vovk et al. (2005) and has gained attention since the work of Lei et al. (2018).

Recent developments have extended conformal prediction to multivariate settings, where the main challenge lies in accounting for dependencies among components. Existing approaches include copula-based methods (Messoudi et al., 2021), directional quantile regression (Feldman et al., 2023), and optimal-transport-based formulations (Klein et al., 2026; Thurin et al., 2025). This literature focuses on constructing joint prediction regions that target joint coverage. Among these, ellipsoidal prediction regions proposed by Johnstone & Cox (2021) and Messoudi et al. (2022) offer a particularly tractable formulation. Although our main interest lies in component-wise coverage, we also consider joint coverage for completeness, by adapting the ellipsoidal approach of Johnstone & Cox (2021) and Messoudi et al. (2022) to hierarchical data (see Section 2.3.1).

**Terminology clarification.** The term "hierarchical" has also appeared in the context of conformal prediction, with a different meaning. For instance, Lee et al. (2026), Dunn et al. (2023), and Duchi et al. (2025) study settings involving data from multiple sources or environments rather than hierarchical aggregation constraints. The latter work explicitly uses the term "multi-environment" to avoid confusion. Similarly, in classification, Mortier et al. (2026) consider a hierarchy over possible classes, which is unrelated to the hierarchical setting we consider and describe in Section 2. Please note that the concept of hierarchical data is also not related to "hierarchical modeling" (Banerjee et al., 2014).

## 1.3 Contributions and challenges

This work combines, for the first time, conformal prediction and forecast reconciliation to construct valid and efficient prediction regions for hierarchical data. We view this combination as a natural synthesis of two post-hoc procedures: conformal prediction provides distribution-free coverage guarantees, while forecast reconciliation exploits the hierarchical structure to improve efficiency and coherence.

**Main contributions.** Our contributions can be summarized as follows:

- **Joint-coverage setting.** We first revisit the classical ellipsoidal conformal prediction method and extend it to hierarchical data to derive an elementary efficiency result whenever joint-coverage is targeted.

- **Component-wise coverage.** We introduce a new criterion of component-wise coverage, more natural for hierarchical data than joint coverage; indeed, for hierarchical data, uncertainty is typically quantified in a component-wise manner (Taieb et al., 2017) since each component can (and perhaps, should) be studied under individual scrutiny. The benchmark method considered is the split conformal prediction procedure (Lei et al., 2018) applied component-wise to signed non-conformity scores as in Linusson et al. (2014).

- **Reconciled conformal prediction with efficiency guarantees.** We propose an improved prediction procedure that differs from the above benchmark by an additional reconciliation step, consisting of a projection applied to non-conformity scores. We show that, for a given coverage level, the reconciled prediction regions leverage the hierarchical structure of the data and are smaller–in a sense made precise–than those obtained without reconciliation. This constitutes one of the few efficiency results for conformal prediction.

**Technical challenges.** Establishing these results requires bridging tools from the two literatures. We rely on known trace inequalities from the forecast-reconciliation framework and introduce new ones tailored to our conformal setting. A detailed discussion of these technical innovations is provided in Appendix C.3.

## 1.4 Outline

In Section 2, we formally state the settings considered, the objectives targeted, and the methodologies followed. The objectives consist of either joint-coverage guarantees or component-wise-coverage guarantees, with associated efficiency results. The methodology consists of taking extensions of the split conformal procedure [SCP] as benchmarks: we show how to improve on them by adding reconciliation steps through projections. The analysis is straightforward for joint coverage, see Section 3. Our core theoretical results concern the component-wise analysis, reported in Section 4: the immediate component-wise coverage guarantees are stated in Theorem 2, and the efficiency results, which are our main results, are stated next (weak and practical version in Theorem 3, strong and oracle version in Theorem 4). We only provide a sketch of the proof of Theorem 3 (highlighting how we connected the tools of conformal prediction with the ones of forecast reconciliation), and defer full proofs of all results in appendices. Finally, Section 5 illustrates the theoretical findings on artificial data, again with full details in appendices.

**Notation.** For an integer $n \geqslant 1$, let $[n] = \{1, \ldots, n\}$. We denote by $\lfloor x \rfloor$ and $\lceil x \rceil$ the lower and upper integer parts of a real number $x \geqslant 0$. We let $\mathfrak{L}_m$ be the Lebesgue measure over $\mathbb{R}^m$. For a vector $\boldsymbol{u} \in \mathbb{R}^m$ and $n \leqslant m$,

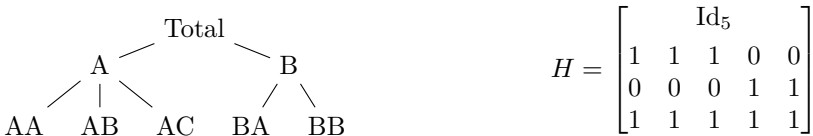

Figure 1: An example of a hierarchical structure (Total=A+B; A=AA+AB+AC; B=BA+BB) with its associated structural matrix $H$.

let $\boldsymbol{u}_{1:n} = (u_1, \ldots, u_n)^\top$ be the vector of the first $n$ components of $\boldsymbol{u}$. The null vector of $\mathbb{R}^m$ is denoted by $\boldsymbol{0} = (0, \ldots, 0)^\top$. We let $\mathrm{diag}(\boldsymbol{w})$ denote the $m \times m$ diagonal matrix with diagonal elements given by $\boldsymbol{w} \in \mathbb{R}^m$. We denote by $\mathrm{Id}_m$ the $m \times m$ identity matrix. The trace of a square matrix $M$ is denoted by $\mathrm{Tr}(M)$. The image of a matrix $H$ is denoted by $\mathrm{Im}(H)$. For a symmetric definite positive matrix $A$ of size $m \times m$, we denote by $\|\boldsymbol{u}\|_A = \sqrt{\boldsymbol{u}^\top A \boldsymbol{u}}$ the $A$–norm of a vector $\boldsymbol{u} \in \mathbb{R}^m$.

## 2 Objectives and methodology

**Setting.** We consider a multivariate regression problem of observations $\boldsymbol{y} \in \mathbb{R}^m$, where $m \geqslant 3$, based on features $\boldsymbol{x} \in \mathbb{R}^d$, where $d \geqslant 1$. The observations enjoy some hierarchical structure: some of their components (henceforth referred to as aggregated levels) are given by sums over subsets of other components (henceforth referred to as the most disaggregated level). More formally, up to reordering the components of $\boldsymbol{y}$, there exist $2 \leqslant n < m$ and a $m \times n$ matrix $H$ of the form

$$H = \begin{bmatrix} \mathrm{Id}_n \\ H_{\mathrm{sub}} \end{bmatrix} \quad \text{such that} \quad \boldsymbol{y} = H \boldsymbol{y}_{1:n},$$

where $H_{\mathrm{sub}}$ is any $(m-n) \times n$ matrix of real numbers. The matrix $H$ encoding the hierarchical summation constraints is called the structural[1] matrix. An example is provided in Figure 1, of a tree-like hierarchy (i.e., with a matrix $H_{\mathrm{sub}}$ of some specific form, but we recall that we will require no specific assumption on $H_{\mathrm{sub}}$).

**Definition 1.** Vectors $\boldsymbol{u} \in \mathbb{R}^m$ satisfying the linear constraints $\boldsymbol{u} = H \boldsymbol{u}_{1:n}$ are called coherent. The subspace $\mathrm{Im}(H)$ of all such vectors is called the coherent subspace.

Intuitively, coherence means that the values at the aggregated levels of the hierarchy are consistent with those observed or predicted at the most disaggregated level. Except for the cases where the training procedure outputs coherent forecasts by design (which, as mentioned in Section 1.1, is computationally demanding, see Doumèche et al., 2025), the forecasts are unlikely to be coherent.

**Example 1.** Following Brégère & Huard (2022), consider the joint prediction of electricity load consumption at regional and national levels based on meteorological and calendar features. The regional consumptions correspond to the most disaggregated components, and the national consumption is obtained by summing the former. This is a simple hierarchy with two levels only.

*Remark* 1. The structural matrix $H$ is fully known and available to the learner, as it is determined by the data and the practitioner's objectives.

**Additional terminology.** Observations of the form above with $n < m$ will be called hierarchical. When $n = m$, i.e., $H$ is the identity matrix, we will use the terminology of (plain) multivariate observations.

### 2.1 Objectives, part 1: joint coverage vs. component-wise coverage guarantees

We are interested in regressing hierarchical observations based on contextual information (that is not necessarily hierarchical). As the form of data will be fixed throughout the article, we state it as a global assumption.

**Assumption 1** (global assumption). We consider data $(\boldsymbol{x}_t, \boldsymbol{y}_t)_{1 \leqslant t \leqslant T}$ given by pairs of features $\boldsymbol{x}_t$ and of hierarchical observations $\boldsymbol{y}_t$.

---

[1]In the literature of forecast reconciliation, this matrix is usually denoted by $S$. We rather keep this letter for non-conformity scores. Also, the most disaggregated level is often composed by the last $n$ components, while we consider the first $n$ components.

For now, we assume that this data is formed by i.i.d. pairs (for the sake of exposition; this assumption will later be slightly relaxed). The primary objective is to construct prediction sets $C$ based on this $T$–sample, where $C : \boldsymbol{x} \in \mathbb{R}^d \mapsto C(\boldsymbol{x})$ is an application taking subsets of $\mathbb{R}^m$ as values. Consider a new data point $(\boldsymbol{x}_{T+1}, \boldsymbol{y}_{T+1})$ i.i.d. from the $T$–sample. Coverage guarantees refer to controlling probabilities of the form $\mathbb{P}\big(\psi(\boldsymbol{y}_{T+1}) \in \psi(C(\boldsymbol{x}_{T+1}))\big)$, where $\psi$ may be the identity or the extraction of a given component, and where the probability $\mathbb{P}$ is with respect to both $(\boldsymbol{x}_{T+1}, \boldsymbol{y}_{T+1})$ and $(\boldsymbol{x}_t, \boldsymbol{y}_t)_{1 \leqslant t \leqslant T}$. We set some miscoverage level $\alpha \in (0, 1)$.

**Joint coverage.** This is the typical objective in other contributions on (plain) multivariate conformal prediction (see Johnstone & Cox, 2021, Messoudi et al., 2021; 2022, Feldman et al., 2023) and corresponds to $\psi$ being the identity:
$$\mathbb{P}\big(\boldsymbol{y}_{T+1} \in C(\boldsymbol{x}_{T+1})\big) \approx 1 - \alpha \, .$$

The prediction regions $C(\boldsymbol{x}_{T+1})$ are typically ellipsoidal in the scope of Section 2.3.1, and in any case, their general shapes are not pre-specified (and should even be optimized by the learning strategies).

**Component-wise coverage.** This coverage objective corresponds to considering all extractions of components for the functions $\psi$, i.e., simultaneous individual coverage guarantees are targeted. Therefore, the prediction set is given by a Cartesian product: $C = C_1 \times \ldots \times C_m$, where each $C_i : \boldsymbol{x} \in \mathbb{R}^d \mapsto C_i(\boldsymbol{x})$ is an application taking subsets of $\mathbb{R}$ as values, for $i \in [m]$. These individual prediction sets $C_i$ should be designed in such a way that each component $y_{T+1,i}$ of the observations $\boldsymbol{y}_{T+1}$ is in $C_i(\boldsymbol{x}_{T+1})$ with probability approximately $1 - \alpha$:
$$\forall i \in [m], \quad \mathbb{P}\big(y_{T+1,i} \in C_i(\boldsymbol{x}_{T+1})\big) \approx 1 - \alpha \, .$$

## 2.2 Objectives, part 2: efficiency

A secondary objective is to ensure that the prediction sets are efficient, i.e., are as small as possible.

**Efficiency criterion under joint coverage constraints.** Joint coverage corresponds to evaluating performance at a global level. It is therefore natural to measure efficiency in terms of volumes, as given by the Lebesgue measure $\mathfrak{L}_m$ over $\mathbb{R}^m$, and to
$$\text{minimize} \quad \mathbb{E}\Big[\mathfrak{L}_m\big(C(\boldsymbol{x}_{T+1})\big)\Big] \, ,$$

where again, the expectation is with respect to both $(\boldsymbol{x}_{T+1}, \boldsymbol{y}_{T+1})$ and $(\boldsymbol{x}_t, \boldsymbol{y}_t)_{1 \leqslant t \leqslant T}$.

We show in Theorem 5 that our approach satisfies this objective, and actually provides a stronger result of uniform domination: a prediction region $C$ is uniformly more efficient than a prediction region $C'$ if $\mathfrak{L}_m\big(C(\boldsymbol{x})\big) \leqslant \mathfrak{L}_m\big(C'(\boldsymbol{x})\big)$ for all $\boldsymbol{x} \in \mathbb{R}^d$.

**Efficiency criterion under component-wise coverage constraints.** In this case, components are under individual scrutiny and some may be more important than others. This is why a vector $\boldsymbol{w} = (w_1, \ldots, w_m)^\top$ of positive numbers may be used to weight the components based on their respective importance. One quantification of the size of the Cartesian product $C(\boldsymbol{x}) = C_1(\boldsymbol{x}) \times \ldots \times C_m(\boldsymbol{x})$ is then given by the sum of the $w_i \, \mathfrak{L}_1\big(C_i(\boldsymbol{x})\big)^2$ over $i \in [m]$, where $\mathfrak{L}_1$ denotes the Lebesgue measure over $\mathbb{R}$. Formally, the corresponding efficiency objective corresponds to
$$\text{minimizing} \quad \mathbb{E}\left[\sum_{i=1}^m w_i \, \mathfrak{L}_1\big(C_i(\boldsymbol{x}_{T+1})\big)^2\right] \, ,$$

where, again, the expectation $\mathbb{E}$ is with respect to both $(\boldsymbol{x}_{T+1}, \boldsymbol{y}_{T+1})$ and $(\boldsymbol{x}_t, \boldsymbol{y}_t)_{1 \leqslant t \leqslant T}$.

## 2.3 Methodology: split conformal prediction [SCP], performed jointly or component-wise

Split conformal prediction [SCP] (Lei et al., 2018), which is a reformulation of inductive conformal prediction (Vovk et al., 2005), is a procedure based on splitting data indexed by $[T]$ between a training set indexed by $\mathcal{D}_{\text{train}}$ (to learn a regressor function), possibly an estimation set indexed by $\mathcal{D}_{\text{estim}}$ (typically to learn

some parameters for the evaluation), and a calibration set indexed by $\mathcal{D}_{\text{calib}}$ (to compute estimation errors, a.k.a. residuals or non-conformity scores). With pairs $(\boldsymbol{x}_t, \boldsymbol{y}_t)$ indexed by $t \in \mathcal{D}_{\text{train}}$, a regressor function $\widehat{\boldsymbol{\mu}} : \boldsymbol{x} \in \mathbb{R}^d \mapsto \widehat{\boldsymbol{\mu}}(\boldsymbol{x}) \in \mathbb{R}^m$ is built, thanks to some regression algorithm $\mathcal{A}$ provided as input parameter to the SCP procedure. On the calibration set, i.e., for each $t \in \mathcal{D}_{\text{calib}}$, point estimates $\widehat{\boldsymbol{y}}_t = \widehat{\boldsymbol{\mu}}(\boldsymbol{x}_t)$ and associated non-conformity scores are computed. The way the latter are defined depends on the specific setting and objectives, as detailed below. We denote by $T_{\text{train}}, T_{\text{estim}}, T_{\text{calib}}$ the respective cardinalities of $\mathcal{D}_{\text{train}}, \mathcal{D}_{\text{estim}}, \mathcal{D}_{\text{calib}}$. The SCP procedure has been extensively studied in the univariate case, and often through considering the absolute values of the residuals as non-conformity scores, which leads to centered intervals. We are interested in two extensions of this basic setting: multivariate SCP based on ellipsoidal sets, and component-wise signed non-conformity scores.

### 2.3.1 Multivariate SCP for joint coverage based on ellipsoidal sets

Multivariate SCP based on ellipsoidal sets was already studied by Johnstone & Cox (2021) and Messoudi et al. (2022) for plain multivariate data. The key is to consider $A$–norms $\|\cdot\|_A$ of estimation errors, where $A$ is a positive definite data-based matrix designed to capture the potential multivariate dependencies of the targets. The matrix $A$ is learned on the estimation set $\mathcal{D}_{\text{estim}}$ (possibly also based on data from $\mathcal{D}_{\text{train}}$, like $\widehat{\boldsymbol{\mu}}$) and its choice is critical for practical purposes. A typical choice is $A = \widehat{\Sigma}^{-1}$, where $\widehat{\Sigma}$ is some estimated covariance matrix of the residuals $\boldsymbol{y}_t - \widehat{\boldsymbol{y}}_t$, where $t \in \mathcal{D}_{\text{estim}}$, and the Moore-Penrose pseudo-inverse is considered. To a certain extent, this approach may be considered a form of de-correlation. We denote by $\mathcal{E}$ the procedure outputting this matrix.

Scalar non-conformity scores given by $A$–norms are then computed on $\mathcal{D}_{\text{calib}}$:

$$\check{s}_t = \|\boldsymbol{y}_t - \widehat{\boldsymbol{y}}_t\|_A \overset{\text{def}}{=} \sqrt{(\boldsymbol{y}_t - \widehat{\boldsymbol{y}}_t)^\top A (\boldsymbol{y}_t - \widehat{\boldsymbol{y}}_t)}.$$

These scores $(\check{s}_t)_{t \in \mathcal{D}_{\text{calib}}}$ are ordered into $\check{s}_{(1)} \leqslant \ldots \leqslant \check{s}_{(T_{\text{calib}})}$, where we used the notation of order statistics. We define $\check{s}_{(0)} = 0$ and $\check{s}_{(T_{\text{calib}}+1)} = +\infty$. The resulting prediction ellipsoid is $\check{E}(\boldsymbol{x}_{T+1})$:

$$\check{q}_{1-\alpha} = \check{s}_{\left(\lceil (T_{\text{calib}}+1)(1-\alpha) \rceil\right)} \quad \text{and} \quad \check{E} : \boldsymbol{x} \in \mathbb{R}^d \longmapsto \left\{ \boldsymbol{y} \in \mathbb{R}^m : \|\boldsymbol{y} - \widehat{\boldsymbol{\mu}}(\boldsymbol{x})\|_A \leqslant \check{q}_{1-\alpha} \right\}. \tag{1}$$

For the convenience of the reader, Algorithm (1) described above and Algorithm (2) discussed right below are summarized in algorithm boxes in Appendix D.

**Hierarchical SCP for joint coverage.** We adapt the above to hierarchical data with the orthogonal projection $P_A$ in $A$–norm onto $\text{Im}(H)$, which equals $P_A = H(H^\top A H)^{-1} H^\top A$ (see Lemma 5 in Appendix A.2), and by considering rather the regressor function $\mathring{\boldsymbol{\mu}}(\cdot) = P_A \widehat{\boldsymbol{\mu}}(\cdot)$. Based on this, predictions $\mathring{\boldsymbol{y}}_t = \mathring{\boldsymbol{\mu}}(\boldsymbol{x}_t)$ and non-conformity scores $\mathring{s}_t = \|\boldsymbol{y}_t - \mathring{\boldsymbol{y}}_t\|_A$ are computed for $t \in \mathcal{D}_{\text{calib}}$, they are ordered into $\mathring{s}_{(1)} \leqslant \ldots \leqslant \mathring{s}_{(T_{\text{calib}})}$, and the resulting prediction ellipsoid is $\mathring{E}(\boldsymbol{x}_{T+1})$, where

$$\mathring{q}_{1-\alpha} = \mathring{s}_{\left(\lceil (T_{\text{calib}}+1)(1-\alpha) \rceil\right)} \quad \text{and} \quad \mathring{E} : \boldsymbol{x} \in \mathbb{R}^d \longmapsto \left\{ \boldsymbol{y} \in \mathbb{R}^m : \|\boldsymbol{y} - \mathring{\boldsymbol{\mu}}(\boldsymbol{x})\|_A \leqslant \mathring{q}_{1-\alpha} \right\}. \tag{2}$$

### 2.3.2 Component-wise SCP for component-wise coverage

Signed non-conformity scores were already considered in the univariate case by Linusson et al. (2014). They are handy in our setting because we consider linear constraints: the signed non-conformity scores $\widehat{\boldsymbol{s}}_t = \boldsymbol{y}_t - \widehat{\boldsymbol{y}}_t$ between coherent observations $\boldsymbol{y}_t$ and forecasts $\widehat{\boldsymbol{y}}_t$ are also coherent, while the vector of their absolute values is not coherent in general.

**Component-wise SCP with signed scores.** When component-wise objectives are targeted, component-wise non-conformity scores should be considered. More precisely, we consider the procedure summarized in Algorithm 3 (where no estimation set is needed, for now), which first considers vector-valued estimation errors $\widehat{\boldsymbol{s}}_t = \boldsymbol{y}_t - \widehat{\boldsymbol{\mu}}(\boldsymbol{x}_t)$, and then builds separately prediction intervals for each component $i \in [m]$, based on the non-conformity scores $(\widehat{s}_{t,i})_{t \in \mathcal{D}_{\text{calib}}}$, in the same spirit as above. More precisely, these scores are ordered as $\widehat{s}_{(1),i} \leqslant \ldots \leqslant \widehat{s}_{(T_{\text{calib}}),i}$, we further define $\widehat{s}_{(0),i} = -\infty$ and $\widehat{s}_{(T_{\text{calib}}+1),i} = +\infty$, and output $\widehat{C}_i(\boldsymbol{x}_{T+1})$, where

$$\widehat{C}_i : \boldsymbol{x} \in \mathbb{R}^d \longmapsto \left[ \widehat{\mu}_i(\boldsymbol{x}) + \widehat{s}_{\left(\lfloor (T_{\text{calib}}+1)\alpha/2 \rfloor\right),i}, \widehat{\mu}_i(\boldsymbol{x}) + \widehat{s}_{\left(\lceil (T_{\text{calib}}+1)(1-\alpha/2) \rceil\right),i} \right],$$

---

**Algorithm 3** Plain component-wise SCP with signed scores

---

**Parameters:** confidence level $1 - \alpha$; regression algorithm $\mathcal{A}$; partition of $[T]$ into subsets $\mathcal{D}_{\mathrm{train}}$ and $\mathcal{D}_{\mathrm{calib}}$ of respective cardinalities $T_{\mathrm{train}}$ and $T_{\mathrm{calib}}$

1: Build the regressor $\widehat{\boldsymbol{\mu}}(\cdot) = \mathcal{A}\big((\boldsymbol{x}_t, \boldsymbol{y}_t)_{t \in \mathcal{D}_{\mathrm{train}}}\big)$ and denote $\widehat{\boldsymbol{\mu}}(\cdot) = \big(\widehat{\mu}_1(\cdot), \ldots, \widehat{\mu}_m(\cdot)\big)^\top$
2: **for** $t \in \mathcal{D}_{\mathrm{calib}}$ **do** let $\widehat{\boldsymbol{y}}_t = \widehat{\boldsymbol{\mu}}(\boldsymbol{x}_t)$ and compute the estimation errors $\widehat{\boldsymbol{s}}_t = \boldsymbol{y}_t - \widehat{\boldsymbol{y}}_t$
3: **for** each component $i \in [m]$ **do**
4:      order the $(\widehat{s}_{t,i})_{t \in \mathcal{D}_{\mathrm{calib}}}$ into $\widehat{s}_{(1),i} \leqslant \ldots \leqslant \widehat{s}_{(T_{\mathrm{calib}}),i}$; set $\widehat{s}_{(0),i} = -\infty$ and $\widehat{s}_{(T_{\mathrm{calib}}+1),i} = +\infty$
5:      let $\widehat{q}^{(i)}_{\alpha/2} = \widehat{s}_{\big(\lfloor (T_{\mathrm{calib}}+1)\alpha/2 \rfloor\big),i}$ and $\widehat{q}^{(i)}_{1-\alpha/2} = \widehat{s}_{\big(\lceil (T_{\mathrm{calib}}+1)(1-\alpha/2) \rceil\big),i}$
6:      set $\widehat{C}_i(\cdot) = \Big[\widehat{\mu}_i(\cdot) + \widehat{q}^{(i)}_{\alpha/2}, \ \widehat{\mu}_i(\cdot) + \widehat{q}^{(i)}_{1-\alpha/2}\Big]$ and **return** $\widehat{C}_i(\boldsymbol{x}_{T+1})$

---

**Algorithm 4** Hierarchical component-wise SCP with signed scores

---

**Parameters:** confidence level $1 - \alpha$; regression algorithm $\mathcal{A}$; partition of $[T]$ into subsets $\mathcal{D}_{\mathrm{train}}$ and $\mathcal{D}_{\mathrm{calib}}$ of respective cardinalities $T_{\mathrm{train}}$ and $T_{\mathrm{calib}}$; matrix $P$

1: Build the regressor $\widehat{\boldsymbol{\mu}}(\cdot) = \mathcal{A}\big((\boldsymbol{x}_t, \boldsymbol{y}_t)_{t \in \mathcal{D}_{\mathrm{train}}}\big)$ and let $\widetilde{\boldsymbol{\mu}}(\cdot) = P\widehat{\boldsymbol{\mu}}(\cdot) = \big(\widetilde{\mu}_1(\cdot), \ldots, \widetilde{\mu}_m(\cdot)\big)^\top$
2: **for** $t \in \mathcal{D}_{\mathrm{calib}}$ **do** let $\widetilde{\boldsymbol{y}}_t = \widetilde{\boldsymbol{\mu}}(\boldsymbol{x}_t)$ and compute the estimation errors $\widetilde{\boldsymbol{s}}_t = \boldsymbol{y}_t - \widetilde{\boldsymbol{y}}_t$
3: **for** each component $i \in [m]$ **do**
4:      order the $(\widetilde{s}_{t,i})_{t \in \mathcal{D}_{\mathrm{calib}}}$ into $\widetilde{s}_{(1),i} \leqslant \ldots \leqslant \widetilde{s}_{(T_{\mathrm{calib}}),i}$; set $\widetilde{s}_{(0),i} = -\infty$ and $\widetilde{s}_{(T_{\mathrm{calib}}+1),i} = +\infty$
5:      let $\widetilde{q}^{(i)}_{\alpha/2} = \widetilde{s}_{\big(\lfloor (T_{\mathrm{calib}}+1)\alpha/2 \rfloor\big),i}$ and $\widetilde{q}^{(i)}_{1-\alpha/2} = \widetilde{s}_{\big(\lceil (T_{\mathrm{calib}}+1)(1-\alpha/2) \rceil\big),i}$
6:      set $\widetilde{C}_i(\cdot) = \Big[\widetilde{\mu}_i(\cdot) + \widetilde{q}^{(i)}_{\alpha/2}, \ \widetilde{\mu}_i(\cdot) + \widetilde{q}^{(i)}_{1-\alpha/2}\Big]$ and **return** $\widetilde{C}_i(\boldsymbol{x}_{T+1})$

---

**Algorithm 5** Hierarchical component-wise SCP with signed scores and data-based projection matrix

---

**Parameters:** confidence level $1 - \alpha$; regression algorithm $\mathcal{A}$; partition of $[T]$ into three subsets $\mathcal{D}_{\mathrm{train}}$, $\mathcal{D}_{\mathrm{estim}}$ and $\mathcal{D}_{\mathrm{calib}}$ of respective cardinalities $T_{\mathrm{train}}$, $T_{\mathrm{estim}}$ and $T_{\mathrm{calib}}$; function $\mathcal{P}$ with values given by $m \times m$ matrices

1: Build the regressor $\widehat{\boldsymbol{\mu}}(\cdot) = \mathcal{A}\big((\boldsymbol{x}_t, \boldsymbol{y}_t)_{t \in \mathcal{D}_{\mathrm{train}}}\big)$
2: Build the projection matrix $P = \mathcal{P}\big(\widehat{\boldsymbol{\mu}}, (\boldsymbol{x}_t, \boldsymbol{y}_t)_{t \in \mathcal{D}_{\mathrm{estim}}}\big)$ and let $\widetilde{\boldsymbol{\mu}}(\cdot) = P\widehat{\boldsymbol{\mu}}(\cdot)$
3: **for** $t \in \mathcal{D}_{\mathrm{calib}}$ **do** let $\widetilde{\boldsymbol{y}}_t = \widetilde{\boldsymbol{\mu}}(\boldsymbol{x}_t)$ and compute the estimation errors $\widetilde{\boldsymbol{s}}_t = \boldsymbol{y}_t - \widetilde{\boldsymbol{y}}_t$
4: **for** each component $i \in [m]$ **do**
5:      order the $(\widetilde{s}_{t,i})_{t \in \mathcal{D}_{\mathrm{calib}}}$ into $\widetilde{s}_{(1),i} \leqslant \ldots \leqslant \widetilde{s}_{(T_{\mathrm{calib}}),i}$; set $\widetilde{s}_{(0),i} = -\infty$ and $\widetilde{s}_{(T_{\mathrm{calib}}+1),i} = +\infty$
6:      let $\widetilde{q}^{(i)}_{\alpha/2} = \widetilde{s}_{\big(\lfloor (T_{\mathrm{calib}}+1)\alpha/2 \rfloor\big),i}$ and $\widetilde{q}^{(i)}_{1-\alpha/2} = \widetilde{s}_{\big(\lceil (T_{\mathrm{calib}}+1)(1-\alpha/2) \rceil\big),i}$
7:      set $\widetilde{C}_i(\cdot) = \Big[\widetilde{\mu}_i(\cdot) + \widetilde{q}^{(i)}_{\alpha/2}, \ \widetilde{\mu}_i(\cdot) + \widetilde{q}^{(i)}_{1-\alpha/2}\Big]$ and **return** $\widetilde{C}_i(\boldsymbol{x}_{T+1})$

---

where $\widehat{\mu}_i(\boldsymbol{x}_{T+1})$ is the $i$–th component of the point estimate $\widehat{\boldsymbol{\mu}}(\boldsymbol{x}_{T+1})$. The thus-defined (plain) component-wise SCP with signed scores is summarized in Algorithm 3. We use it as a benchmark and now introduce a generalization of this algorithm taking the hierarchical structure $H$ into account.

**Hierarchical component-wise SCP with signed scores.** The hierarchical version of SCP is stated in Algorithm 4 and only differs from the plain multivariate version stated as Algorithm 3 in line 1, where a projection matrix $P$ onto the coherent subspace $\mathrm{Im}(H)$ should be used: the regressor function considered is $\widetilde{\boldsymbol{\mu}} = P\widehat{\boldsymbol{\mu}}$, instead of simply $\widehat{\boldsymbol{\mu}}$, and thus outputs point estimates that are coherent in the case where $\mathrm{Im}(P) \subseteq \mathrm{Im}(H)$. The rest of the procedure is similar.

Algorithm 3 is a special case of Algorithm 4, for the choice $P = \mathrm{Id}_m$. We however provide two separate statements to clarify the notation: $\widehat{\cdot}$–type quantities are for the plain multivariate version (Algorithm 3),

which we use as a benchmark, while $\widetilde{\cdot}$–type quantities refer to their reconciled versions (as in Algorithms 4 and 5), obtained by projection onto $\mathrm{Im}(H)$.

**Hierarchical component-wise SCP with signed scores and data-based projection matrix.** The matrix $A$ for multivariate SCP based on ellipsoidal sets for joint coverage may be learned on an estimation set $\mathcal{D}_{\mathrm{estim}}$, and we may well do so also for the projection matrix $P$. This leads to Algorithm 5, which is a generalization of Algorithm 4 and for which modifications to the latter are stated in blue. Typical functions $\mathcal{P}$ (examples are provided in Section 4.3) use $\widehat{\Sigma}$, an estimated covariance matrix of the residuals $\boldsymbol{y}_t - \widehat{\boldsymbol{y}}_t$, where $t \in \mathcal{D}_{\mathrm{estim}}$:

$$\overline{\boldsymbol{s}} = \frac{1}{T_{\mathrm{estim}}} \sum_{t \in \mathcal{D}_{\mathrm{estim}}} \widehat{\boldsymbol{s}}_t \quad \text{and} \quad \widehat{\Sigma} = \frac{1}{T_{\mathrm{estim}}} \sum_{t \in \mathcal{D}_{\mathrm{estim}}} \left(\widehat{\boldsymbol{s}}_t - \overline{\boldsymbol{s}}\right)\left(\widehat{\boldsymbol{s}}_t - \overline{\boldsymbol{s}}\right)^{\top} . \tag{3}$$

# 3 Analysis of hierarchical SCP through ellipsoidal sets for joint coverage

This section only shows how straightforward it is to take the hierarchy into account in this setting. We state informally the results achieved. Formal statements and short proofs thereof may be found in Appendices D and E.

In conformal prediction, results typically hold in great generality. In particular, we will require no direct assumption on the regression algorithm $\mathcal{A}$, which will be treated as a black-box regression procedure that does not even have to output coherent point estimates (hence the consideration of projection matrices).

**Theorem 1** (informal statement of Theorems 5 and 6). *Algorithms* (1) *and* (2), *used with any regression algorithm $\mathcal{A}$ and any estimation procedure $\mathcal{E}$, ensure that whenever data $(\boldsymbol{x}_t, \boldsymbol{y}_t)_{1 \leqslant t \leqslant T+1}$ is i.i.d.,*

$$\mathbb{P}\big(\boldsymbol{y}_{T+1} \in \check{E}(\boldsymbol{x}_{T+1})\big) \approx 1 - \alpha \qquad \text{and} \qquad \mathbb{P}\big(\boldsymbol{y}_{T+1} \in \mathring{E}(\boldsymbol{x}_{T+1})\big) \approx 1 - \alpha .$$

*In addition, and under no assumption on the data, Algorithm* (2) *outputs prediction ellipsoids $\mathring{E}$ that are uniformly more efficient than the prediction ellipsoids $\check{E}$ output by Algorithm* (1)*:*

$$\forall \boldsymbol{x} \in \mathbb{R}^d, \qquad \mathfrak{L}_m\big(\mathring{E}(\boldsymbol{x})\big) \leqslant \mathfrak{L}_m\big(\check{E}(\boldsymbol{x})\big) .$$

*Remark* 2. Our approach involves projection matrices that, by definition, leave coherent forecasts unchanged. Therefore, the inequality in the theorem above (and in each of the subsequent efficiency results) is not strict in general: it is an equality when the forecasts output by the regression algorithm $\mathcal{A}$ are already coherent. However, we recall that we are not ready to assume this, see Section 1.1.

# 4 Analysis of hierarchical component-wise SCP algorithms

Coverage guarantees are immediate, under the typical assumptions. The standard proof of Theorem 2 (together with references to earlier similar proofs) may be found in Appendix E. Theorem 2 of course holds for Algorithms 3 and 4, given that they are special cases of Algorithm 5, using matrices $P$ satisfying $PH = H$, namely $P = \mathrm{Id}_m$ for Algorithm 3 and a projection onto $\mathrm{Im}(H)$ for algorithm 4.

**Assumption 2.** The residuals $\widehat{\boldsymbol{s}}_t = \boldsymbol{y}_t - \widehat{\boldsymbol{\mu}}(\boldsymbol{x}_t)$, for $t \in \mathcal{D}_{\mathrm{calib}} \cup \{T+1\}$, are i.i.d. This is in particular the case when data $(\boldsymbol{x}_t, \boldsymbol{y}_t)_{1 \leqslant t \leqslant T+1}$ is i.i.d.

**Theorem 2** (Coverage). *Fix $\alpha \in (0, 1)$. Algorithm 5, used with any regression algorithm $\mathcal{A}$ and any function $\mathcal{P}$ outputting matrices $P$ such that $PH = H$, ensures that whenever Assumption 2 (i.i.d. scores) holds,*

$$\forall i \in [m], \quad \mathbb{P}\big(y_{T+1,i} \in \widetilde{C}_i(\boldsymbol{x}_{T+1})\big) \geqslant 1 - \alpha .$$

*In addition, if the non-conformity scores $\big(\widetilde{\boldsymbol{s}}_t\big)_{t \in \mathcal{D}_{\mathrm{calib}} \cup \{T+1\}}$ are almost surely distinct, then*

$$\forall i \in [m], \quad \mathbb{P}\big(y_{T+1,i} \in \widetilde{C}_i(\boldsymbol{x}_{T+1})\big) \leqslant 1 - \alpha + 2/(T_{\mathrm{calib}} + 1) .$$

### 4.1 Efficiency results: additional assumption

Efficiency results rely on an additional assumption on the distribution of scores. We explain below in detail why this assumption makes sense, even though this assumption goes against the distribution-free gist of conformal prediction. We also underline that the aforementioned coverage guarantees did not require this distributional assumption and explain that previous efficiency results for conformal prediction all required additional assumptions of the same nature.

**Definition 2.** A random vector $\boldsymbol{z}$ follows a spherical distribution over $\mathbb{R}^k$ if $\boldsymbol{z}$ and $\Gamma\boldsymbol{z}$ have the same distribution for all $k \times k$ orthogonal matrices $\Gamma$.

**Definition 3.** An elliptical distribution over $\mathbb{R}^m$ is of the form $\boldsymbol{c} + M\boldsymbol{z}$, for a deterministic vector $\boldsymbol{c} \in \mathbb{R}^m$, a $m \times k$ matrix $M$ such that $MM^\top$ has rank $k$, and a random vector $\boldsymbol{z}$ following a spherical distribution over $\mathbb{R}^k$.

A given spherical distribution thus generates a family $\mathcal{F}$ of elliptical distributions enjoying a stability property through affine transformations as expressed in Lemma 3 of Appendix A. The latter appendix provides more details on elliptical distributions (and in turn refers to Kollo & von Rosen, 2005, Section 2.3).

**Examples.** The simplest example of elliptical distributions consists of multivariate normal distributions (which are light-tailed distributions). Other examples include multivariate $t$–distributions and symmetric multivariate Laplace distributions (both heavy tailed) and the uniform distribution on an ellipse (no tail).

**Assumption 3.** The (i.i.d.) residuals $\widehat{\boldsymbol{s}}_t = \boldsymbol{y}_t - \widehat{\boldsymbol{\mu}}(\boldsymbol{x}_t)$, for $t \in \mathcal{D}_{\text{calib}}$, follow some elliptical distribution (whose shape and parameters are unknown).

We justify in detail why this additional assumption may not be considered unnatural nor too restrictive compared to existing assumptions for efficiency results. **First,** it is used only for the efficiency results, not for the coverage results (Theorem 2), which remain distribution-free. Finite-sample efficiency results are scarce in conformal prediction and are achieved under additional assumptions—either on the model or on the non-conformity scores. For example, Burnaev & Vovk (2014) show that in a Gaussian model, conformal ridge regression achieves near-optimal efficiency. Lei et al. (2018) assume symmetry of the noise and stability under resampling and small perturbations of the base regressor to compare the conformal prediction bands with the oracle band. Le Bars & Humbert (2025) issued assumptions on the regression function (it has to be the minimizer over a rich class of functions $\mathcal{F}$ of the empirical $(1 - \alpha)$–quantile absolute error) to derive upper bounds on the lengths of prediction sets. **Second**, the very assumption of elliptical residuals appears explicitly in Henderson et al., 2024, Theorem 1.1, to show that conformal ellipsoidal sets are more efficient than conformal balls. We also believe that this assumption is implicit in Johnstone & Cox (2021); Messoudi et al. (2022); Xu et al. (2024), since targeting ellipsoidal prediction regions is only meaningful if the underlying distribution is (at least approximately) elliptical. **Third**, as written above, elliptical distributions form a broad family of distributions with diverse tail behaviors. The residuals are also often assumed elliptical in the literature of forecast reconciliation, as in Panagiotelis et al. (2023).

### 4.2 Efficiency results: weak version for a fixed vector $w$ of weights

We start with an efficiency result between the reconciled prediction sets $\widetilde{C}_i(\boldsymbol{x}_{T+1})$ output by Algorithm 4 and the plain prediction sets $\widehat{C}_i(\boldsymbol{x}_{T+1})$ output by Algorithm 3 for a single fixed vector $\boldsymbol{w}$ of weights.

**Theorem 3.** *Fix $\boldsymbol{w} \in (0, +\infty)^m$. Under Assumptions 2 and 3 (i.i.d. scores with elliptical distribution), the hierarchical component-wise SCP (Algorithm 4) run with $P = P_{\boldsymbol{w}}$, where*

$$P_{\boldsymbol{w}} \overset{\text{def}}{=} H\big(H^\top \operatorname{diag}(\boldsymbol{w})H\big)^{-1}H^\top \operatorname{diag}(\boldsymbol{w})\,, \tag{4}$$

*provides prediction sets that are more efficient than the ones output by the plain component-wise SCP (Algorithm 3) in the following sense:*

$$\mathbb{E}\left[\sum_{i=1}^m w_i\,\mathfrak{L}_1\big(\widetilde{C}_i(\boldsymbol{x}_{T+1})\big)^2\right] \leqslant \mathbb{E}\left[\sum_{i=1}^m w_i\,\mathfrak{L}_1\big(\widehat{C}_i(\boldsymbol{x}_{T+1})\big)^2\right]. \tag{5}$$

*Sketch of proof.* The centered residuals $\widetilde{\boldsymbol{s}}_t - \mathbb{E}\big[\widetilde{\boldsymbol{s}}_t\big]$ are i.i.d. according to a centered elliptical distribution as $t \in \mathcal{D}_{\text{calib}}$. Thus, their $i$–th components have the same distribution $\nu$ up to scaling factors denoted by $\sqrt{\gamma_i}$. Let $(v_t)_{t \in \mathcal{D}_{\text{calib}}}$ be i.i.d. variables distributed according to $\nu$, consider their order statistics $v_{(1)} \leqslant \ldots \leqslant v_{(T_{\text{calib}})}$, set $L_\alpha = v_{\big(\lceil (T_{\text{calib}}+1)(1-\alpha/2) \rceil\big)} - v_{\big(\lfloor (T_{\text{calib}}+1)\alpha/2 \rfloor\big)}$. Thus,

$$\mathbb{E}\left[ \sum_{i=1}^m w_i \ \mathfrak{L}_1\big(\widetilde{C}_i(\boldsymbol{x}_{T+1})\big)^2 \right] = \mathbb{E}\big[L_\alpha^2\big] \sum_{i \in [m]} w_i \gamma_i . \tag{6}$$

It may be shown that $\gamma_i$ is the $(i,i)$–th element of a matrix of the form $P_{\boldsymbol{w}} \Gamma P_{\boldsymbol{w}}^\top$, where $\Gamma$ is symmetric positive semi-definite. A similar result holds for the $\widehat{C}_i$, with scaling factors given by the diagonal elements of $\Gamma$. It thus suffices to show that

$$\sum_{i \in [m]} w_i \Gamma_{i,i} = \text{Tr}\big(\text{diag}(\boldsymbol{w})\, \Gamma\big) \geqslant \text{Tr}\big(\text{diag}(\boldsymbol{w})\, P_{\boldsymbol{w}} \Gamma P_{\boldsymbol{w}}^\top\big) = \sum_{i \in [m]} w_i \big(P_{\boldsymbol{w}} \Gamma P_{\boldsymbol{w}}^\top\big)_{i,i} .$$

The inequality above is a result of our own, though inspired by the literature of forecast reconciliation (e.g., Panagiotelis et al., 2021, Theorem 3.2). The complete proof may be found in Appendix A.

*Remark* 3. The same comments as in Remark 2 apply: no improvement is achieved when the forecasts output by the regression algorithm $\mathcal{A}$ are already coherent.

**Challenges overcome.** As detailed in Appendix C.3 the main hurdle in the proof above was to relate the minimization of squared lengths (6) to some trace minimization. Such relationships are classic in the literature of forecast reconciliation, but they rely on assumptions of unbiasedness (i.e., of centered residuals, which we preferred not to assume). Thanks to signed residuals, a cancellation takes place: the distribution of $L_\alpha$ is stable by translations of the $(v_t)_{t \in \mathcal{D}_{\text{calib}}}$.

### 4.3 Efficiency results: stronger but oracle version

We improve the result of Theorem 3 to have it hold simultaneously over all possible positive weight vectors $\boldsymbol{w}$. However, this improvement is only for an oracle strategy relying on a projection matrix $P_{\Sigma^{-1}}$ depending on the covariance matrix $\Sigma$ of the scores (unknown to the learner). Yet, our use of the covariance matrix does not involve de-correlating the scores, a process that would have contravened the distribution-free nature of conformal prediction. One way to see this is to note that $P_{\Sigma^{-1}}$ does not modify coherent forecasts, which are inherently correlated. We stated (see Algorithm 5) a "practical" implementation of this oracle, adding an estimation step for $\Sigma$.

**Assumption 4.** The (i.i.d.) residuals $\widehat{\boldsymbol{s}}_t = \boldsymbol{y}_t - \widehat{\boldsymbol{\mu}}(\boldsymbol{x}_t)$, for $t \in \mathcal{D}_{\text{calib}}$, have a bounded second-order moment, with positive definite covariance matrix $\Sigma$.

We crucially use the following minimum-trace result, that is central in the theory of forecast reconciliation (and provide an elementary proof thereof in Appendix B, of independent interest).

**Lemma 1** (Minimum-trace projection, Wickramasuriya et al., 2019). *Let $W$ and $\Sigma$ be two symmetric $m \times m$ matrices, where $W$ is positive semi-definite and $\Sigma$ is positive definite. Then, for all projection matrices $P$ onto $\text{Im}(H)$,*

$$\text{Tr}\big(W P_{\Sigma^{-1}} \Sigma P_{\Sigma^{-1}}^\top\big) \leqslant \text{Tr}\big(W P \Sigma P^\top\big), \qquad \text{where} \quad P_{\Sigma^{-1}} \overset{\text{def}}{=} H\big(H^\top \Sigma^{-1} H\big)^{-1} H^\top \Sigma^{-1} . \tag{7}$$

We denote by $\widetilde{C}_1^\star(\boldsymbol{x}_{T+1}) \times \ldots \times \widetilde{C}_m^\star(\boldsymbol{x}_{T+1})$ the prediction set output by the hierarchical component-wise SCP (Algorithm 4) run with $P_{\Sigma^{-1}}$, and denote $\widetilde{C}_1(\boldsymbol{x}_{T+1}) \times \ldots \times \widetilde{C}_m(\boldsymbol{x}_{T+1})$ the prediction set obtained by the same algorithm with another choice of a projection matrix $P$.

We obtain the following efficiency result, that yields the inequalities (5) of Theorem 3 for all positive weight vectors $\boldsymbol{w}$, not just a single fixed one. (The proof is located in Appendix B and consists of direct adaptations of the proof of Theorem 3, together with an application of Lemma 1.)

**Theorem 4.** *Under Assumptions 2, 3, and 4 (i.i.d. scores with elliptical distribution admitting a second-order moment), the hierarchical version of SCP (Algorithm 4) run with $P_{\Sigma^{-1}}$ provides more efficient prediction sets than with any other choice of a projection matrix $P$ onto $\mathrm{Im}(H)$:*

$$\forall i \in [m], \quad \mathbb{E}\left[\mathfrak{L}_1\big(\widetilde{C}_i^\star(\boldsymbol{x}_{T+1})\big)^2\right] \leqslant \mathbb{E}\left[\mathfrak{L}_1\big(\widetilde{C}_i(\boldsymbol{x}_{T+1})\big)^2\right].$$

**Corollary 1.** *Under the setting and assumptions of Theorem 4, more efficient prediction sets are obtained than with the ones $\widehat{C}_i(\boldsymbol{x}_{T+1})$ from the plain component-wise version of SCP (Algorithm 3):*

$$\forall i \in [m], \quad \mathbb{E}\left[\mathfrak{L}_1\big(\widetilde{C}_i^\star(\boldsymbol{x}_{T+1})\big)^2\right] \leqslant \mathbb{E}\left[\mathfrak{L}_1\big(\widehat{C}_i(\boldsymbol{x}_{T+1})\big)^2\right].$$

**Practical implementation.** Theorem 4 motivated the introduction of Algorithm 5. Examples of functions $\mathcal{P}$ used therein are listed below, all of the form $\mathcal{P}\big(\widehat{\boldsymbol{\mu}}, (\boldsymbol{x}_t, \boldsymbol{y}_t)_{t \in \mathcal{D}_{\mathrm{estim}}}\big) = \mathcal{P}'\big(\widehat{\Sigma}\big)$, where $\widehat{\Sigma}$ was defined in (3) and whose Moore-Penrose pseudo-inverse is considered. These functions indeed return projection matrices onto $\mathrm{Im}(H)$, see Lemma 5 in Appendix A:

| Nickname | Algorithms | Parameter | Expression |
|----------|-----------|-----------|------------|
| Direct | Alg. 3 | – | – |
| OLS | Alg. 4 | $P_{\mathbf{1}}$ | $H\big(H^\top H\big)^{-1} H^\top$ |
| WLS | Alg. 5 | $\mathcal{P}'_{\mathrm{WLS}}\big(\widehat{\Sigma}\big)$ | $H\big(H^\top \mathrm{Diag}\big(\widehat{\Sigma}\big)^{-1} H\big)^{-1} H^\top \mathrm{Diag}\big(\widehat{\Sigma}\big)^{-1}$ |
| MinT | Alg. 5 | $\mathcal{P}'_{\mathrm{MinT}}\big(\widehat{\Sigma}\big)$ | $H\big(H^\top \widehat{\Sigma}^{-1} H\big)^{-1} H^\top \widehat{\Sigma}^{-1}$ |
| Combi | Alg. 5 | $\mathcal{P}'_{\mathrm{Combi}}\big(\widehat{\Sigma}\big)$ | $\frac{1}{3}\big(P_{\mathbf{1}} + \mathcal{P}'_{\mathrm{WLS}}\big(\widehat{\Sigma}\big) + \mathcal{P}'_{\mathrm{MinT}}\big(\widehat{\Sigma}\big)\big)$ |

$\mathcal{P}'_{\mathrm{MinT}}$ mimics the expression for $P_{\Sigma^{-1}}$ in Theorem 4, corresponding to the minimum-trace [MinT] projection. When data is scarce, the estimates $\widehat{\Sigma}$ may be poor, and a more robust approach is to consider only the associated diagonal matrices $\mathrm{Diag}\big(\widehat{\Sigma}\big)$; this corresponds to some data-based weighted least squares [WLS], as pointed out by Hyndman et al. (2016). Another alternative is to consider the orthogonal projection onto $\mathrm{Im}(H)$, whose closed-form expression is $P_{\mathbf{1}}$ in (4), with $\mathbf{1} = (1, \ldots, 1)^\top$; it performs an ordinary least squares [OLS] approach, and corresponds to a constant function $\mathcal{P}'_{\mathrm{OLS}}$ when implemented in Algorithm 5. Finally, another robust approach could be to use a combination [Combi] of the $\mathcal{P}$ functions defined above, as suggested by Hollyman et al. (2021).

## 5 Experiments on synthetic data

We provide detailed numerical experiments on synthetic hierarchical i.i.d. data in Appendix F.

*Remark 4.* Real-world hierarchical i.i.d. data would include survey data with hierarchically structured answers (e.g., household budget surveys, where expenditures are decomposed across various categories[2]); however, due to confidentiality issues, we could not get access to a sufficient amount of such data. We thank a reviewer for pointing out an interesting potential application with accessible hierarchical i.i.d. data, namely, image processing through downsampling and upsampling; we let interested readers work it out. That being said, other examples of accessible hierarchical data are formed by some time series, where the assumption of i.i.d. residuals is particularly strong, as it is model-dependent, and therefore do not fall into the scope of the present contribution; additional methodological developments are required to tackle them, see Section 6.

Given our data generation process, we consider base forecasts given by generalized additive models learned independently on each component (details provided in Appendix F.1.4). We compare the performance achieved by the algorithms presented in this article, both in terms of coverage and efficiency. The target coverage is $1 - \alpha = 90\%$ for joint coverage and for all component-wise coverages. As expected, all algorithms do achieve the required coverage level for all configurations tested, which is why we do not detail these coverage results in the main body of this article.

---

[2]As in `https://ec.europa.eu/eurostat/web/microdata/household-budget-survey`

We consider 6 hierarchical configurations (referred to as Configurations 1–6, see Appendix F.1.2 for details), ordered by increasing complexity, with total numbers $m$ of nodes ranging from 16 to 1,801 (and with depths 3 or 4). We consider $T = 10^6$ observations and generate $N = 10^3$ runs for each configuration–algorithm pair. In the tables below, we compute (normalized) empirical averages of the volumes of the ellipsoidal sets output by Algorithms (1) and (2), and root-empirical averages $\sqrt{\overline{L}_\bullet}$ of the uniform total lengths output by Algorithms 3–4–5: by indexing the outcomes of runs by $(r)$,

$$\overline{L}_\bullet = \frac{1}{N} \sum_{r=1}^{N} \sum_{i=1}^{m} \Big( \mathfrak{L}_1\big(\widetilde{C}_i^{(r)}(\cdot)\big) \Big)^2 \, ;$$

the lengths of the intervals $\widetilde{C}_i(\boldsymbol{x})$ do not depend on $\boldsymbol{x}$, hence the notation $\mathfrak{L}_1\big(\widetilde{C}_i^{(r)}(\cdot)\big)$. These root-empirical averages are reported in the tables below with $\pm\sqrt{1.96 \times \text{standard errors}}$. Again, complete details may be found in Appendix F. Note that our generation process involves a broad family of distributions, which explains why the standard errors reported below are relatively large.

**SCP for joint coverage based on ellipsoidal sets.** The table illustrates the second part of Theorem 1 with a proper choice of $A$: Algorithm (2) provides smaller prediction ellipsoids than Algorithm (1). We report here empirical averages of the normalized volumes for the typical matrix $A = \widehat{\Sigma}^{-1}$, but two other choices are considered in Appendix F.

| Matrix $A$ | Config. | Alg. (1) | Alg. (2) |
|:---:|:---:|:---:|:---:|
| $\widehat{\Sigma}^{-1}$ | 1 | $17.4 \pm 0.6$ | $\mathbf{16.5 \pm 0.55}$ |
| | 2 | $3.36 \pm 0.12$ | $\mathbf{3.19 \pm 0.11}$ |

**Component-wise SCP.** The table below illustrates Theorems 3 and 4, with the specific algorithms described in Section 4.3; it reports the root-empirical averages $\sqrt{\overline{L}_\bullet}$ with $\pm\sqrt{1.96 \times \text{standard errors}}$. The best and second best intervals of each line are overlapping and no statistically significant differences in performance between them are exhibited with the number $N = 10^3$ of runs considered.

| Config. | Direct | OLS | WLS | Combi | MinT |
|:---:|:---:|:---:|:---:|:---:|:---:|
| 1 | $876 \pm 254$ | $787 \pm 226$ | $322 \pm 131$ | $364 \pm 101$ | $\mathbf{216 \pm 47}$ |
| 2 | $871 \pm 253$ | $753 \pm 216$ | $308 \pm 116$ | $361 \pm 92$ | $\mathbf{246 \pm 51}$ |
| 3 | $3032 \pm 467$ | $2954 \pm 455$ | $1869 \pm 377$ | $1758 \pm 395$ | $\mathbf{1502 \pm 578}$ |
| 4 | $3036 \pm 479$ | $2901 \pm 458$ | $1581 \pm 340$ | $1604 \pm 349$ | $\mathbf{1404 \pm 571}$ |
| 5 | $10424 \pm 885$ | $10358 \pm 880$ | $\mathbf{8861 \pm 853}$ | $9664 \pm 850$ | $10613 \pm 918$ |
| 6 | $10621 \pm 889$ | $10460 \pm 875$ | $\mathbf{7673 \pm 785}$ | $9068 \pm 806$ | $10503 \pm 905$ |

Three algorithms perform uniformly better than the benchmark algorithm Direct, namely: WLS (with reductions in total lengths in the $15\% - 65\%$ range) and to a smaller extent, Combi and OLS. Algorithm MinT has a dual behavior and is somewhat unreliable: it is the most efficient one for the smallest hierarchies but performs worse than the benchmark Direct for the largest hierarchies. These limitations of MinT, and the robustness of WLS, are further discussed in Appendix F.

# 6 Discussion and future work

In this article, we combined conformal prediction and forecast reconciliation, thereby unifying distribution-free uncertainty quantification with the structural information encoded in hierarchical data.

Our theoretical analysis was conducted under a favorable setting, assuming i.i.d. non-conformity scores drawn from an elliptical distribution. For our strongest results, we further assumed that the covariance matrix of the scores was known. Although simplified, this setting captures key structural properties that are expected to remain relevant in practical applications. We therefore believe that the principles and efficiency gains demonstrated here will translate empirically to applied forecasting contexts.

In practice, many natural applications of the proposed framework arise in hierarchical time series forecasting, where temporal dependence typically violates the i.i.d. assumption on non-conformity scores. One must therefore rely on techniques designed for non-exchangeable data. Extending the present framework to hierarchical time series is a promising and challenging direction for future work. A natural avenue consists in leveraging the adaptive conformal inference [ACI] framework of Gibbs & Candès (2021) and its recent extensions by Zaffran et al. (2022) and Gibbs & Candès (2024), which relax exchangeability by incorporating temporal dynamics while maintaining long-term coverage guarantees. Another interesting direction is to consider dynamic hierarchies, where nodes may be added or removed over time.

### Acknowledgments

This work was supported by Défi Inria–EDF.

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

## Appendices

The appendices provide detailed proofs of all claims made in the main body, as well as some background material.

We organized them so that the most important material, related to the efficiency results for hierarchical component-wise SCP (Theorems 3 and 4), comes first. Standard proofs, like the ones for coverage guarantees, are provided later. Numerical experiments conclude the appendices.

More precisely:

- Appendix A provides a proof of the efficiency result for fixed weights $w$ (Theorem 3), and does so by providing first some background on elliptical distributions as well as a first trace-minimization result.

- Appendix B proves the stronger component-wise efficiency results stated in Theorem 4 and Corollary 1, and does so by first proving a central trace-minimization result known as the minimum-trace projection lemma (Lemma 1 of Section 4.3, whose elementary proof is of independent interest).

- Appendix C provides some background on the theory of forecast reconciliation and discusses the challenges overcome when connecting this literature to conformal prediction.

- Appendix D proves in a straightforward manner the efficiency results for hierarchical SCP for joint coverage (Theorem 1), which illustrates, again, the challenges overcome when obtaining efficiency results for component-wise SCP.

- Appendix E formally states and proves all coverage results (see Theorems 1 and 2); the proofs are extremely standard and are only provided for the sake of completeness.

- Finally, Appendix F provides detailed and extensive numerical experiments on synthetic data.

## A   Proof of the efficiency result for fixed weights $w$ (Theorem 3)

We restate all definitions, facts, etc., as well as Theorem 3, so that this appendix is fully self-contained and may be read without reading back the main body of the article.

### A.1 Background on elliptical distributions

We first recall the definition of elliptical distributions and then state some elementary properties thereof.

**Definition 2.** A random vector $\boldsymbol{z}$ follows a spherical distribution over $\mathbb{R}^k$ if $\boldsymbol{z}$ and $\Gamma\boldsymbol{z}$ have the same distribution for all $k \times k$ orthogonal matrices $\Gamma$.

**Definition 3.** An elliptical distribution over $\mathbb{R}^m$ is of the form $\boldsymbol{c} + M\boldsymbol{z}$, for a deterministic vector $\boldsymbol{c} \in \mathbb{R}^m$, a $m \times k$ matrix $M$ such that $MM^\top$ has rank $k$, and a random vector $\boldsymbol{z}$ following a spherical distribution over $\mathbb{R}^k$.

**Examples.** The simplest example of elliptical distributions consists of multivariate normal distributions (which are light-tailed distributions). Other examples include multivariate $t$–distributions and symmetric multivariate Laplace distributions (both heavy tailed) and the uniform distribution on an ellipse (no tail).

**Property 1.** The marginals of a spherical distribution are identically distributed. A spherical distribution with a first-order moment is centered: $\mathbb{E}[\boldsymbol{z}] = \boldsymbol{0}$. A spherical distribution with a second-order moment has a covariance matrix proportional to the identity: there exists $\sigma^2 \in [0, +\infty)$ such that $\mathbb{E}[\boldsymbol{z}\boldsymbol{z}^\top] = \sigma^2\,\mathrm{Id}_k$.

*Proof.* The first property is proved by considering permutation matrices $\Gamma$. The second property holds because $\boldsymbol{u} = \boldsymbol{0}$ is the only vector $\boldsymbol{u} \in \mathbb{R}^k$ such that $\Gamma\boldsymbol{u} = \boldsymbol{u}$ for all orthogonal matrices (first consider permutation matrices to get that all components of $\boldsymbol{u}$ are equal). For the third property, denote by $\Sigma$ the covariance matrix of $\boldsymbol{z}$. Since it is symmetric (positive semi-definite), there exists an orthogonal matrix $\Gamma$ and a vector $\boldsymbol{\lambda} \in \mathbb{R}^k$ (with non-negative elements) such that $\Gamma\Sigma\Gamma = \mathrm{diag}(\boldsymbol{\lambda})$. Now, $\Gamma^\top\boldsymbol{z}$ has the same distribution as $\boldsymbol{z}$, thus their covariance matrices are equal, which shows that $\Sigma = \mathrm{diag}(\boldsymbol{\lambda})$. As marginals have the same distribution, we finally get $\Sigma = \sigma^2\,\mathrm{Id}_k$ for some $\sigma^2 \in [0, +\infty)$, which is actually positive except if the distribution of $\boldsymbol{z}$ is a Dirac at $\boldsymbol{0}$. $\qquad\square$

A slightly more advanced result provides the form of the characteristic function of an elliptical distribution. Its proof is based on first showing that characteristic functions of spherical distributions are exactly of the form $\boldsymbol{u} \mapsto \phi(\boldsymbol{u}^\top\boldsymbol{u})$, which is consistent with the fact that spherical distributions are centered. Actually, it may be seen that $\phi$ is the characteristic function of the common distribution of the marginals of $\boldsymbol{z}$.

**Lemma 2** (Kollo & von Rosen, 2005, Theorem 2.3.5)**.** *Consider a random variable following an elliptical distribution over $\mathbb{R}^m$, of the form $\boldsymbol{c} + M\boldsymbol{z}$, for a deterministic vector $\boldsymbol{c} \in \mathbb{R}^m$, a $m \times k$ matrix $M$ such that $MM^\top$ has rank $k$, and a random vector $\boldsymbol{z}$ following a spherical distribution over $\mathbb{R}^k$. The characteristic function of $\boldsymbol{c} + M\boldsymbol{z}$ is of the form*

$$\forall\boldsymbol{u} \in \mathbb{R}^m, \qquad \mathbb{E}\Big[\exp\big(\mathrm{i}\boldsymbol{u}^\top(\boldsymbol{c} + M\boldsymbol{z})\big)\Big] = \exp\big(\mathrm{i}\boldsymbol{u}^\top\boldsymbol{c}\big)\ \phi\big(\boldsymbol{u}^\top MM^\top\boldsymbol{u}\big)\,,$$

*for some function $\phi : \mathbb{R} \to \mathbb{C}$ that only depends on the distribution of $\boldsymbol{z}$.*

Lemma 2 is instrumental in showing that the normalized marginals of (a linear transformation of) an elliptical distribution have comparable univariate distributions (that are homothetical), as stated next.

**Lemma 3.** *With the setting and the notation of Lemma 2, let $N$ be any $m \times m$ matrix and consider the random vector $\boldsymbol{s} = N(\boldsymbol{c} + M\boldsymbol{z})$. Let $\Lambda = NMM^\top N^\top$. There exists a random variable $v$, following a univariate distribution induced by the spherical distribution of $\boldsymbol{z}$, such that*

$$\forall i \in [m], \qquad s_i - \mathbb{E}[s_i] \stackrel{(\mathrm{d})}{=} \sqrt{\Lambda_{i,i}}\ v\,.$$

*Proof.* By Lemma 2, the characteristic function of $\boldsymbol{s} - \mathbb{E}[\boldsymbol{s}]$ is $\boldsymbol{u} \in \mathbb{R}^m \mapsto \phi(\boldsymbol{u}^\top\Lambda\boldsymbol{u})$. Thus, the characteristic function of each $s_i - \mathbb{E}[s_i]$ is $u \in \mathbb{R} \mapsto \phi(\Lambda_{i,i}u^2)$. This shows the stated result, for a random variable $v$ with characteristic function $\phi$. $\qquad\square$

### A.2 Proof of Theorem 3

We first prove that the matrix $P_{\boldsymbol{w}}$ introduced in Theorem 3 (re-stated below) is well defined.

**Lemma 4.** *The matrices $H^{\top}H$ and $H^{\top}WH$ are $n \times n$ symmetric positive definite matrices, where $W$ is itself a $m \times m$ symmetric positive definite matrix. Thus, these matrices are invertible.*

*Proof.* As indicated in Section 2, the structural matrix $H$ is of the form

$$H = \begin{bmatrix} \mathrm{Id}_n \\ H_{\mathrm{sub}} \end{bmatrix} \tag{8}$$

where $H_{\mathrm{sub}}$ is any $(m-n) \times n$ matrix of real numbers. This entails that $H^{\top}H = \mathrm{Id}_n + H_{\mathrm{sub}}^{\top}H_{\mathrm{sub}}$, where $H_{\mathrm{sub}}^{\top}H_{\mathrm{sub}}$ is symmetric positive semi-definite. Thus, $H^{\top}H$ is symmetric positive definite. Given it is symmetric positive definite, the matrix $W$ may be decomposed as $W = N^{\top}N$ for some $m \times m$ invertible matrix $N$. The matrix $H^{\top}WH = (NH)^{\top}NH$ is symmetric positive semi-definite. We show that it is even symmetric positive definite: $\boldsymbol{u}^{\top}(NH)^{\top}NH\boldsymbol{u} = 0$ is equivalent to the standard Euclidean norm of $NH\boldsymbol{u}$ being null, thus to $H\boldsymbol{u} = \boldsymbol{0}$ (as $N$ is invertible); given the form (8) of $H$, we conclude that $\boldsymbol{u}^{\top}(NH)^{\top}NH\boldsymbol{u} = 0$ is equivalent to $\boldsymbol{u} = \boldsymbol{0}$, which is the definition of $H^{\top}WH = (NH)^{\top}NH$ being definite. □

We are now ready to actually prove Theorem 3, which we restate first, together with its key assumption. Assumption 2 is that the residuals considered in Assumption 3 are i.i.d.

**Assumption 3.** The (i.i.d.) residuals $\widehat{\boldsymbol{s}}_t = \boldsymbol{y}_t - \widehat{\boldsymbol{\mu}}(\boldsymbol{x}_t)$, for $t \in \mathcal{D}_{\mathrm{calib}}$, follow some elliptical distribution (whose shape and parameters are unknown).

**Theorem 3.** *Fix $\boldsymbol{w} \in (0, +\infty)^m$. Under Assumptions 2 and 3 (i.i.d. scores with elliptical distribution), the hierarchical component-wise SCP (Algorithm 4) run with $P = P_{\boldsymbol{w}}$, where*

$$P_{\boldsymbol{w}} \stackrel{\mathrm{def}}{=} H\big(H^{\top}\mathrm{diag}(\boldsymbol{w})H\big)^{-1}H^{\top}\mathrm{diag}(\boldsymbol{w}), \tag{4}$$

*provides prediction sets that are more efficient than the ones output by the plain component-wise SCP (Algorithm 3) in the following sense:*

$$\mathbb{E}\left[\sum_{i=1}^{m} w_i \, \mathfrak{L}_1\big(\widetilde{C}_i(\boldsymbol{x}_{T+1})\big)^2\right] \leqslant \mathbb{E}\left[\sum_{i=1}^{m} w_i \, \mathfrak{L}_1\big(\widehat{C}_i(\boldsymbol{x}_{T+1})\big)^2\right]. \tag{5}$$

*Proof.* The matrix $P_{\boldsymbol{w}}$ satisfies $P_{\boldsymbol{w}}H = H$, i.e., $P_{\boldsymbol{w}}$ leaves elements of $\mathrm{Im}(H)$ unchanged. Since observations $\boldsymbol{y}_t$ are coherent, we have, for all $t \in \mathcal{D}_{\mathrm{calib}}$,

$$\widetilde{\boldsymbol{s}}_t \stackrel{\mathrm{def}}{=} \boldsymbol{y}_t - P_{\boldsymbol{w}}\widehat{\boldsymbol{y}}_t = P_{\boldsymbol{w}}\big(\boldsymbol{y}_t - \widehat{\boldsymbol{y}}_t\big) = P_{\boldsymbol{w}}\,\widehat{\boldsymbol{s}}_t\,.$$

We let, for all $t \in \mathcal{D}_{\mathrm{calib}}$ and $i \in [m]$,

$$\widehat{\xi}_{t,i} = \widehat{s}_{t,i} - \mathbb{E}\big[\widehat{s}_{1,i}\big] \qquad \text{and} \qquad \widetilde{\xi}_{t,i} = \widetilde{s}_{t,i} - \mathbb{E}\big[\widetilde{s}_{1,i}\big]\,.$$

By Assumption 2 (i.i.d. scores), for each $i \in [m]$, the univariate random variables $\widehat{\xi}_{t,i}$, where $t \in \mathcal{D}_{\mathrm{calib}}$ are i.i.d.; a similar statement holds for the $\widetilde{\xi}_{t,i}$, where $t \in \mathcal{D}_{\mathrm{calib}}$. By Assumption 3 and Lemma 3, there exist a matrix $\Gamma$ of the form $\Gamma = MM^{\top}$ and a random variable $v$ such that, for each $i \in [m]$,

$$\widehat{\xi}_{t,i} \stackrel{(\mathrm{d})}{=} \sqrt{\Gamma_{i,i}}\, v \qquad \text{and} \qquad \widetilde{\xi}_{t,i} \stackrel{(\mathrm{d})}{=} \sqrt{\Gamma'_{i,i}}\, v\,, \qquad \text{where} \qquad \Gamma' = P_{\boldsymbol{w}}\Gamma P_{\boldsymbol{w}}^{\top}\,.$$

Let $(v_t)_{t \in \mathcal{D}_{\mathrm{calib}}}$ be i.i.d. random variables with the same distribution as $v$. We conclude from the facts above that for each $i \in [m]$,

$$\big(\widehat{s}_{t,i}\big)_{t \in \mathcal{D}_{\mathrm{calib}}} \stackrel{(\mathrm{d})}{=} \left(\mathbb{E}\big[\widehat{s}_{1,i}\big] + \sqrt{\Gamma_{i,i}}\, v_t\right)_{t \in \mathcal{D}_{\mathrm{calib}}} \qquad \text{and} \qquad \big(\widetilde{s}_{t,i}\big)_{t \in \mathcal{D}_{\mathrm{calib}}} \stackrel{(\mathrm{d})}{=} \left(\mathbb{E}\big[\widetilde{s}_{1,i}\big] + \sqrt{\Gamma'_{i,i}}\, v_t\right)_{t \in \mathcal{D}_{\mathrm{calib}}}.$$

The same equalities in distributions hold for the corresponding order statistics: for each $i \in [m]$,

$$\left(\widehat{s}_{(t),i}\right)_{1 \leqslant t \leqslant T_{\text{calib}}} \stackrel{\text{(d)}}{=} \left(\mathbb{E}\big[\widehat{s}_{1,i}\big] + \sqrt{\Gamma_{i,i}}\ v_{(t)}\right)_{1 \leqslant t \leqslant T_{\text{calib}}}$$

$$\text{and} \qquad \left(\widetilde{s}_{(t),i}\right)_{1 \leqslant t \leqslant T_{\text{calib}}} \stackrel{\text{(d)}}{=} \left(\mathbb{E}\big[\widetilde{s}_{1,i}\big] + \sqrt{\Gamma'_{i,i}}\ v_{(t)}\right)_{1 \leqslant t \leqslant T_{\text{calib}}}.$$

By following the conventions of Section 2.3 and letting $v_{(0)} = -\infty$ and $v_{(T_{\text{calib}}+1)} = +\infty$, we even have these equalities in distribution over vectors indexed by $0 \leqslant t \leqslant T_{\text{calib}} + 1$: for each $i \in [m]$,

$$\left(\widehat{s}_{(t),i}\right)_{0 \leqslant t \leqslant T_{\text{calib}}+1} \stackrel{\text{(d)}}{=} \left(\mathbb{E}\big[\widehat{s}_{1,i}\big] + \sqrt{\Gamma_{i,i}}\ v_{(t)}\right)_{0 \leqslant t \leqslant T_{\text{calib}}+1}$$

$$\text{and} \qquad \left(\widetilde{s}_{(t),i}\right)_{0 \leqslant t \leqslant T_{\text{calib}}+1} \stackrel{\text{(d)}}{=} \left(\mathbb{E}\big[\widetilde{s}_{1,i}\big] + \sqrt{\Gamma'_{i,i}}\ v_{(t)}\right)_{0 \leqslant t \leqslant T_{\text{calib}}+1}.$$

Now, for each $i \in [m]$, by design of Algorithms 3 and 4, the lengths of the intervals $\widehat{C}_i(\boldsymbol{x}_{T+1})$ and $\widetilde{C}_i(\boldsymbol{x}_{T+1})$ output are

$$\mathfrak{L}_1\big(\widehat{C}_i(\boldsymbol{x}_{T+1})\big) = \widehat{s}_{\left(\lceil (T_{\text{calib}}+1)(1-\alpha/2)\rceil\right),i} - \widehat{s}_{\left(\lfloor (T_{\text{calib}}+1)\alpha/2\rfloor\right),i}$$

$$\text{and} \qquad \mathfrak{L}_1\big(\widetilde{C}_i(\boldsymbol{x}_{T+1})\big) = \widetilde{s}_{\left(\lceil (T_{\text{calib}}+1)(1-\alpha/2)\rceil\right),i} - \widetilde{s}_{\left(\lfloor (T_{\text{calib}}+1)\alpha/2\rfloor\right),i}.$$

Thus, letting $L_\alpha = v_{\left(\lceil (T_{\text{calib}}+1)(1-\alpha/2)\rceil\right)} - v_{\left(\lfloor (T_{\text{calib}}+1)\alpha/2\rfloor\right)}$, where $L_\alpha \geqslant 0$ a.s., we finally proved

$$\forall i \in [m], \qquad \mathfrak{L}_1\big(\widehat{C}_i(\boldsymbol{x}_{T+1})\big) \stackrel{\text{(d)}}{=} \sqrt{\Gamma_{i,i}}\ L_\alpha \qquad \text{and} \qquad \mathfrak{L}_1\big(\widetilde{C}_i(\boldsymbol{x}_{T+1})\big) \stackrel{\text{(d)}}{=} \sqrt{\Gamma'_{i,i}}\ L_\alpha.$$

We showed so far that

$$\mathbb{E}\left[\sum_{i=1}^m w_i\ \mathfrak{L}_1\big(\widehat{C}_i(\boldsymbol{x}_{T+1})\big)^2\right] = \left(\sum_{i=1}^m w_i\ \Gamma_{i,i}\right) \mathbb{E}\big[L_\alpha^2\big]$$

$$\text{and} \qquad \mathbb{E}\left[\sum_{i=1}^m w_i\ \mathfrak{L}_1\big(\widetilde{C}_i(\boldsymbol{x}_{T+1})\big)^2\right] = \left(\sum_{i=1}^m w_i\ \Gamma'_{i,i}\right) \mathbb{E}\big[L_\alpha^2\big],$$

where $\mathbb{E}\big[L_\alpha^2\big] \geqslant 0$ is possibly infinite (in which case the stated result holds). The proof is concluded in the case $\mathbb{E}\big[L_\alpha^2\big] < +\infty$ by noting that

$$\text{Tr}\big(\text{diag}(\boldsymbol{w})\,\Gamma\big) = \sum_{i=1}^m w_i\,\Gamma_{i,i} \geqslant \sum_{i=1}^m w_i\,\Gamma'_{i,i} = \text{Tr}\big(\text{diag}(\boldsymbol{w})\,\Gamma'\big) = \text{Tr}\big(\text{diag}(\boldsymbol{w})\,P_{\boldsymbol{w}}\Gamma P_{\boldsymbol{w}}^\top\big),$$

which is guaranteed by the lemma below with $W = \text{diag}(\boldsymbol{w})$, since $\Gamma = MM^\top$ for some $m \times k$ matrix. $\qquad\square$

The first part of Lemma 5 is elementary. Its second part is inspired by Panagiotelis et al. (2021, Theorem 3.2), which is a result about using orthogonal projections in the $\|\cdot\|_W$–norm to derive distance-reducing properties, and by trace-minimization results that are classic in the literature of forecast reconciliation (like Lemma 1 of Section 4.3). We however see this second part as a new result of our own. See Appendix C.2, and in particular, the comments after (11), for more details on the challenges overcome when connecting the theory of forecast reconciliation to the one of conformal learning.

**Lemma 5.** *Fix a symmetric positive definite matrix $W$ and consider the associated inner product and induced norm*

$$\boldsymbol{u}, \boldsymbol{u}' \in \mathbb{R}^m \longmapsto \langle \boldsymbol{u},\,\boldsymbol{u}'\rangle_W = \sqrt{\boldsymbol{u}^\top W \boldsymbol{u}'} \qquad \text{and} \qquad \boldsymbol{u} \in \mathbb{R}^m \longmapsto \|\boldsymbol{u}\|_W \stackrel{\text{def}}{=} \sqrt{\boldsymbol{u}^\top W \boldsymbol{u}}.$$

*Then, $P_W \stackrel{\text{def}}{=} H\big(H^\top W H\big)^{-1} H^\top W$ is the orthogonal projection onto $\text{Im}(H)$ in the $\|\cdot\|_W$–norm.*

*Furthermore, for all $m \times k$ matrices $M$,*

$$0 \leqslant \text{Tr}\big(W P_W M M^\top P_W^\top\big) \leqslant \text{Tr}\big(W M M^\top\big).$$

*Proof.* First, $P_W$ is indeed a projection onto $\mathrm{Im}(H)$: namely, $P_W P_W = P_W$ and $P_W H = H$. To show that $P_W$ is an orthogonal projection for the $\|\cdot\|_W$–norm, it suffices to note that for all $\boldsymbol{u}, \boldsymbol{u}' \in \mathbb{R}^m$,

$$\langle P_W \boldsymbol{u}, \, \boldsymbol{u}' \rangle_W \overset{\mathrm{def}}{=} (P_W \boldsymbol{u})^\top W \boldsymbol{u}' = \boldsymbol{u}^\top W P_W \boldsymbol{u}' \overset{\mathrm{def}}{=} \langle \boldsymbol{u}, \, P_W \boldsymbol{u}' \rangle_W \, ,$$

where we used that $P_W^\top W = W P_W$, given the closed-form expression of $P_W$.

Now, let $\boldsymbol{z}'$ be a standard Gaussian random $k$–vector: $\boldsymbol{z}' \sim \mathcal{N}(\boldsymbol{0}, \mathrm{Id}_k)$. On the one hand, given the orthogonality proved for $P_W$ and by a Pythagorean theorem,

$$\|P_W M \boldsymbol{z}'\|_W^2 \leqslant \|M \boldsymbol{z}'\|_W^2 \quad \text{a.s.} \tag{9}$$

Now, by definition of the $\|\cdot\|_W$–norm and by elementary properties of the trace,

$$\begin{aligned}
\mathbb{E}\big[\|P_W M \boldsymbol{z}'\|_W^2\big] &= \mathbb{E}\big[(P_W M \boldsymbol{z}')^\top W P_W M \boldsymbol{z}'\big] \\
&= \mathbb{E}\Big[\mathrm{Tr}\big(W P_W M \boldsymbol{z}' (P_W M \boldsymbol{z}')^\top\big)\Big] = \mathrm{Tr}\Big(W P_W M \overbrace{\mathbb{E}\big[\boldsymbol{z}'(\boldsymbol{z}')^\top\big]}^{=\mathrm{Id}_k} M^\top P_W^\top\Big) \\
&= \mathrm{Tr}\Big(W P_W M M^\top P_W^\top\Big) \, .
\end{aligned}$$

Similarly, $\quad \mathbb{E}\big[\|M \boldsymbol{z}'\|_W^2\big] = \mathrm{Tr}\big(W M M^\top\big).$

The inequality (9) and the two equalities proved above conclude the proof. $\qquad\square$

# B    Proof of the component-wise efficiency results (Theorem 4 and Corollary 1)

In this section (as in Appendix A), we restate all the material needed so that this appendix is fully self-contained and may be read without reading back the main body of the article. The proof of Theorem 4 is based on a key equality established in the proof of Theorem 3 and on a minimum-trace-projection result that is central in the theory of forecast reconciliation. We start with the latter.

## B.1    Minimum-trace projection

The following lemma is a deep and central result in the theory of forecast reconciliation. First stated for $W = \mathrm{Id}_m$ by Wickramasuriya et al. (2019), this result has since been extended to symmetric positive semi-definite matrices $W$ in Panagiotelis et al. (2021) and Ando & Narita (2024). We provide a short and elementary proof, which may actually be seen as a simplification of the proof by Ando & Narita (2024), an article entirely devoted to proving Lemma 1. The latter article sees the minimization problem at hand as a constrained minimization problem (given how projections onto $\mathrm{Im}(H)$ may be written), thus introduced a Lagrangian and discussed Karush-Kuhn-Tucker conditions to solve it.

**Lemma 1** (Minimum-trace projection, Wickramasuriya et al., 2019)**.** *Let $W$ and $\Sigma$ be two symmetric $m \times m$ matrices, where $W$ is positive semi-definite and $\Sigma$ is positive definite. Then, for all projection matrices $P$ onto $\mathrm{Im}(H)$,*

$$\mathrm{Tr}\big(W P_{\Sigma^{-1}} \Sigma P_{\Sigma^{-1}}^\top\big) \leqslant \mathrm{Tr}\big(W P \Sigma P^\top\big), \qquad \text{where} \quad P_{\Sigma^{-1}} \overset{\mathrm{def}}{=} H\big(H^\top \Sigma^{-1} H\big)^{-1} H^\top \Sigma^{-1} \, . \tag{7}$$

*Proof.* We first show that projection matrices $P$ onto $\mathrm{Im}(H)$ are exactly the matrices of the form $HG$, where $G$ is a $n \times m$ matrix such that $GH = \mathrm{Id}_n$. Indeed, such a matrix $HG$ satisfies $HG\,HG = HG$ and $HG\,H = H$, which characterizes projections onto $\mathrm{Im}(H)$. Conversely, fix a projection $P$ onto $\mathrm{Im}(H)$ and a basis $\boldsymbol{u}_1, \ldots, \boldsymbol{u}_m$ of $\mathbb{R}^m$: each $P\boldsymbol{u}_i$ belongs to $\mathrm{Im}(H)$, thus is of the form $H\boldsymbol{g}_i$ for some $\boldsymbol{g}_i \in \mathbb{R}^n$. Denote by $G$ the $n \times m$ matrix with columns given by $\boldsymbol{g}_1, \ldots, \boldsymbol{g}_m$. By linearity of $P$ and given that $\boldsymbol{u}_1, \ldots, \boldsymbol{u}_m$ is a basis, we have $P = HG$. We denote by $\boldsymbol{h}_1, \ldots, \boldsymbol{h}_n$ the columns of the $m \times n$ structural matrix $H$. Since $P$ is a projection onto $\mathrm{Im}(H)$, we have in particular $P\boldsymbol{h}_i = \boldsymbol{h}_i$ for all $i \in [n]$, or put differently, $PH = H$. Substituting $P = HG$ and multiplying both sides by $H^\top$, we proved so far that $H^\top HGH = H^\top H$, where (see Lemma 4 in Appendix A.2), the matrix $H^\top H$ is invertible. All in all, we thus proved $GH = \mathrm{Id}_n$.

Given the above characterization, the projection matrices $P$ onto $\mathrm{Im}(H)$ are also exactly the matrices of the form

$$P = P_{\Sigma^{-1}} + HA = H\left(\left(H^\top\Sigma^{-1}H\right)^{-1}H^\top\Sigma^{-1} + A\right), \quad \text{for } n \times m \text{ matrices } A \text{ such that} \quad AH = [0]_n\,,$$

where $[0]_n$ denotes the $n \times n$ null matrix. Keeping in mind that $\Sigma$ and $\Sigma^{-1}$ are symmetric, this decomposition entails that

$$
\begin{aligned}
WP\Sigma P^\top = \ & W\left(H\left(H^\top\Sigma^{-1}H\right)^{-1}H^\top\Sigma^{-1}\right)\Sigma\left(\Sigma^{-1}H\left(H^\top H\right)^{-1}H^\top\right) \\
& + W\left(HA\right)\Sigma\left(\Sigma^{-1}H\left(H^\top\Sigma^{-1}H\right)^{-1}H^\top\right) \\
& + W\left(H\left(H^\top\Sigma^{-1}H\right)^{-1}H^\top\Sigma^{-1}\right)\Sigma\left(A^\top H^\top\right) \\
& + W\left(HA\right)\Sigma\left(HA\right)^\top.
\end{aligned}
$$

The second term in the decomposition simplifies into

$$W\left(HA\right)\Sigma\left(\Sigma^{-1}H\left(H^\top\Sigma^{-1}H\right)^{-1}H^\top\right) = WH\,\overbrace{AH}^{=[0]_n}\left(H^\top\Sigma^{-1}H\right)^{-1}H^\top = [0]_m\,.$$

Similarly, the third term is also null, due to the term $H^\top\Sigma^{-1}\Sigma A^\top = (AH)^\top$. The proof is concluded by noting that for all matrices $A$, the trace of the fourth term in the decomposition of $WP\Sigma P^\top$ is non-negative. Indeed, given that $W$ and $\Sigma$ are positive semi-definite, we may write them as $W = MM^\top$ and $\Sigma = NN^\top$ for $m \times m$ matrices $M$ and $N$. Then, together with elementary properties of the trace,

$$
\begin{aligned}
\mathrm{Tr}\left(W\left(HA\right)\Sigma\left(HA\right)^\top\right) &= \mathrm{Tr}\left(MM^\top(HA)NN^\top(HA)^\top\right) \\
&= \mathrm{Tr}\left(M^\top(HA)NN^\top(HA)^\top M\right) \\
&= \mathrm{Tr}\left(\left(M^\top(HA)N\right)\left(M^\top(HA)N\right)^\top\right) \geqslant 0\,,
\end{aligned}
$$

given that the trace of a symmetric positive semi-definite matrix is non-negative. □

## B.2  Proofs of Theorem 4 and Corollary 1

We recall that we denoted by

$$\widetilde{C}_1^\star(\boldsymbol{x}_{T+1}) \times \ldots \times \widetilde{C}_m^\star(\boldsymbol{x}_{T+1}) \qquad \text{and} \qquad \widetilde{C}_1(\boldsymbol{x}_{T+1}) \times \ldots \times \widetilde{C}_m(\boldsymbol{x}_{T+1})$$

the prediction rectangles output by the hierarchical component-wise SCP (Algorithm 4) run with $P_{\Sigma^{-1}} = H\left(H^\top\Sigma^{-1}H\right)^{-1}H^\top\Sigma^{-1}$ and any other choice of a projection matrix $P$ onto $\mathrm{Im}(H)$, respectively.

**Assumption 4.** The (i.i.d.) residuals $\widehat{\boldsymbol{s}}_t = \boldsymbol{y}_t - \widehat{\boldsymbol{\mu}}(\boldsymbol{x}_t)$, for $t \in \mathcal{D}_{\mathrm{calib}}$, have a bounded second-order moment, with positive definite covariance matrix $\Sigma$.

**Theorem 4.** *Under Assumptions 2, 3, and 4 (i.i.d. scores with elliptical distribution admitting a second-order moment), the hierarchical version of SCP (Algorithm 4) run with $P_{\Sigma^{-1}}$ provides more efficient prediction sets than with any other choice of a projection matrix $P$ onto $\mathrm{Im}(H)$:*

$$\forall i \in [m], \quad \mathbb{E}\left[\mathfrak{L}_1\left(\widetilde{C}_i^\star(\boldsymbol{x}_{T+1})\right)^2\right] \leqslant \mathbb{E}\left[\mathfrak{L}_1\left(\widetilde{C}_i(\boldsymbol{x}_{T+1})\right)^2\right].$$

*Proof.* The proof of Theorem 3 did not rely on the existence of a second-order moment, i.e., of a covariance matrix $\Sigma$ for the distribution of the scores $\widehat{\boldsymbol{s}}_t$. (It did not even rely on the existence of a first-order moment.)

When such a second-order moment exists, we may modify the proof of Theorem 3 in the following way, to obtain expected lengths depending on $\Sigma$. We also note that though we wrote the beginning of that proof for a

specific projection matrix $P_{\boldsymbol{w}}$ onto $\mathrm{Im}(H)$, it holds for all projection matrices $P$ onto $\mathrm{Im}(H)$, and even for all matrices $P$ such that $PH = H$. Namely, when Algorithm 4 is run with any projection matrix $P$ onto $\mathrm{Im}(H)$,

$$
\mathbb{E}\left[\sum_{i=1}^m w_i\ \mathfrak{L}_1\big(\widetilde{C}_i(\boldsymbol{x}_{T+1})\big)^2\right] = \left(\sum_{i=1}^m w_i\,\Gamma'_{i,i}\right)\mathbb{E}\big[L_\alpha^2\big] = \mathrm{Tr}\big(\mathrm{diag}(\boldsymbol{w})\,P\Gamma P^\top\big)\,\mathbb{E}\big[L_\alpha^2\big]\,,
$$

where $\Gamma = MM^\top$ for some matrix $M$ such that scores $\widehat{\boldsymbol{s}}_t$ have the same distribution as some $\boldsymbol{c} + M\boldsymbol{z}$ with $\boldsymbol{z}$ following some spherical distribution. In particular, Assumption 4 and Property 1 impose that $M$ is a $m \times m$ matrix and they entail that there exists $\sigma^2 > 0$ such that $\Sigma = \sigma^2 MM^\top = \sigma^2\Gamma$.

Therefore, we actually have, when Algorithm 4 is run with any projection matrix $P$ onto $\mathrm{Im}(H)$,

$$
\mathbb{E}\left[\sum_{i=1}^m w_i\ \mathfrak{L}_1\big(\widetilde{C}_i(\boldsymbol{x}_{T+1})\big)^2\right] = \left(\sum_{i=1}^m w_i\,\Gamma'_{i,i}\right)\mathbb{E}\big[L_\alpha^2\big] = \mathrm{Tr}\big(\mathrm{diag}(\boldsymbol{w})\,P\Sigma P^\top\big)\,\frac{\mathbb{E}\big[L_\alpha^2\big]}{\sigma^2}\,.
$$

Lemma 5 shows that $P_{\Sigma^{-1}}$ is a projection matrix onto $\mathrm{Im}(H)$. Therefore, Lemma 1 shows that for all projections $P$ onto $\mathrm{Im}(H)$ and all positive vectors $\boldsymbol{w} \in \mathbb{R}^m$,

$$
\mathrm{Tr}\big(\mathrm{diag}(\boldsymbol{w})\,P_{\Sigma^{-1}}\Sigma P_{\Sigma^{-1}}^\top\big) \leqslant \mathrm{Tr}\big(\mathrm{diag}(\boldsymbol{w})\,P\Sigma P^\top\big)\,.
$$

Collecting all elements, whether $\mathbb{E}\big[L_\alpha^2\big] = +\infty$ or $\mathbb{E}\big[L_\alpha^2\big] \in [0, +\infty)$, we proved so far that when Algorithm 4 is run with any projection matrix $P$ onto $\mathrm{Im}(H)$ to output prediction intervals $\widetilde{C}_i$,

$$
\forall \boldsymbol{w} \in (0, +\infty)^m, \qquad \mathbb{E}\left[\sum_{i=1}^m w_i\ \mathfrak{L}_1\big(\widetilde{C}_i^\star(\boldsymbol{x}_{T+1})\big)^2\right] \leqslant \mathbb{E}\left[\sum_{i=1}^m w_i\ \mathfrak{L}_1\big(\widetilde{C}_i(\boldsymbol{x}_{T+1})\big)^2\right]. \tag{10}
$$

We obtain the claimed component-wise inequalities by taking $w_i = 1$ for one component $i$ and letting $w_j \to 0$ for $j \neq i$. $\qquad\square$

We now move on to the proof of Corollary 1.

**Corollary 1.** *Under the setting and assumptions of Theorem 4, more efficient prediction sets are obtained than with the ones $\widehat{C}_i(\boldsymbol{x}_{T+1})$ from the plain component-wise version of SCP (Algorithm 3):*

$$
\forall i \in [m], \quad \mathbb{E}\Big[\mathfrak{L}_1\big(\widetilde{C}_i^\star(\boldsymbol{x}_{T+1})\big)^2\Big] \leqslant \mathbb{E}\Big[\mathfrak{L}_1\big(\widehat{C}_i(\boldsymbol{x}_{T+1})\big)^2\Big].
$$

*Proof.* The result follows from Theorems 3 and 4 (which both hold under the stronger set of assumptions of Theorem 4). More precisely, for each $\boldsymbol{w} \in (0, +\infty)^m$, denote by $\widetilde{C}_i^{\boldsymbol{w}}$ the prediction intervals output by Algorithm 4 run with $P = P_{\boldsymbol{w}}$. Theorem 3 ensures that

$$
\forall \boldsymbol{w} \in (0, +\infty)^m, \qquad \mathbb{E}\left[\sum_{i=1}^m w_i\ \mathfrak{L}_1\big(\widetilde{C}_i^{\boldsymbol{w}}(\boldsymbol{x}_{T+1})\big)^2\right] \leqslant \mathbb{E}\left[\sum_{i=1}^m w_i\ \mathfrak{L}_1\big(\widehat{C}_i(\boldsymbol{x}_{T+1})\big)^2\right].
$$

Equality (10) in the proof of Theorem 4 states that

$$
\forall \boldsymbol{w} \in (0, +\infty)^m, \qquad \mathbb{E}\left[\sum_{i=1}^m w_i\ \mathfrak{L}_1\big(\widetilde{C}_i^\star(\boldsymbol{x}_{T+1})\big)^2\right] \leqslant \mathbb{E}\left[\sum_{i=1}^m w_i\ \mathfrak{L}_1\big(\widetilde{C}_i^{\boldsymbol{w}}(\boldsymbol{x}_{T+1})\big)^2\right].
$$

Combining these two inequalities, we have

$$
\forall \boldsymbol{w} \in (0, +\infty)^m, \qquad \mathbb{E}\left[\sum_{i=1}^m w_i\ \mathfrak{L}_1\big(\widetilde{C}_i^\star(\boldsymbol{x}_{T+1})\big)^2\right] \leqslant \mathbb{E}\left[\sum_{i=1}^m w_i\ \mathfrak{L}_1\big(\widehat{C}_i(\boldsymbol{x}_{T+1})\big)^2\right],
$$

and we conclude the proof with the same limit arguments as after (10) in the proof of Theorem 4. $\qquad\square$

## C  Forecast reconciliation: review, connections made, challenges overcome

The aim of this appendix is to provide some background on the theory of forecast reconciliation and to explain how we connected it to conformal prediction in the proofs of Appendices A and B.

### C.1  Brief overview of the literature on forecast reconciliation

For a complete review on the forecast reconciliation literature, we refer the reader to Athanasopoulos et al. (2024) and only provide a brief overview below.

At first, forecasts in the hierarchical setting were conducted using a single-level approach (most notably, in the bottom-up or top-down fashion), i.e., by choosing a level of the hierarchy (typically, either the bottom level or the top level) to generate forecasts, and then, by propagating these forecasts (typically in a linear fashion). A notable pitfall of the single-level approaches is that potentially valuable information from all other levels are ignored. To overcome this issue, the concept of forecast reconciliation was introduced by Athanasopoulos et al. (2009) and Hyndman et al. (2011): the idea is to combine forecasts from different levels of aggregation through linear combinations. Recently, developments were made in reconciliation through projections (Wickramasuriya et al., 2019, Panagiotelis et al., 2021), which we review and detail in the next section.

Probabilistic hierarchical forecasting and reconciliation is an emerging field. Notable works include the one by Wickramasuriya (2024), which studied probabilistic forecast reconciliation for Gaussian distributions, while Panagiotelis et al. (2023) provided reconciled forecasts based on the minimization of a probabilistic score through gradient descent. However, we did not leverage results from this literature for our own probabilistic approach.

### C.2  Background on forecast reconciliation through projections

We summarize the approach followed by Hyndman et al. (2011), Wickramasuriya et al. (2019), and Panagiotelis et al. (2021).

The setting is the one described in Section 2, with stochastic observations following some hierarchical structure $\boldsymbol{y} = H\boldsymbol{y}_{1:n}$; features are possibly available. Initial point forecasts $\widehat{\boldsymbol{y}}$ are provided by some regression method $\mathcal{A}$; these forecasts are possibly incoherent, i.e., do not belong to $\mathrm{Im}(H)$. The goal of forecast reconciliation is to leverage the hierarchical structure to improve the point forecasts.

A typical assumption made in this literature is that the point forecasts $\widehat{\boldsymbol{y}}$ are unbiased, or, put differently, that the forecasting errors $\widehat{\boldsymbol{s}} = \boldsymbol{y} - \widehat{\boldsymbol{y}}$ are centered. A natural performance criterion then is the mean-square error [MSE]: letting $\|\cdot\|_2$ denote the Euclidean norm and $\Sigma$ the covariance matrix of $\widehat{\boldsymbol{s}} = \boldsymbol{y} - \widehat{\boldsymbol{y}}$,

$$\mathrm{MSE}\big(\widehat{\boldsymbol{y}}, \boldsymbol{y}\big) \stackrel{\mathrm{def}}{=} \mathbb{E}\big[\|\widehat{\boldsymbol{s}}\|_2^2\big] = \mathbb{E}\big[\widehat{\boldsymbol{s}}^{\top}\widehat{\boldsymbol{s}}\big] = \mathbb{E}\Big[\mathrm{Tr}\big(\widehat{\boldsymbol{s}}\,\widehat{\boldsymbol{s}}^{\top}\big)\Big] = \mathrm{Tr}\Big(\mathbb{E}\big[\widehat{\boldsymbol{s}}\,\widehat{\boldsymbol{s}}^{\top}\big]\Big) \stackrel{\mathrm{def}}{=} \mathrm{Tr}(\Sigma)\,.$$

The equalities above may be generalized to $W$–norms (as defined in Lemma 5), where $W$ is a symmetric definite positive matrix:

$$\mathrm{MSE}\big(\widehat{\boldsymbol{y}}, \boldsymbol{y}, W\big) \stackrel{\mathrm{def}}{=} \mathbb{E}\big[\|\widehat{\boldsymbol{s}}\|_W^2\big] = \mathbb{E}\big[\widehat{\boldsymbol{s}}^{\top}W\widehat{\boldsymbol{s}}\big] = \mathbb{E}\Big[\mathrm{Tr}\big(W\widehat{\boldsymbol{s}}\widehat{\boldsymbol{s}}^{\top}\big)\Big] = \mathrm{Tr}(W\Sigma)\,.$$

Natural improvements of the unbiased point forecasts are exactly given by projections thereof onto $\mathrm{Im}(H)$, as justified below in Lemma 6. Let $P$ be a projection onto $\mathrm{Im}(H)$ and denote $\widetilde{\boldsymbol{y}} = P\widehat{\boldsymbol{y}}$. By linearity of a projection, the point forecasts $\widetilde{\boldsymbol{y}}$ are also unbiased. Since observations are coherent, we have

$$\boldsymbol{y} - \widetilde{\boldsymbol{y}} \stackrel{\mathrm{def}}{=} \boldsymbol{y} - P\widehat{\boldsymbol{y}} = P\big(\boldsymbol{y} - \widehat{\boldsymbol{y}}\big) = P\widehat{\boldsymbol{s}} \stackrel{\mathrm{def}}{=} \widetilde{\boldsymbol{s}}\,.$$

The mean-squared error of $\widetilde{\boldsymbol{y}}$ in $W$–norm thus equals

$$\mathrm{MSE}\big(\widetilde{\boldsymbol{y}}, \boldsymbol{y}, W\big) = \mathbb{E}\Big[\mathrm{Tr}\big(W\widetilde{\boldsymbol{s}}\,\widetilde{\boldsymbol{s}}^{\top}\big)\Big] = \mathrm{Tr}\Big(W\,P\,\mathbb{E}\big[\widehat{\boldsymbol{s}}\,\widehat{\boldsymbol{s}}^{\top}\big]P^{\top}\Big) = \mathrm{Tr}\big(W\,P\Sigma P^{\top}\big)\,.$$

Actually, the formula above holds more generally for all matrices $P$ such that $PH = H$.

Optimal unbiased point forecasts in the sense of the mean-square error thus exactly correspond to minimizing $\text{Tr}(W\,P\Sigma P^\top)$, a problem that we discuss below. Before we do so, we justify why (only) projections onto $\text{Im}(H)$ are considered.

**Why (only) projections onto $\text{Im}(H)$ are considered.** This follows from the lemma below, given that the literature of forecast reconciliation considers, implicitly or explicitly, two restrictions: that forecasts should be unbiased; that improved forecasts should be obtained by linear combinations of the original forecasts (and be coherent, of course).

**Lemma 6** (Hyndman et al., 2011)**.** *Assume that the point forecasts $\widehat{\boldsymbol{y}}$ are unbiased. Let $M$ be a $m \times m$ matrix taking values in the coherent subspace $\text{Im}(H)$. Then the linear combinations $\widetilde{\boldsymbol{y}} = M\widehat{\boldsymbol{y}}$ are unbiased if and only if $M$ is a projection onto $\text{Im}(H)$.*

*Proof.* Being unbiased means the following in Hyndman et al. (2011): we denote by $\boldsymbol{m} = H\boldsymbol{\beta}$ the expectation of $\boldsymbol{y}$, i.e., $\mathbb{E}[\boldsymbol{y}] = \boldsymbol{m} = H\boldsymbol{\beta}$, and assume that the model is rich enough so that all values of $\boldsymbol{\beta} \in \mathbb{R}^n$, i.e., all values of $\boldsymbol{m} \in \text{Im}(H)$, may be obtained when the specifications of the model vary.

That $\widetilde{\boldsymbol{y}} = M\widehat{\boldsymbol{y}}$ is unbiased thus corresponds to the equalities

$$\forall \boldsymbol{\beta} \in \mathbb{R}^n, \quad MH\boldsymbol{\beta} = H\boldsymbol{\beta}, \quad \text{i.e.,} \quad MH = H.$$

Now, the proof of Lemma 1 in Appendix B shows that since $M$ takes values in $\text{Im}(H)$, it is of the form $M = HG$ for some $n \times m$ matrix $G$. The equality $MH = H$ may be rewritten as $HGH = H$. Again as in the proof of Lemma 1, by multiplying both sides of this equality by $(H^\top H)^{-1}H^\top$, we obtain $GH = \text{Id}_n$, which yields $M^2 = HGHG = HG = M$. Thus, $M$ is indeed a projection onto $\text{Im}(H)$. $\qquad\square$

**Trace optimization, part 1: known covariance matrix.** As explained above, original unbiased forecasts $\widehat{\boldsymbol{y}}$ and their (still unbiased, linear) transformations $\widetilde{\boldsymbol{y}} = P\widehat{\boldsymbol{y}}$, where $P$ is a projection onto $\text{Im}(H)$ may be compared through their mean-squared errors in $W$–norm:

$$\text{MSE}(\widehat{\boldsymbol{y}}, \boldsymbol{y}, W) = \text{Tr}(W\Sigma) \qquad \text{vs.} \qquad \text{MSE}(\widetilde{\boldsymbol{y}}, \boldsymbol{y}, W) = \text{Tr}(W\,P\Sigma P^\top).$$

This consideration leads to the central result in forecast reconciliation: the optimality of the so-called minimum-trace reconciliation method from Wickramasuriya et al. (2019), formally stated as Lemma 1 in Section 4.3 and Appendix B.1.

**Trace optimization, part 2: a more practical approach.** The drawback with the approach above is that it relies on the knowledge of the covariance matrix $\Sigma$, but its advantage is that it holds for all weight matrices $W$. We now show how to exchange the roles of $W$ and $\Sigma$, and get a trace-reduction result for a given weight matrix $W$ but for all possible covariance matrices $\Gamma$, i.e., symmetric positive semi-definite matrices.

This result is inspired from Panagiotelis et al. (2021), who recommend to use the orthogonal projection in $W$–norm, whose closed-form expression (see Lemma 5) reads

$$P_W \stackrel{\text{def}}{=} H(H^\top W H)^{-1}H^\top W.$$

A Pythagorean theorem ensures that, for all point forecasts $\widehat{\boldsymbol{y}}$ and (coherent) observations $\boldsymbol{y}$,

$$\left\| \boldsymbol{y} - P_W\widehat{\boldsymbol{y}} \right\|_W^2 = \left\| P_W(\boldsymbol{y} - \widehat{\boldsymbol{y}}) \right\|_W^2 \leqslant \left\| (\boldsymbol{y} - \widehat{\boldsymbol{y}}) \right\|_W^2 \quad \text{a.s.,}$$

thus, by taking expectations,

$$\text{Tr}(W\,P_W\Sigma P_W^\top) = \text{MSE}(P_W\widehat{\boldsymbol{y}}, \boldsymbol{y}, W) \leqslant \text{MSE}(\widehat{\boldsymbol{y}}, \boldsymbol{y}, W) = \text{Tr}(W\Sigma).$$

The equality above holds no matter the specific value of the covariance matrix $\Sigma$, which corresponds to the following trace-reduction inequality, stated as the second part of Lemma 5: for all symmetric positive semi-definite matrices $\Gamma$,

$$0 \leqslant \text{Tr}(W P_W \Gamma P_W^\top) \leqslant \text{Tr}(W\Gamma). \tag{11}$$

The inequality above (i.e., Lemma 5) is a result of our own though it was inspired by both Lemma 1 and the approach by Panagiotelis et al. (2021) relying on $P_W$–projections.

### C.3 How we leveraged and transferred these results (and why it was not immediate)

**On the unnecessity of unbiasedness.** As we made clear several times in Section C.2, a key assumption in the literature of forecast reconciliation is that point forecasts are unbiased, or put differently, that the forecasting errors $\widehat{s}$ are centered.

This is in sharp contrast with the residuals $\widehat{s}$ considered in this article, thought of as signed vector-valued non-conformity scores, which we do not want (nor need) to assume centered. None of Assumptions 2–3–4 are about this. We rather assume that these scores follows a so-called elliptical distribution, with possibly a non-null expectation. Elliptical distributions were considered, not in the literature of reconciliation of point forecasts but of probabilistic forecasts, see Panagiotelis et al. (2023). Now, the proof of Theorem 3 in Appendix A reveals that our construction of prediction rectangles is such that the length of the $i$–th defining interval is given by

$$\mathfrak{L}_1\big(\widehat{C}_i(\boldsymbol{x}_{T+1})\big) = \widehat{s}_{\big(\lceil (T_{\text{calib}}+1)(1-\alpha/2)\rceil\big),i} - \widehat{s}_{\big(\lfloor (T_{\text{calib}}+1)\alpha/2\rfloor\big),i} \, .$$

Non-null expectations of the underlying elliptical distribution cancel out in the above equation, hence the unnecessity of an assumption of unbiasedness. The cancellation is only possible because we considered signed non-conformity scores (which is slightly unusual in the literature of conformal prediction).

**On the component-wise objectives targeted.** As should be clear from Sections 2 and 3, the theory provided in this article is only worth being detailed because we do not target joint-coverage guarantees (that are straightforward to get, see Appendix D below) but component-wise coverage guarantees (which are more difficult to achieve, see Appendices A and B). We had to find out a component-wise efficiency objective that we could handle. With the literature of forecast reconciliation in mind, we somehow had to build an intuition of such an efficiency criterion.

The proof of Theorem 3 in Appendix A explains how we could relate our (component-wise) small-volume objective, namely,

$$\text{minimizing} \quad \mathbb{E}\left[\sum_{i=1}^{m} w_i \, \mathfrak{L}_1\big(C_i(\boldsymbol{x}_{T+1})\big)^2\right], \tag{12}$$

to problems of the form

$$\text{minimizing} \quad \text{Tr}\big(\text{diag}(\boldsymbol{w}) \, P\Gamma P\big), \tag{13}$$

for some symmetric positive semi-definite matrix $\Gamma$, so as to leverage inequality (11), which is of our own. The proof of Theorem 4 reveals that when non-conformity scores have a bounded second-order moment, the matrix $\Gamma$ is proportional to their covariance matrix $\Sigma$, which opened the avenue of the minimum-trace approaches of Lemma 1.

**Summary of the challenges overcome.** In a nutshell, the main challenge overcome was to relate the two minimization problems (12) and (13), and in the first place, state suitably the efficiency criterion (12), which, to the best of our knowledge, is a novel criterion. The main tools used were to resort to signed vector-valued non-conformity scores, which are not necessarily unbiased, and to exploit properties of elliptical distributions, in terms of stability of the shapes of these distributions under certain affine transformations.

## D  Hierarchical SCP for joint coverage: Efficiency results

The SCP algorithms for joint coverage described in Section 2.3 and based on ellipsoidal sets are restated in algorithm boxes (with the same numbers: Algorithms 1 and 2). As their names suggest, they output prediction regions given by ellipsoids; Messoudi et al. (2022) empirically illustrated that ellipsoidal predictive regions are more efficient than hyper-rectangular ones in terms of volumes, when joint-coverage guarantees

---

**Algorithm 1** Plain multivariate SCP for joint coverage based on ellipsoidal sets

---

**Parameters:** confidence level $1 - \alpha$; regression algorithm $\mathcal{A}$; partition of $[T]$ into subsets $\mathcal{D}_{\text{train}}$, $\mathcal{D}_{\text{estim}}$ and $\mathcal{D}_{\text{calib}}$ of respective cardinalities $T_{\text{train}}$, $T_{\text{estim}}$ and $T_{\text{calib}}$; estimation procedure $\mathcal{E}$ for the matrix used to define the norm

1: Build the regressor $\widehat{\boldsymbol{\mu}}(\cdot) = \mathcal{A}\big((\boldsymbol{x}_t, \boldsymbol{y}_t)_{t \in \mathcal{D}_{\text{train}}}\big)$
2: Compute a symmetric definite positive matrix $A = \mathcal{E}\big((\boldsymbol{x}_t, \boldsymbol{y}_t)_{t \in \mathcal{D}_{\text{train}} \cup \mathcal{D}_{\text{estim}}}\big)$
3: **for** $t \in \mathcal{D}_{\text{calib}}$ **do** let $\widehat{\boldsymbol{y}}_t = \widehat{\boldsymbol{\mu}}(\boldsymbol{x}_t)$ and $\check{s}_t = \|\boldsymbol{y}_t - \widehat{\boldsymbol{y}}_t\|_A$
4: Order the $(\check{s}_t)_{t \in \mathcal{D}_{\text{calib}}}$ into $\check{s}_{(1)} \leqslant \ldots \leqslant \check{s}_{(T_{\text{calib}})}$ and define $\check{s}_{(0)} = 0$ and $\check{s}_{(T_{\text{calib}}+1)} = +\infty$
5: Let $\check{q}_{1-\alpha} = \check{s}_{\big(\lceil (T_{\text{calib}}+1)(1-\alpha) \rceil\big)}$
6: Set $\check{E}(\cdot) = \Big\{ \boldsymbol{y} \in \mathbb{R}^m : \big\| \boldsymbol{y} - \widehat{\boldsymbol{\mu}}(\cdot) \big\|_A \leqslant \check{q}_{1-\alpha} \Big\}$
7: **return** $\check{E}(\boldsymbol{x}_{T+1})$

---

**Algorithm 2** Hierarchical SCP for joint coverage based on ellipsoidal sets

---

**Parameters:** confidence level $1 - \alpha$; regression algorithm $\mathcal{A}$; partition of $[T]$ into subsets $\mathcal{D}_{\text{train}}$, $\mathcal{D}_{\text{estim}}$ and $\mathcal{D}_{\text{calib}}$ of respective cardinalities $T_{\text{train}}$, $T_{\text{estim}}$ and $T_{\text{calib}}$; estimation procedure $\mathcal{E}$ for the matrix used to define the norm

1: Build the regressor $\widehat{\boldsymbol{\mu}}(\cdot) = \mathcal{A}\big((\boldsymbol{x}_t, \boldsymbol{y}_t)_{t \in \mathcal{D}_{\text{train}}}\big)$
2: Compute a symmetric definite positive matrix $A = \mathcal{E}\big((\boldsymbol{x}_t, \boldsymbol{y}_t)_{t \in \mathcal{D}_{\text{train}} \cup \mathcal{D}_{\text{estim}}}\big)$
3: Let $P_A = H(H^\top A H)^{-1} H^\top A$ and consider $\mathring{\boldsymbol{\mu}}(\cdot) = P_A \widehat{\boldsymbol{\mu}}(\cdot)$
4: **for** $t \in \mathcal{D}_{\text{calib}}$ **do** let $\mathring{\boldsymbol{y}}_t = \mathring{\boldsymbol{\mu}}(\boldsymbol{x}_t)$ and $\mathring{s}_t = \|\boldsymbol{y}_t - \mathring{\boldsymbol{y}}_t\|_A$
5: Order the $(\mathring{s}_t)_{t \in \mathcal{D}_{\text{calib}}}$ into $\mathring{s}_{(1)} \leqslant \ldots \leqslant \mathring{s}_{(T_{\text{calib}})}$ and define $\mathring{s}_{(0)} = 0$ and $\mathring{s}_{(T_{\text{calib}}+1)} = +\infty$
6: Let $\mathring{q}_{1-\alpha} = \mathring{s}_{\big(\lceil (T_{\text{calib}}+1)(1-\alpha) \rceil\big)}$
7: Set $\mathring{E}(\cdot) = \Big\{ \boldsymbol{y} \in \mathbb{R}^m : \big\| \boldsymbol{y} - \mathring{\boldsymbol{\mu}}(\cdot) \big\|_A \leqslant \mathring{q}_{1-\alpha} \Big\}$
8: **return** $\mathring{E}(\boldsymbol{x}_{T+1})$

---

are targeted. To do so, these algorithms pick definite positive matrices $A$ based on data and rely on $A$–norms, defined as

$$\boldsymbol{u} \in \mathbb{R}^m \longmapsto \|\boldsymbol{u}\|_A \overset{\text{def}}{=} \sqrt{\boldsymbol{u}^\top A \boldsymbol{u}}$$

For instance, Johnstone & Cox (2021) suggests using the so-called Mahalanobis distance, which corresponds to taking $A$ as the inverse of the (estimated) covariance matrix of the forecasting errors. We actually present a slightly different methodology and algorithm than the one considered by Johnstone & Cox (2021), in the spirit of Algorithm 5. Indeed, in Algorithms 1 and 2, the estimation of the sample covariance matrix is made on a separate data subset (namely, $\mathcal{D}_{\text{estim}}$) to avoid potential overfitting concerns.

**Analysis.** Observations $\boldsymbol{y}_t$ are coherent, i.e., belong to $\text{Im}(H)$, and $P_A$ is the orthogonal projection in $A$–norm onto $\text{Im}(H)$, as indicated by Lemma 5 of Appendix A.2. Thus, by a Pythagorean theorem,

$$\forall t \in \mathcal{D}_{\text{calib}}, \qquad \mathring{s}_t = \|\boldsymbol{y}_t - P_A \widehat{\boldsymbol{y}}_t\|_A \leqslant \|\boldsymbol{y}_t - \widehat{\boldsymbol{y}}_t\|_A = \check{s}_t \,.$$

Thus, in particular

$$\mathring{q}_{1-\alpha} = \mathring{s}_{\big(\lceil (T_{\text{calib}}+1)(1-\alpha) \rceil\big)} \leqslant \check{s}_{\big(\lceil (T_{\text{calib}}+1)(1-\alpha) \rceil\big)} = \check{q}_{1-\alpha} \,.$$

The ellipsoids

$$\check{E}(\cdot) = \Big\{ \boldsymbol{y} \in \mathbb{R}^m : \big\| \boldsymbol{y} - \widehat{\boldsymbol{\mu}}(\cdot) \big\|_A \leqslant \check{q}_{1-\alpha} \Big\}$$

$$\text{and} \qquad \mathring{E}(\cdot) = \Big\{ \boldsymbol{y} \in \mathbb{R}^m : \big\| \boldsymbol{y} - \mathring{\boldsymbol{\mu}}(\cdot) \big\|_A \leqslant \mathring{q}_{1-\alpha} \Big\}$$

have different centers and different radii, but their shapes are similar. The inequality $\mathring{q}_{1-\alpha} \leqslant \check{q}_{1-\alpha}$ between the radii entails a similar inequality about volumes: $\mathfrak{L}_m\big(\mathring{E}(\boldsymbol{x})\big) \leqslant \mathfrak{L}_m\big(\check{E}(\boldsymbol{x})\big)$ for all $\boldsymbol{x} \in \mathbb{R}^d$.

Note that the argument above is fully deterministic and relies on no assumption on data. We therefore proved in a straightforward manner the theorem below.

**Theorem 5.** *Fix $\alpha \in (0,1)$. Under no assumption on the data, Algorithm* (2) *outputs prediction ellipsoids $\mathring{E}$ that are uniformly more efficient than the prediction ellipsoids $\check{E}$ output by Algorithm* (1)*:*

$$\forall \boldsymbol{x} \in \mathbb{R}^d, \qquad \mathfrak{L}_m\big(\mathring{E}(\boldsymbol{x})\big) \leqslant \mathfrak{L}_m\big(\check{E}(\boldsymbol{x})\big).$$

**Concluding remark: no challenge.** There was no challenge in providing a theory of efficient conformal prediction based on ellipsoidal sets for hierarchical data under a joint-coverage objective. This was not the case at all for component-wise coverage objectives, as the tools of forecast reconciliation (all related to considering projections) are not component-wise tools. The proof above actually emphasizes the complexity of results such as Theorems 3–4 and Corollary 1.

# E   Proofs of the coverage results (Theorems 1 and 2)

We conclude the theoretical part of the appendices with the proofs of the coverage results. They rely on an absolutely standard methodology in the literature of conformal prediction (see, for instance, Tibshirani et al., 2019, proof of Theorem 1), with rather immediate adaptations due to the multivariate context and to the choice of signed non-conformity scores.

The coverage results for Algorithms 3–4–5 were formally stated in Theorem 2, recalled below. The ones for Algorithms 1 and 2 were informally stated in the first part of Theorem 1 are formally stated next.

**Assumption 5.** The non-conformity scores $\check{s}_t = \|\boldsymbol{y}_t - \widehat{\boldsymbol{\mu}}(\boldsymbol{x}_t)\|_A$ are i.i.d. for $t \in \mathcal{D}_{\text{calib}} \cup \{T+1\}$, and similarly for the scores $\mathring{s}_t = \|\boldsymbol{y}_t - \mathring{\boldsymbol{\mu}}(\boldsymbol{x}_t)\|_A$. This is in particular the case when data $(\boldsymbol{x}_t, \boldsymbol{y}_t)_{1\leqslant t\leqslant T+1}$ is i.i.d.

The second part of Assumption 2 follows from the fact that $\widehat{\boldsymbol{\mu}}$, $A$, and thus $\mathring{\boldsymbol{\mu}} = P_A\widehat{\boldsymbol{\mu}}$ only depend on data from $\mathcal{D}_{\text{train}} \cup \mathcal{D}_{\text{estim}}$ and are therefore independent from the data from $\mathcal{D}_{\text{calib}} \cup \{T+1\}$.

**Theorem 6.** *Fix $\alpha \in (0,1)$. Algorithms 1 and 2, used with any regression algorithm $\mathcal{A}$ and any estimation procedure $\mathcal{E}$, ensure that whenever Assumption 5 (i.i.d. scores) holds,*

$$\mathbb{P}\big(\boldsymbol{y}_{T+1} \in \check{E}(\boldsymbol{x}_{T+1})\big) \geqslant 1 - \alpha \qquad and \qquad \mathbb{P}\big(\boldsymbol{y}_{T+1} \in \mathring{E}(\boldsymbol{x}_{T+1})\big) \geqslant 1 - \alpha.$$

*In addition, if the non-conformity scores $\big(\check{s}_t\big)_{t\in\mathcal{D}_{\text{calib}}\cup\{T+1\}}$ and $\big(\mathring{s}_t\big)_{t\in\mathcal{D}_{\text{calib}}\cup\{T+1\}}$ are almost surely distinct, then, respectively,*

$$\mathbb{P}\big(\boldsymbol{y}_{T+1} \in \check{E}(\boldsymbol{x}_{T+1})\big) \leqslant 1 - \alpha + \frac{1}{T_{\text{calib}} + 1} \qquad and \qquad \mathbb{P}\big(\boldsymbol{y}_{T+1} \in \mathring{E}(\boldsymbol{x}_{T+1})\big) \leqslant 1 - \alpha + \frac{1}{T_{\text{calib}} + 1}.$$

We recall that Theorem 2 is stated for Algorithm 5 and thus entails the same results for Algorithms 3 and 4, which are special cases of Algorithm 5.

**Assumption 2.** The residuals $\widehat{\boldsymbol{s}}_t = \boldsymbol{y}_t - \widehat{\boldsymbol{\mu}}(\boldsymbol{x}_t)$, for $t \in \mathcal{D}_{\text{calib}} \cup \{T+1\}$, are i.i.d. This is in particular the case when data $(\boldsymbol{x}_t, \boldsymbol{y}_t)_{1\leqslant t\leqslant T+1}$ is i.i.d.

The second part of Assumption 2 follows from its first part based on an argument similar to the one stated after Assumption 5.

**Theorem 2** (Coverage). *Fix $\alpha \in (0,1)$. Algorithm 5, used with any regression algorithm $\mathcal{A}$ and any function $\mathcal{P}$ outputting matrices $P$ such that $PH = H$, ensures that whenever Assumption 2 (i.i.d. scores) holds,*

$$\forall i \in [m], \quad \mathbb{P}\big(y_{T+1,i} \in \widetilde{C}_i(\boldsymbol{x}_{T+1})\big) \geqslant 1 - \alpha.$$

*In addition, if the non-conformity scores $\big(\widetilde{s}_t\big)_{t\in\mathcal{D}_{\text{calib}}\cup\{T+1\}}$ are almost surely distinct, then*

$$\forall i \in [m], \quad \mathbb{P}\big(y_{T+1,i} \in \widetilde{C}_i(\boldsymbol{x}_{T+1})\big) \leqslant 1 - \alpha + 2/(T_{\text{calib}} + 1).$$

We now formally prove these results, by starting with the most important one given the angle of this article, namely, Theorem 2. We recall that the proof schemes used are absolutely standard.

### E.1 Proof of Theorem 2

The condition $PH = H$ means that $P$ leaves elements of $\mathrm{Im}(H)$ unchanged. Since observations $\boldsymbol{y}_t$ are coherent, i.e., belong to $\mathrm{Im}(H)$, we have, for all $t \in \mathcal{D}_{\mathrm{calib}}$,

$$\widetilde{\boldsymbol{s}}_t \stackrel{\mathrm{def}}{=} \boldsymbol{y}_t - P\widehat{\boldsymbol{y}}_t = P(\boldsymbol{y}_t - \widehat{\boldsymbol{y}}_t) = P\widehat{\boldsymbol{s}}_t \,.$$

In addition, $P$ depends only on data from $\mathcal{D}_{\mathrm{train}} \cup \mathcal{D}_{\mathrm{estim}}$ and is therefore independent from data in $\mathcal{D}_{\mathrm{calib}} \cup \{T+1\}$. Assumption 2 thus entails that the projected residuals $\widetilde{\boldsymbol{s}}_t$, where $t \in \mathcal{D}_{\mathrm{calib}} \cup \{T+1\}$, are also i.i.d., thus exchangeable—which is the only property we will use in the rest of this proof.

Fix $i \in [m]$. By definition of $\widetilde{C}_i(\boldsymbol{x}_{T+1})$ and of the score $\widetilde{\boldsymbol{s}}_{T+1} = \boldsymbol{y}_{T+1} - \widetilde{\boldsymbol{\mu}}(\boldsymbol{x}_{T+1})$, the event of interest may be rewritten as

$$\left\{ y_{T+1,i} \in \widetilde{C}_i(\boldsymbol{x}_{T+1}) \right\} = \left\{ \widetilde{s}_{\left(\lfloor (T_{\mathrm{calib}}+1)\alpha/2 \rfloor\right),i} \leqslant \widetilde{s}_{T+1,i} \leqslant \widetilde{s}_{\left(\lceil (T_{\mathrm{calib}}+1)(1-\alpha/2) \rceil\right),i} \right\}. \tag{14}$$

If $\alpha \in (0,1)$ is so small that $(T_{\mathrm{calib}} + 1)\alpha/2 < 1$, i.e., $\alpha < 2/(T_{\mathrm{calib}} + 1)$, then

$$\widetilde{s}_{\left(\lfloor (T_{\mathrm{calib}}+1)\alpha/2 \rfloor\right),i} = \widetilde{s}_{(0)} = -\infty \quad \text{and} \quad \widetilde{s}_{\left(\lceil (T_{\mathrm{calib}}+1)(1-\alpha/2) \rceil\right),i} = \widetilde{s}_{(T_{\mathrm{calib}}+1)} = +\infty \,.$$

Thus, $\mathbb{P}\big(y_{T+1,i} \in \widetilde{C}_i(\boldsymbol{x}_{T+1})\big) = 1$ satisfies the claims $\geqslant 1 - \alpha$ and $\leqslant 1 - \alpha + 2/(T_{\mathrm{calib}} + 1)$.

Otherwise, $\widetilde{s}_{\left(\lfloor (T_{\mathrm{calib}}+1)\alpha/2 \rfloor\right),i}$ and $\widetilde{s}_{\left(\lceil (T_{\mathrm{calib}}+1)(1-\alpha/2) \rceil\right),i}$ correspond to some $\widetilde{s}_{t,i}$ and $\widetilde{s}_{t',i}$ for some $t, t' \in \mathcal{D}_{\mathrm{calib}}$. We apply arguments of exchangeability in the latter case. The new score $\widetilde{s}_{T+1,i}$ is equally likely to fall into any of the $T_{\mathrm{calib}} + 1$ intervals defined by the $(\widetilde{s}_t)_{t \in \mathcal{D}_{\mathrm{calib}}}$. More formally, by Assumption 2, and when scores are almost-surely distinct,

$$\mathbb{P}\big(\widetilde{s}_{T+1,i} < \widetilde{s}_{(1),i}\big) = \mathbb{P}\big(\widetilde{s}_{T+1,i} > \widetilde{s}_{(T_{\mathrm{calib}}),i}\big) = \frac{1}{T_{\mathrm{calib}} + 1}$$

$$\text{and} \qquad \forall k \in [T_{\mathrm{calib}} - 1], \qquad \mathbb{P}\big(\widetilde{s}_{(k),i} < \widetilde{s}_{T+1,i} < \widetilde{s}_{(k+1),i}\big) = \frac{1}{T_{\mathrm{calib}} + 1} \,.$$

Therefore, when scores are almost-surely distinct, the event of interest (14) rewrites

$$\left\{ y_{T+1,i} \in \widetilde{C}_i(\boldsymbol{x}_{T+1}) \right\} \stackrel{\text{a.s.}}{=} \left\{ \widetilde{s}_{\left(\lfloor (T_{\mathrm{calib}}+1)\alpha/2 \rfloor\right),i} < \widetilde{s}_{T+1,i} < \widetilde{s}_{\left(\lceil (T_{\mathrm{calib}}+1)(1-\alpha/2) \rceil\right),i} \right\}$$

and has a probability

$$\begin{aligned} \mathbb{P}\big(y_{T+1,i} \in \widetilde{C}_i(\boldsymbol{x}_{T+1})\big) &= \frac{\lceil (T_{\mathrm{calib}}+1)(1-\alpha/2) \rceil - \lfloor (T_{\mathrm{calib}}+1)\alpha/2 \rfloor}{T_{\mathrm{calib}} + 1} \\ &\leqslant \frac{\big((T_{\mathrm{calib}}+1)(1-\alpha/2) + 1\big) - \big((T_{\mathrm{calib}}+1)\alpha/2 - 1\big)}{T_{\mathrm{calib}} + 1} \\ &= 1 - \alpha + \frac{2}{T_{\mathrm{calib}} + 1} \,, \end{aligned}$$

as claimed.

We now prove that $\mathbb{P}\big(y_{T+1,i} \in \widetilde{C}_i(\boldsymbol{x}_{T+1})\big) \geqslant 1 - \alpha$ whether or not scores are almost-surely distinct. To do so, we show below that

$$\forall k \in [T_{\mathrm{calib}}], \quad \mathbb{P}\big(\widetilde{s}_{T+1,i} \leqslant \widetilde{s}_{(k),i}\big) \geqslant \frac{k}{T_{\mathrm{calib}} + 1} \quad \text{and} \quad \mathbb{P}\big(\widetilde{s}_{T+1,i} < \widetilde{s}_{(k),i}\big) \leqslant \frac{k}{T_{\mathrm{calib}} + 1} \,, \tag{15}$$

so that, given the rewriting (14), we will end up with

$$\begin{aligned} \mathbb{P}\big(y_{T+1,i} \in \widetilde{C}_i(\boldsymbol{x}_{T+1})\big) &\geqslant \frac{\lceil (T_{\mathrm{calib}}+1)(1-\alpha/2) \rceil - \lfloor (T_{\mathrm{calib}}+1)\alpha/2 \rfloor}{T_{\mathrm{calib}} + 1} \\ &\geqslant \frac{(T_{\mathrm{calib}}+1)(1-\alpha/2) - (T_{\mathrm{calib}}+1)\alpha/2}{T_{\mathrm{calib}} + 1} = 1 - \alpha \,. \end{aligned}$$

It only remains to show (15). The event $\left\{\widetilde{s}_{T+1,i} \leqslant \widetilde{s}_{(k),i}\right\}$ is exactly the fact that $\widetilde{s}_{T+1,i}$ is among the $k$ smallest elements of the $(\widetilde{s}_t)_{t \in \mathcal{D}_{\mathrm{calib}} \cup \{T+1\}}$. By exchangeability, the probability of the latter event is at least $k/(T_{\mathrm{calib}} + 1)$; it may be larger if several scores take the same value as the $k$–th smallest value. Similarly, the event $\left\{\widetilde{s}_{T+1,i} < \widetilde{s}_{(k),i}\right\}$ is exactly the fact that $\widetilde{s}_{T+1,i}$ is among the $k$ smallest elements of the $(\widetilde{s}_t)_{t \in \mathcal{D}_{\mathrm{calib}} \cup \{T+1\}}$ and that there are no ties at the $k$–th smallest value. Due to the additional no-tie condition, and by exchangeability, the probability of the latter event is at most $k/(T_{\mathrm{calib}} + 1)$.

### E.2 Proof of Theorem 6

We first note that by definition,

$$\left\{\boldsymbol{y}_{T+1} \in \check{E}(\boldsymbol{x}_{T+1})\right\} = \left\{\check{s}_{T+1} \leqslant \check{s}_{\left(\lceil (T_{\mathrm{calib}}+1)(1-\alpha)\rceil\right)}\right\},$$

which replaces the equality (14) in the proof above. The same classical arguments that were already detailed above may then be adapted. The proof is identical for $\mathring{E}(\boldsymbol{x}_{T+1})$.

## F Full details for the simulations: settings, methodology, results

In this appendix, we provide the full details of the specifications and of the results of the numerical experiments summarized in Section 5. The experimental setting described in Section F.1 is common to joint and component-wise coverage results.

### F.1 Experimental setting

The objective of the experiments on synthetic data is to illustrate how performance varies depending on the complexities of the hierarchies (in terms of depths and nodes) considered while controlling for the number $T = 10^6$ of observations available and for the forecasting task.

#### F.1.1 Computational resources used

All experiments were conducted on a high-performance computing environment with a limited amount of 95 compute nodes per user. Each node is equipped with two CPUs, 36 cores in total, and 192 GiB of RAM. The computational workload was parallelized at the level of simulation jobs, where each job corresponds to one run for a given configuration. One run consists of the following steps: 1. data generation (Appendix F.1.2); 2. forecasting (Appendix F.1.4); 3. conformal prediction (Section 2.3.2).

The complete experiment (with $N = 10^3$ runs) on the 4 smallest hierarchies was completed within approximately 2 hours using the parallelized setup, which emphasizes the computational efficiency of our approach. Each run for the 2 most complex hierarchies took approximatively 7 hours because of high dimension $m$. Due to memory constraints, the parallelization for these two configurations requires an entire node for each run. Since $N = 10^3$ runs were computed for each hierarchy, the experiment took approximately 3 days in the 2 most complex cases – accounting for the 95-node parallelism ($10^3 \times 7/95 \approx 73.7$ hours). In addition, we ran preliminary experiments to determine the types of hierarchies that would be possibly interesting.

#### F.1.2 Data generation

Several parameters need to be set to generate i.i.d. hierarchical data $(\boldsymbol{x}_t, \boldsymbol{y}_t)_{1 \leqslant t \leqslant T}$. The most critical one in our simulations is the structural matrix $H$.

**Choice of the structural matrix.** We used 6 different hierarchical configurations, of two main types: A and B. The simplest type-A hierarchical configuration (numbered Configuration 1) is the following:

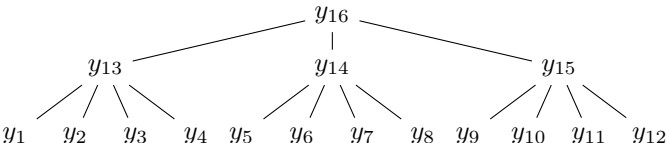

It is of depth 3, features a root node, $3^k$ child nodes, each of them having $4^k$ leaves, where $k = 1$. Configurations 3 and 5 are also of type A, for values $k = 2$ and $k = 3$, respectively.

The simplest type-B hierarchical configuration (numbered Configuration 2) has the same number of leaves but is deeper:

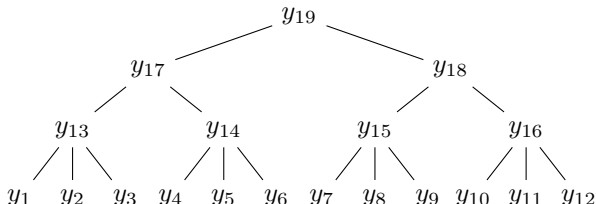

It is of depth 4, features a root node, $2^k$ child nodes, $2^k \times 2^k$ grandchild nodes, each of them having $3^k$ leaves, where $k = 1$. Configurations 4 and 6 are also of type B, for values $k = 2$ and $k = 3$, respectively.

We summarize the complexities of the hierarchical configurations considered in the table below, where we recall that $m$ denotes the total number of nodes, and $n = 12^k$ the number of leaves (which only depends on $k$, not on the type).

| Config. | Type | $k$ | depth | $n$ | $m$ |
|---|---|---|---|---|---|
| 1 | A | 1 | 3 | 12 | 16 |
| 2 | B | 1 | 4 | 12 | 19 |
| 3 | A | 2 | 3 | 144 | 154 |
| 4 | B | 2 | 4 | 144 | 165 |
| 5 | A | 3 | 3 | 1,728 | 1,756 |
| 6 | B | 3 | 4 | 1,728 | 1,801 |

For the sake of completeness, and for readability of the code submitted, we also display the general form of the structural matrices $H$, which are of size $m \times n$, for type-A and type-B configurations, respectively:

$$
H = \begin{pmatrix}
 & & & & \mathrm{Id}_{12^k} & & & \\
\underbrace{1 \quad 1 \quad \cdots \quad 1}_{4^k} & & & & & & \\
 & & \underbrace{1 \quad 1 \quad \cdots \quad 1}_{4^k} & & & & \\
 & & & & \ddots & & \\
 & & & & & \underbrace{1 \quad 1 \quad \cdots \quad 1}_{4^k} & \\
1 & & & \cdots & & & 1
\end{pmatrix}
$$

and

$$H = \begin{pmatrix} & & & & & & \mathrm{Id}_{12^k} & & & & \\ \underbrace{1 \quad \cdots \quad 1}_{3^k} & & & & & & & & & \\ & & \cdots & & & & & & & \\ & & \underbrace{1 \quad \cdots \quad 1}_{3^k} & & & & & & \\ & & & & \cdots & & & & & \\ & & & & & \underbrace{1 \quad \cdots \quad 1}_{3^k} & & & \\ & & & & & & \cdots & & & \\ & & & & & & & \underbrace{1 \quad \cdots \quad 1}_{3^k} & \\ \underbrace{1 \quad\quad\quad\quad \cdots \quad\quad\quad\quad 1}_{6^k} & & & & & & & & \\ & & & & \cdots & & & & & \\ & & & & & \underbrace{1 \quad\quad\quad\quad \cdots \quad\quad\quad\quad 1}_{6^k} & & & \\ 1 & & & & \cdots & & & & 1 \end{pmatrix}$$

**The other data-generation steps.** We now explain how to generate the observations at $n = 12^k$ leaves (i.e., at the most disaggregated level). They are obtained as realizations of a model with additive effects of three covariates and with an additional multivariate noise. All of the covariates, effect functions and correlations will be drawn at random as described in the following paragraphs.

**Data generation: initial draw of the parameters.** For each of the $N$ runs, we first pick at random a function $f : \mathbb{R}^3 \to \mathbb{R}^n$ and a correlation matrix $R$. We do so as explained later in this description.

**Data generation: draw of $T$–sample.** Then, given $H$, $f$, and $R$, we draw a $T$–sample $(\boldsymbol{x}_t, \boldsymbol{y}_t)_{1 \leqslant t \leqslant T}$ of data as follows. First, the features $\boldsymbol{x}_t \in \mathbb{R}^3$ are drawn i.i.d. according to a Gaussian distribution:

$$\boldsymbol{x}_t = \begin{bmatrix} x_{t,1} \\ x_{t,2} \\ x_{t,3} \end{bmatrix} \sim \mathcal{N}\left( \begin{bmatrix} 10 \\ -5 \\ 5 \end{bmatrix}, \begin{bmatrix} 2 & 0 & 0 \\ 0 & 2 & 0 \\ 0 & 0 & 1 \end{bmatrix} \right).$$

Next, the observations $\boldsymbol{y}_{t,1:n} \in \mathbb{R}^n$ at the most disaggregated level are generated i.i.d. according to the following additive model:

$$\boldsymbol{y}_{t,1:n} = f(\boldsymbol{x}_t) + \boldsymbol{\varepsilon}_t, \qquad \text{where} \qquad \boldsymbol{\varepsilon}_t \sim \mathcal{N}\left( \begin{bmatrix} 10 \\ \vdots \\ 10 \end{bmatrix}, R \right). \tag{16}$$

The complete vectors of observations are finally given by $\boldsymbol{y}_t = H\boldsymbol{y}_{t,1:n}$.

**Data generation: initial draw of the parameters, continued.** To obtain the correlation matrix $R$, a matrix $M$ is drawn component-wise, in an i.i.d. manner: the $M_{i,j}$, where $i, j \in [n]$, follow a standard Gaussian distribution $\mathcal{N}(0, 1)$. Then,

$$D = \sqrt{\mathrm{Diag}(M^\top M)} \qquad \text{and} \qquad R = 100 \, D^{-1} M^\top M D^{-1} \,.$$

We draw $f = (f_1, \ldots, f_n)^\top$ component-wise. To do so, we consider the following basis functions $\mathbb{R}^3 \to \mathbb{R}$:

$$\begin{aligned} g_1(\boldsymbol{x}_t) &= x_{1,t} \,, & g_5(\boldsymbol{x}_t) &= x_{2,t} \,, & g_9(\boldsymbol{x}_t) &= x_{3,t} \,, \\ g_2(\boldsymbol{x}_t) &= x_{1,t}^2 \,, & g_6(\boldsymbol{x}_t) &= x_{2,t}^2 \,, & g_{10}(\boldsymbol{x}_t) &= x_{3,t}^2 \,, \\ g_3(\boldsymbol{x}_t) &= \sin(x_{1,t}) \,, & g_7(\boldsymbol{x}_t) &= \cos(x_{2,t}) \,, & g_{11}(\boldsymbol{x}_t) &= \exp(x_{3,t}) \,, \\ g_4(\boldsymbol{x}_t) &= \log\left(|x_{1,t}| + 1\right) \,, & g_8(\boldsymbol{x}_t) &= \sqrt{x_{2,t}} \,. \end{aligned}$$

We now explain how $f_i$ is drawn for each component $i \in [n]$. First, the number $k_i$ of effects to consider is drawn uniformly in the set $[11]$. Then, we sample with replacement $k_i$ basis functions in the set $\{g_1, \ldots, g_{11}\}$; we denote them by $h_{i,1}, \ldots, h_{i,k_i}$. Finally, we add signs: we draw $k_i$ i.i.d. symmetric Rademacher random variables $r_{i,1}, \ldots, r_{i,k_i}$ (i.e., variables that take values $-1$ and $1$ with respective probabilities $1/2$). All in all, we let

$$f_i = \sum_{j=1}^{k_i} r_{i,j}\, h_{i,j}\,.$$

### F.1.3  Data splitting

We take $T = 10^6$ since a large number of data points is necessary to provide an accurate estimate of the covariance matrix $\Sigma$ for large number of nodes $m$. These $T$ observations are first randomly split into two subsets, containing 80% and 20% of the data.

The smaller subset is referred to as the test set and is denoted by $\mathcal{D}_{\text{test}}$. Its data points will play the role of the $(\boldsymbol{x}_{T+1}, \boldsymbol{y}_{T+1})$, as explained later in Appendices F.2–F.3.

The larger subset of 80% of the data is split again in three sub-subsets, containing 40% (train set $\mathcal{D}_{\text{train}}$), 20% (estimation set $\mathcal{D}_{\text{estim}}$), and 20% (calibration set $\mathcal{D}_{\text{calib}}$) of the total data. These data points are used to construct the prediction regions, which are either of an ellipsoidal form (for Algorithms 1–2):

$$\boldsymbol{x} \longmapsto E(\boldsymbol{x}) = \left\{ \boldsymbol{y} \in \mathbb{R}^m : \left\| \boldsymbol{y} - \widetilde{\boldsymbol{\mu}}(\boldsymbol{x}) \right\|_A \leqslant q_{1-\alpha} \right\},$$

or are hyper-rectangular (in Algorithms 3–4–5):

$$\boldsymbol{x} \longmapsto \prod_{i=1}^m \widetilde{C}_i(\boldsymbol{x}) = \prod_{i=1}^m \left[ \widetilde{\mu}_i(\boldsymbol{x}) + \widetilde{q}^{(i)}_{\alpha/2},\ \widetilde{\mu}_i(\boldsymbol{x}) + \widetilde{q}^{(i)}_{1-\alpha/2} \right],$$

and only depend on the features $\boldsymbol{x}$ through the centers $\widetilde{\mu}_i(\boldsymbol{x})$. The algorithms that do not use an estimation set $\mathcal{D}_{\text{estim}}$, i.e., Algorithms 3–4, simply ignore data points in $\mathcal{D}_{\text{estim}}$.

### F.1.4  Train set: regression algorithm $\mathcal{A}$

The last piece to fully define the procedures implemented is to describe the regression algorithm $\mathcal{A}$ given as input to Algorithms 1–2–3–4–5. This algorithm will be given by a base forecasting method run independently at each node.

Before we describe the base forecasting method, we mention a constraint that we impose. It turns out that in the practice of hierarchical forecasting, explanatory variables are not necessarily all available at every level of granularity within the hierarchical structure. This also makes the hierarchy more interesting from a forecasting viewpoint since the observations at some nodes are harder to predict than others.

To reproduce this specificity, for each of the nodes at the most disaggregated level, indexed by $i \in [n]$, we draw independently a Bernoulli variable $\rho_i$ with parameter 0.8: if $\rho_i = 1$, then the forecasting strategy may use the entire vectors $\boldsymbol{x}_t$; otherwise, the forecasting strategy only accesses to $\boldsymbol{x}'_t = (x_{t,1}, x_{t,2})^\top$. For inner nodes $i > n$, we set $\rho_i = 1$.

It only remains to describe the forecasting strategy used independently at each node $i \in [m]$, based on features that lie in $\mathbb{R}^2$ or $\mathbb{R}^3$. Given the additive nature (16) of the data, a natural choice is to resort to the theory of estimation of generalized additive models, see a reminder below.

For each $i \in [m]$, depending on $\rho_i$, the regression estimate $\widehat{\mu}_i$ produced for the $i$–th component of the $\boldsymbol{y}$ is thus of the form

$$\widehat{\mu}_i : \boldsymbol{x} \longmapsto \begin{cases} \widehat{\mu}^{(1)}_i(x_1) + \widehat{\mu}^{(2)}_i(x_2) + \widehat{\mu}^{(3)}_i(x_3), & \text{if } \rho_i = 1 \\ \widehat{\mu}^{(1)}_i(x_1) + \widehat{\mu}^{(2)}_i(x_2), & \text{otherwise.} \end{cases}$$

**Reminder on generalized additive models.** Generalized additive models (GAMs, Wood, 2017) are a popular modeling for many real-world problems, like electricity demand (see Wood et al., 2015). They form a good compromise between forecast efficiency and interpretability. In that setting, univariate response variables $z_t$ based on features $\boldsymbol{x}_t \in \mathbb{R}^d$, where $t \in [T]$, are expressed as

$$z_t = \beta_0 + \sum_{j=1}^{d} m_j(x_{t,j}) + \varepsilon_t \,, \tag{17}$$

where the $m_j : \mathbb{R} \to \mathbb{R}$ do not depend on $t$ and are called the non-linear effects, and where the $\varepsilon_t$ are i.i.d. random noises. The non-linear effects $m_j$ are each possibly decomposed on a given spline basis $(B_{j,k})$, chosen by the forecasting agent:

$$m_j : x \in \mathbb{R} \longmapsto \sum_{k=1}^{K_j} \beta_{j,k} \, B_{j,k}(x) \,,$$

where $K_j$ depends on the dimension of the spline basis. Estimating the model (17) then amounts to estimating the coefficients $\beta_{j,k}$.

At a high level, we may write that the estimation of these coefficients $\beta_{j,k}$ is performed via by penalized least-squares, where the penalty term therein involves the second derivatives of the functions $m_j$, forcing the effects to be smooth. We resorted to the R package `mgcv` of Wood (2023) in our simulations, with the basis by default: the thin plate spline basis, with a maximum number of degrees of freedom of 10, and coefficient `sp=1` for fixed penalties. To improve computational efficiency, we estimate the spline coefficients using the `bam` function with the `discrete=TRUE` option, as described in Wood et al. (2015). This allows for optimal parallelization and data compression.

### F.2 Evaluation and results: SCP for joint coverage based on ellipsoidal sets

We first illustrate SCP for joint coverage, i.e., Theorem 1 (and its formal restatements, Theorems 5 and 6).

#### F.2.1 Evaluation on one given test set

The metrics we consider in Section 2 for ellipsoidal prediction sets are both in terms of joint-coverage probability and of expected volume, where in both cases, probabilities are with respect to all data (the observations to be predicted as well as the data used to compute the prediction regions).

For the (conditional) joint-coverage probability, we resort to Monte-Carlo-type estimates, based on data in the test set $\mathcal{D}_{\text{test}}$, with cardinality $T_{\text{test}} = 2 \cdot 10^5$: given the specifications of the experiment (i.e., $f$, $R$, and the $\rho_i$) and given data in the sets $\mathcal{D}_{\text{train}}$, $\mathcal{D}_{\text{estim}}$, and $\mathcal{D}_{\text{calib}}$,

$$\widehat{c} \stackrel{\text{def}}{=} \frac{1}{T_{\text{test}}} \sum_{t \in \mathcal{D}_{\text{test}}} \mathbb{1}_{\left\{ \boldsymbol{y}_t \in \mathring{E}(\boldsymbol{x}_t) \right\}}$$

The volume of a $m$–dimensional ellipsoid with center $\boldsymbol{y}_0$, determined by an $A$–norm, and with radius $r$, i.e.,

$$E(\boldsymbol{y}_0, A, r) = \left\{ \boldsymbol{y} \in \mathbb{R}^m : \|\boldsymbol{y} - \boldsymbol{y}_0\|_A \leqslant r \right\} \,,$$

equals

$$\mathfrak{L}_m\big(E(\boldsymbol{y}_0, A, r)\big) = r^m \det\big(A\big)^{-1/2} \mathfrak{L}_m\big(B_m\big) \,,$$

where $B_m = \{\boldsymbol{y} \in \mathbb{R}^m : \|\boldsymbol{y}\|_2 \leqslant 1\}$ is the Euclidean unit ball. To control for the values of $m$ in the hierarchical configurations considered, we rather report the following normalized version of the volume:

$$v\big(E(\boldsymbol{y}_0, A, r)\big) = r \det\big(A\big)^{-1/(2m)} \,,$$

which only depends on $A$ and $r$. Now, the matrix $A$ and the radius $r$ of the ellipsoidal prediction regions are constant over $\mathcal{D}_{\text{test}}$ (only the centers vary) and we may thus compute with the formula above the conditional

expectation of the normalized volume, conditionally to the data in the sets $\mathcal{D}_{\text{train}}$, $\mathcal{D}_{\text{estim}}$, and $\mathcal{D}_{\text{calib}}$ and to the specifications of the experiment.

We actually run the entire procedure a large number of times ($N = 10^3$) to get unconditional probabilities and expectations, as described next.

### F.2.2 Evaluation thanks to Monte-Carlo estimates based on large numbers of runs

We run $N = 10^3$ times the entire procedure described above and get, for each run, an estimate of the conditional coverage probability and the exact value of the conditional expectation of the normalized volume, which we denote by

$$\widehat{c}^{(r)} \quad \text{and} \quad v^{(r)}, \qquad \text{where} \quad r \in [10^3].$$

We in turn get the following estimates for the unconditional coverage probability and unconditional expectation of the normalized volume:

$$\overline{c} \overset{\text{def}}{=} \frac{1}{10^3} \sum_{r=1}^{10^3} \widehat{c}^{(r)} \qquad \text{and} \qquad \overline{v} \overset{\text{def}}{=} \frac{1}{10^3} \sum_{r=1}^{10^3} v^{(r)}.$$

These empirical means estimate the underlying true values up to 95%–confidence errors margins given by

$$\gamma_c \overset{\text{def}}{=} 1.96 \, \frac{\text{std}\left(\widehat{c}^{(1)}, \ldots, \widehat{c}^{(10^3)}\right)}{\sqrt{10^3}} \qquad \text{and} \qquad \gamma_v \overset{\text{def}}{=} 1.96 \, \frac{\text{std}\left(v^{(1)}, \ldots, v^{(10^3)}\right)}{\sqrt{10^3}},$$

where $\text{std}(x_1, \ldots, x_{10^3})$ denotes the standard deviation of the arguments provided:

$$\text{std}(x_1, \ldots, x_{10^3}) = \sqrt{\frac{1}{10^3} \sum_{r=1}^{10^3} (x_r - \overline{x}_{10^3})^2}, \qquad \text{where} \quad \overline{x}_{10^3} = \frac{1}{10^3} \sum_{r=1}^{10^3} x_r.$$

Finally, we report in the table below the following point estimates and associated confidence intervals on the underlying unconditional probabilities and expectations:

$$\overline{c} \quad \text{and} \quad \overline{v}, \qquad \qquad \left[\overline{c} \pm \gamma_c\right] \quad \text{and} \quad \left[\overline{v} \pm \gamma_v\right]. \tag{18}$$

### F.2.3 Results: joint coverages and normalized volumes of ellipsoidal prediction regions

Section 5 only reported the results in terms of volumes and for the Mahalanobis distance, as in Johnstone & Cox (2021), which corresponds to considering

$$A = \mathcal{E}\big((\boldsymbol{x}_t, \boldsymbol{y}_t)_{t \in \mathcal{D}_{\text{train}} \cup \mathcal{D}_{\text{estim}}}\big) = \widehat{\Sigma}^{-1}$$

in Algorithms (1) and (2). In particular, the non-conformity scores are given by the $\widehat{\Sigma}^{-1}$–norm $\|\cdot\|_{\widehat{\Sigma}^{-1}}$ of the forecast errors on the calibration set. This choice is natural to produce prediction regions that fit the underlying distribution, as it takes into account the dependencies within the components of the multivariate target (Johnstone & Cox, 2021, Messoudi et al., 2022). However, for the sake of completeness, we also consider estimation procedures $\mathcal{E}$ that produce diagonal matrices, namely $A = \text{Id}_m$ and $A = \text{Diag}(\widehat{\Sigma})^{-1}$, the Moore-Penrose pseudo-inverse of the diagonal matrix $\text{Diag}(\widehat{\Sigma})$ defined at the end of Section 4.3.

We only report the results achieved for the smallest hierarchies, i.e., Configurations 1–2, and illustrate Theorem 1 (and its formal restatements, Theorems 5 and 6)

Coverage-wise, the next table indicates that the nominal joint-coverage of $1 - \alpha = 90\%$ is achieved irrespective of the algorithm considered or choice of matrix $A$.

| Matrix $A$ | Config. | Alg. (1) | Alg. (2) |
|---|---|---|---|
| $\mathrm{Id}_m$ | 1 | $90\% \pm 0.0059\%$ | $90\% \pm 0.0059\%$ |
| | 2 | $90\% \pm 0.0061\%$ | $90\% \pm 0.0061\%$ |
| $\mathrm{Diag}(\widehat{\Sigma})^{-1}$ | 1 | $90\% \pm 0.0060\%$ | $90\% \pm 0.0063\%$ |
| | 2 | $90\% \pm 0.0060\%$ | $90\% \pm 0.0059\%$ |
| $\widehat{\Sigma}^{-1}$ | 1 | $90\% \pm 0.0058\%$ | $90\% \pm 0.0059\%$ |
| | 2 | $90\% \pm 0.0060\%$ | $90\% \pm 0.0061\%$ |

Efficiency-wise, and as expected, we obtain smaller volumes with Algorithm (2), that performs a projection step, than with the benchmark, Algorithm (1). Note also that we recover one key finding by Johnstone & Cox (2021) and Messoudi et al. (2022): that prediction regions are particularly small with the choice $A = \widehat{\Sigma}^{-1}$.

| Matrix $A$ | Config. | Alg. (1) | Alg. (2) |
|---|---|---|---|
| $\mathrm{Id}_m$ | 1 | $284 \pm 12$ | $\mathbf{257 \pm 11}$ |
| | 2 | $285 \pm 12$ | $\mathbf{249 \pm 10}$ |
| $\mathrm{Diag}(\widehat{\Sigma})^{-1}$ | 1 | $314 \pm 6.8$ | $\mathbf{310 \pm 6.7}$ |
| | 2 | $370 \pm 7.9$ | $\mathbf{362 \pm 7.7}$ |
| $\widehat{\Sigma}^{-1}$ | 1 | $17.4 \pm 0.6$ | $\mathbf{16.5 \pm 0.55}$ |
| | 2 | $3.36 \pm 0.12$ | $\mathbf{3.19 \pm 0.11}$ |

### F.3 Evaluation and results: component-wise SCP

We illustrate the component-wise results for SCP, namely, Theorems 3 and 4. We follow the same structure as in Appendix F.2.

#### F.3.1 Evaluation on one given test set

Component-wise SCP prediction sets should be evaluated both in terms of component-wise coverage probabilities and of expected total squared lengths, for a given vector of weights; we actually pick $\boldsymbol{w} = \mathbf{1} = (1, \ldots, 1)^\top$.

On the test set of a given run, we estimate the conditional coverage probabilities and compute exactly the conditional expectations of the lengths, given the specifications of the experiment (i.e., $f$, $A$, and the $\rho_i$) and given data in the sets $\mathcal{D}_{\mathrm{train}}$, $\mathcal{D}_{\mathrm{estim}}$, and $\mathcal{D}_{\mathrm{calib}}$: for each $i \in [m]$,

$$\widehat{c}_i \stackrel{\mathrm{def}}{=} \frac{1}{T_{\mathrm{test}}} \sum_{t \in \mathcal{D}_{\mathrm{test}}} \mathbb{1}_{\left\{ y_{t,i} \in \widetilde{C}_i(\boldsymbol{x}_t) \right\}} \qquad \text{and} \qquad \ell_i \stackrel{\mathrm{def}}{=} \mathfrak{L}_1\big(\widetilde{C}_i(\cdot)\big) \, ;$$

we note as before that the lengths of the intervals $\widetilde{C}_i(\boldsymbol{x})$ do not depend on $\boldsymbol{x}$ and denote by $\mathfrak{L}_1\big(\widetilde{C}_i(\cdot)\big)$ their common value.

#### F.3.2 Evaluation thanks to Monte-Carlo estimates based on large numbers of runs

We propose component-wise metrics (for the figures) and global metrics (for the tables).

**Component-wise metrics.** We perform $N = 10^3$ runs and get, for each run and each component $i \in [m]$, an estimate of the conditional coverage probability and the exact value of the conditional expectation of the length, which we denote by:

$$\widehat{c}_i^{(r)} \quad \text{and} \quad \ell_i^{(r)}, \qquad \text{where} \quad i \in [m] \quad \text{and} \quad r \in [10^3] \, .$$

We in turn get the following estimates for the unconditional coverage probabilities and unconditional expectation of the squared lengths: for each $i \in [m]$,

$$\bar{c}_i \stackrel{\mathrm{def}}{=} \frac{1}{10^3} \sum_{r=1}^{10^3} \widehat{c}_i^{(r)} \qquad \text{and} \qquad \bar{\ell}_i \stackrel{\mathrm{def}}{=} \frac{1}{10^3} \sum_{r=1}^{10^3} \big(\ell_i^{(r)}\big)^2 \, .$$

These empirical means estimate the underlying true values up to 95%–confidence errors margins given by

$$\gamma_{c,i} \stackrel{\text{def}}{=} 1.96 \, \frac{\text{std}\left(\widehat{c}_i^{(1)}, \ldots, \widehat{c}_i^{(10^3)}\right)}{\sqrt{10^3}} \qquad \text{and} \qquad \gamma_{\ell,i} \stackrel{\text{def}}{=} 1.96 \, \frac{\text{std}\left(\left(\ell_i^{(1)}\right)^2, \ldots, \left(\ell_i^{(10^3)}\right)^2\right)}{\sqrt{10^3}} \, .$$

For scaling issues on the lengths, we rather report, in our experiments, when dealing with component-wise quantities (i.e., in the figures), the following point estimates and associated confidence intervals on the underlying unconditional probabilities and expectations: for all $i \in [m]$,

$$\overline{c}_i \quad \text{and} \quad \sqrt{\overline{\ell}_i} \,, \qquad\qquad \left[\overline{c}_i \pm \gamma_{c,i}\right] \quad \text{and} \quad \left[\sqrt{\overline{\ell}_i - \gamma_{\ell,i}}, \sqrt{\overline{\ell}_i + \gamma_{\ell,i}}\right]. \qquad (19)$$

**Global metrics: total lengths.** We also report results on the total lengths, i.e., for the quantities appearing in the efficiency result of Theorem 3, where we recall that $\boldsymbol{w} = \mathbf{1}$.

In the same spirit as right above, we consider

$$L_\bullet^{(r)} = \sum_{i=1}^{m} \left(\ell_i^{(r)}\right)^2, \quad \text{where} \ \ r \in \left[10^3\right], \qquad \text{and} \qquad \overline{L}_\bullet \stackrel{\text{def}}{=} \frac{1}{10^3} \sum_{r=1}^{10^3} L_\bullet^{(r)} = \sum_{i=1}^{m} \overline{\ell}_i \,.$$

This empirical mean estimates the underlying expected sum of the squared lengths up to 95%–confidence errors margins given by

$$\gamma_{L,\bullet} \stackrel{\text{def}}{=} 1.96 \, \frac{\text{std}\left(L_\bullet^{(1)}, \ldots, L_\bullet^{(10^3)}\right)}{\sqrt{10^3}} \,,$$

For the same scaling issues as above, we rather report in our experiments roots of the quantities defined above. Actually, to get symmetric intervals and report more easily the results in the table of Appendix F.3.3, we provide slightly larger confidence intervals (based on the inequality $\sqrt{u+v} \leqslant \sqrt{u} + \sqrt{v}$). More precisely, we report in the table the following point estimates and associated confidence intervals:

$$\sqrt{\overline{L}_\bullet} \,, \qquad\qquad \left[\sqrt{\overline{L}_\bullet} - \sqrt{\gamma_{L,\bullet}}, \sqrt{\overline{L}_\bullet} + \sqrt{\gamma_{L,\bullet}}\right]. \qquad (20)$$

### F.3.3 Results: total lengths of hyper-rectangular prediction sets

In this section, we go over the results presented in Section 5. For the convenience of the reader, we copy below the table presented in the aforementioned section, as well as the comments made therein.

| Config. | Direct | OLS | WLS | Combi | MinT |
|---------|--------|-----|-----|-------|------|
| 1 | $876 \pm 254$ | $787 \pm 226$ | $322 \pm 131$ | $364 \pm 101$ | $\mathbf{216 \pm 47}$ |
| 2 | $871 \pm 253$ | $753 \pm 216$ | $308 \pm 116$ | $361 \pm 92$ | $\mathbf{246 \pm 51}$ |
| 3 | $3032 \pm 467$ | $2954 \pm 455$ | $1869 \pm 377$ | $1758 \pm 395$ | $\mathbf{1502 \pm 578}$ |
| 4 | $3036 \pm 479$ | $2901 \pm 458$ | $1581 \pm 340$ | $1604 \pm 349$ | $\mathbf{1404 \pm 571}$ |
| 5 | $10424 \pm 885$ | $10358 \pm 880$ | $\mathbf{8861 \pm 853}$ | $9664 \pm 850$ | $10613 \pm 918$ |
| 6 | $10621 \pm 889$ | $10460 \pm 875$ | $\mathbf{7673 \pm 785}$ | $9068 \pm 806$ | $10503 \pm 905$ |

Three algorithms perform uniformly better than the benchmark algorithm Direct, namely: WLS (with reductions in total lengths in the 15% – 65% range) and to a smaller extent, Combi and OLS. Algorithm MinT has a dual behavior and is somewhat unreliable: it is the most efficient one for the smallest hierarchies but performs worse than the benchmark Direct for the largest hierarchies.

**Additional comments.** MinT is unreliable on the largest hierarchies, namely, for Configurations 5 and 6. This limitation appears when numerous components are to be predicted: $m = 1{,}756$ and $m = 1{,}801$, which may suggest either a poor estimation of the covariance matrix or non-invertibility issues. Our understanding of the phenomenon is that if the base forecasts are good, then they should be almost coherent. Consequently,

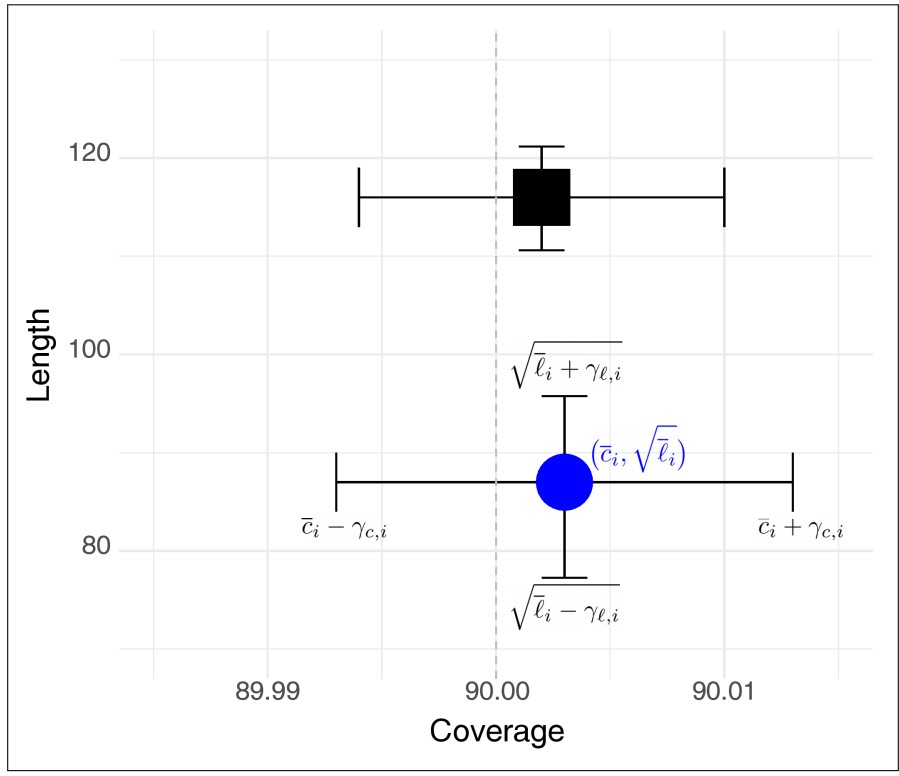

Figure 2: How to report concisely the indicators defined in (19) for a given element $i$ in the hierarchy (and two methods).

some forecasts are (almost) linear combinations of the others and the covariance matrix of the forecast errors can be (near) singular, especially for a large number of nodes. Our intuition is that this very phenomenon is the origin of the lack of robustness encountered during the experiment (indeed, we have $T = 10^6$ observations, which leaves a descent amount of $2 \cdot 10^5$ observations to estimate the covariance matrix). WLS does not encounter this issue because the diagonal matrix $\mathrm{Diag}(\widehat{\Sigma})$ used in the reconciliation step remains invertible. The takeaway message of this limitation is that, in practice, we advocate for the robust approach WLS instead of MinT, which attempts to mimic the theoretically optimal approach.

### F.3.4 Results: component-wise coverages and lengths

Section 5 (and the section above) only reported global results of efficiency. We now move to a component-wise study, and want to determine whether the conclusions made at a global level—in particular, that WLS is a robust improvement to the benchmark—hold also at individual levels.

To do so, we must be able to report concisely the indicators defined in (19), for all $i \in [m]$.

**Report for a given** $i$**.** We first explain how we report these indicators for a given $i$. We do so via a graph whose $x$–axis is dedicated to coverage levels (in %) and whose $y$–axis indicates lengths.

The center of the cross is formed by $\left(\bar{c}_i, \sqrt{\bar{\ell}_i}\right)$ and the horizontal and vertical whiskers report, respectively,

$$\left[\bar{c}_i \pm \gamma_{c,i}\right] \quad \text{and} \quad \left[\sqrt{\bar{\ell}_i - \gamma_{\ell,i}}, \sqrt{\bar{\ell}_i + \gamma_{\ell,i}}\right].$$

We summarize these elements in Figure 2. We evaluate performance as follows: the closer the cross to the lower center of the plot, the better the performance.

**Report for all** $i \in [m]$**.** Figure 2 considered one given $i$ and we should produce such pictures for all nodes $i \in [m]$ of a given hierarchy. We do so in the figures of the next two pages, where we organized the subgraphs by hierarchy levels (with horizontal separations between levels); these figures correspond, respectively, to Configurations 1 and 2 described in Appendix F.1.2.

We now comment their outcomes.

**Component-wise coverages.** The nominal coverage levels of $1 - \alpha = 90\%$ are achieved, at each node and irrespective of the method considered, as stated Theorem 2.

**Component-wise efficiency.** The graphs reported on the next two pages depict a noteworthy dual behavior.

At the most disaggregated level (at the leaves), all algorithms exhibit a performance superior to the one for the Direct approach. This improvement is only mild for OLS, but is substantial for the other three algorithms: MinT, WLS, and Combi. In particular, MinT provides shorter prediction sets at each node of the disaggregated level (most often in a statistically significant way).

At aggregated levels, MinT and WLS have a performance comparable to the one of the Direct approach, which seems marginally superior (but not in statistically significant way). This is not the case for Combi and OLS, which perform poorly at these aggregated levels. This phenomenon might be linked to the superior performance of the base forecasts at these aggregated levels: as a result, it becomes more challenging to leverage the information provided by the base forecasts at the most disaggregated level.

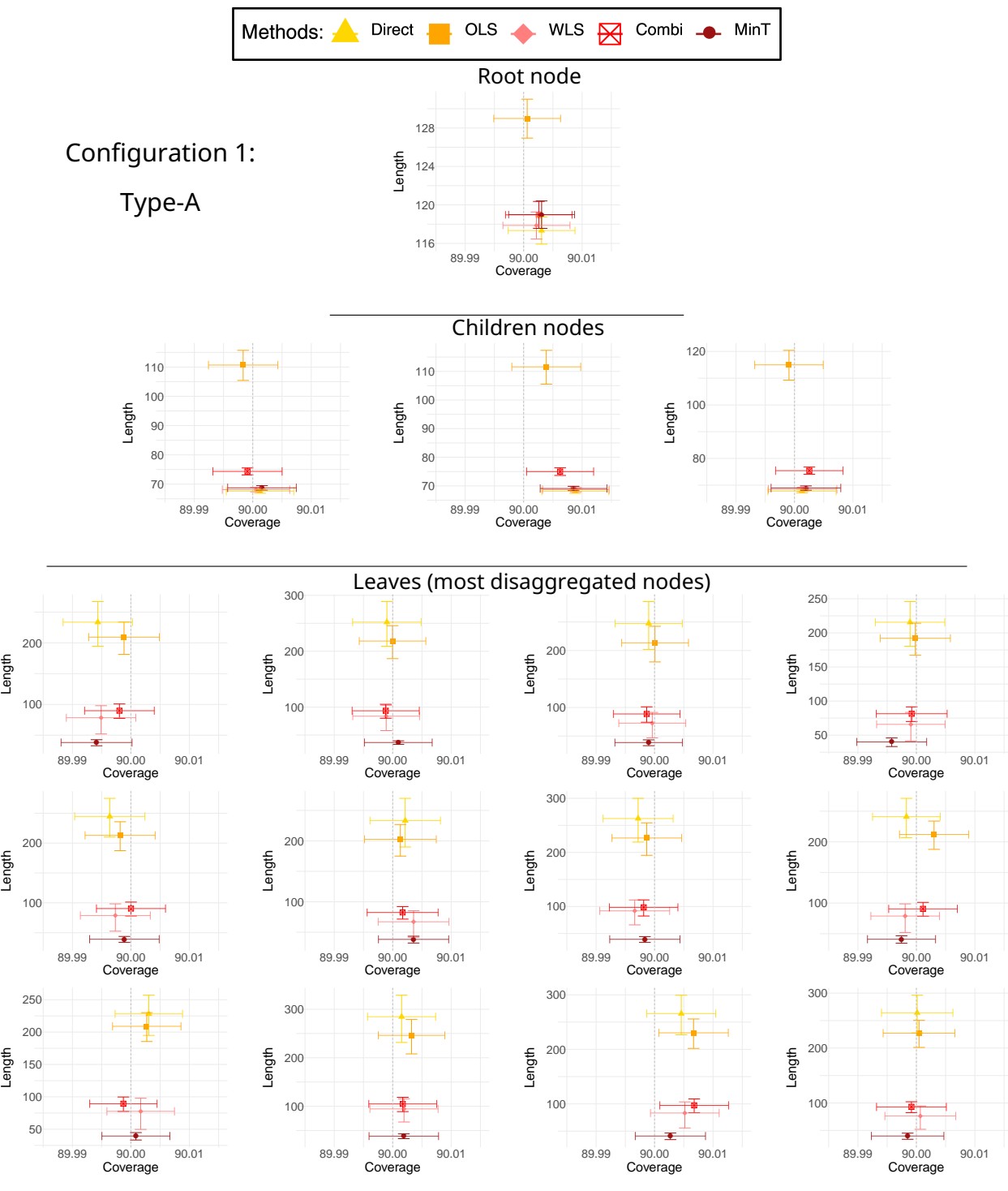

Figure 3: Component-wise coverages and efficiencies achieved for the methods considered on Configuration 1 of Appendix F.1.2. This figure should be read as indicated in Figure 2.

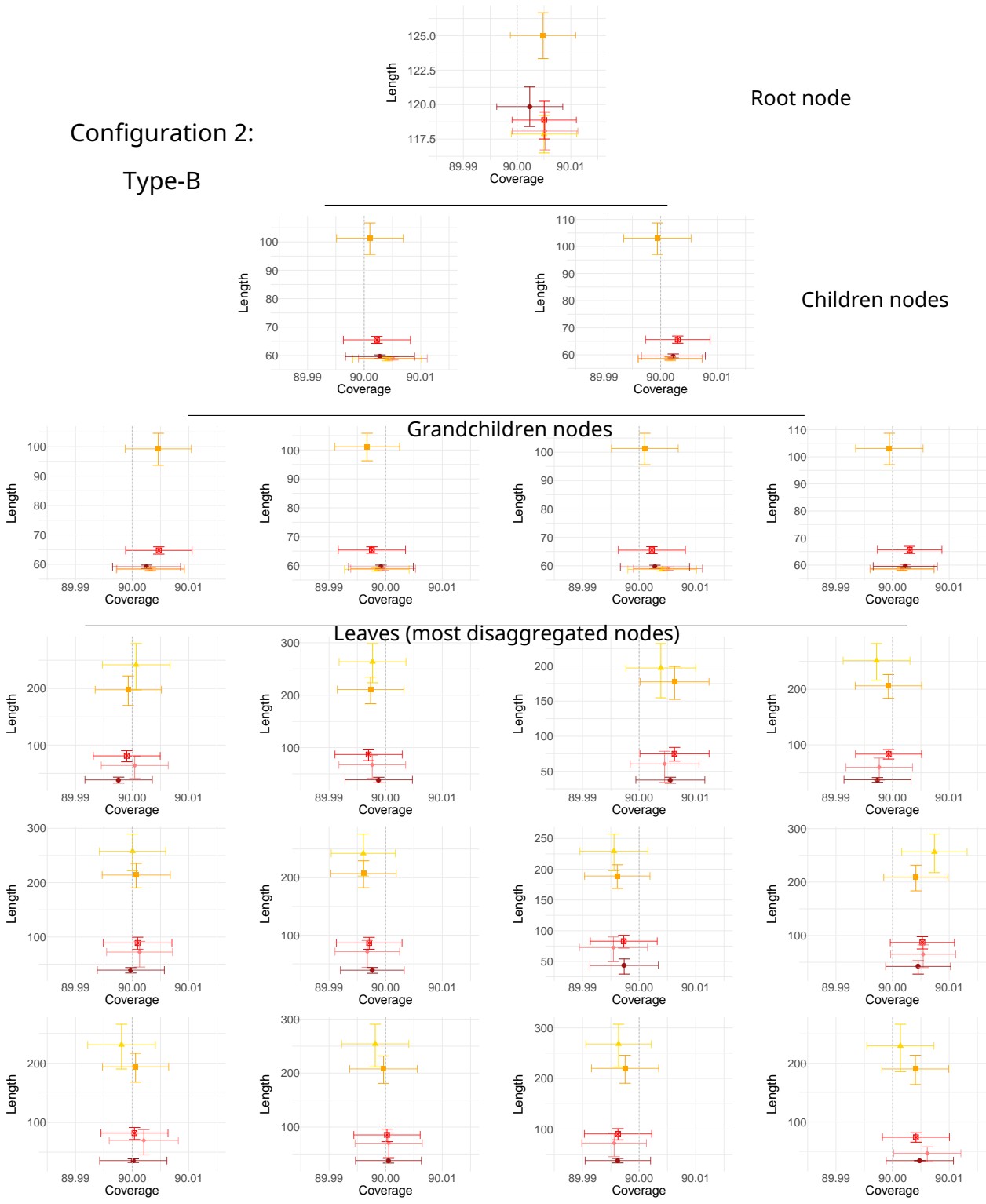

Figure 4: Component-wise coverages and efficiencies achieved for the methods considered on Configuration 2 of Appendix F.1.2. This figure should be read as indicated in Figure 2.

