# OpenReview forum: "Conformal Prediction for Hierarchical Data"
_TMLR — Accepted by TMLR_

### Review · Reviewer_Hu5R · 2025-11-11

**Summary Of Contributions:**

This paper is about doing conformal prediction for regression tasks with linear hierarchical structure in the response variable. I.e., we observe covariates $x$ and outputs $y \in R^m$ such that there are known linear constraints relating some entries of $y$ to other entries of $y$; an example comes from the electricity usage each household in a town adding up to the total electricity usage of the town. A matrix $H$ is assumed to encode this hierarchical structure such that $Hy_{1:n} = y$ for some $n \leq m$. Such $y$'s meeting the constraints of $H$ are called "coherent"

The paper proposes to modify existing conformal prediction methods to account for this hierarchical structure. The proposed adaptations boil down to projecting the conformal residuals $y_t - \hat y_t$ (where $\hat y_t$ is the predicted value of $y_t$) onto the range of $H$. The resulting conformal residuals are thus themselves coherent. The paper proves conditions under which such a projection results in savings in efficiency over non-hierarchical-aware conformal methods (i.e., size of the resulting conformal uncertainty set is smaller) while still maintaining correct coverage. Notably, these algorithms and theoretical results are extended to settings in which component-wise coverage is desired (i.e., we want the conformal uncertainty set to simultaneously cover each component $y_1, \dots, y_m$).

In experiments with synthetic data, the paper demonstrates the efficiency gains of the proposed method.

**Additional Comments:**

Thanks for submitting the paper!

**Audience:**

Yes

**Audience Explanation:**

I think conformal prediction is of broad interest to anyone interested in regression within the machine learning community. And this paper extends conformal methods to an interesting special case, that of hierarchical regression. The motivating examples of household expenditure or energy consumption make a convincing case that such hierarchical methods are needed in a breadth of applications of machine learning.

**Broader Impact Concerns:**

No concerns here.

**Claims And Evidence:**

Yes

**Claims Explanation:**

The paper provides clearly stated algorithms extending conformal prediction methods to the hierarchical case and then rigorous proofs of their claimed results about these algorithms. This is the main claim of the paper.

**Requested Changes:**

# Changes for acceptance

Overall, I feel this paper essentially already meets TMLR's acceptance criteria in that it's well written, technically correct, and provides an at least modest contribution to the literature. One point that I think is missing from its discussion of related work is the issue of the fitted regression function $\hat \mu$ being coherent or not (i.e, whether or not it meets the hierarchical constraints).

A decent amount of the discussion (like the point of defining $\mathring{\mu} := P_A \hat\mu$) and some of the gains in efficiency (like those in Theorem 1) seem to only hold if $\hat\mu$ is not already coherent. When would this be the case? It seems to me that if one understands the hierarchical structure and is going to use it for the purposes of conformal prediction, one would also just use the hierarchical structure for the purposes of regression. Or do practitioners sometimes make a bias variance tradeoff by allowing $\hat\mu$ to be non-coherent?

This issue is the only point on which I thought the paper was somewhat unclear. Adding discussion of whether we should always expect $\hat\mu$ to be coherent and how that impacts the results / contributions of the paper will significantly strengthen it.

# Changes to strengthen the work
I have a few scattered comments. These are listed chronologically in the paper, and I've marked more significant ones with two asterisks.

- Example 1: it would be good to motivate covariates here before they're introduced mathematically in Section 2.1
- "We assume that i.i.d. hierarchical data" -- it's not defined immediately which parts are hierarchical (covariates or responses)
- "Theorem 5 actually provides a stronger result of uniform domination..." -- this sentence reads more like a definition of uniform domination than the summary of a theorem.
- ** The motivation for component-wise coverage was never really discussed; why is it worth doing all of the extra technical work to get component-wise guarantees? Some extra references / discussion of existing work would help here. In particular, how would the authors answer a practitioner who says "the component-wise stuff seems complicated, so I'll just always ignore it and use Algorithm 2 and call it a day"?
- "We however provide two separate statements to clarify the notation..." -- it would be good to state this right after this notation is used for the first time. Also, this seems inconsistent with Section 2.3.1, which uses the notation $\mathring{\mu}$ to denote the reconciled version. Also, the use of the word "reconciled" here seems like a stand-in for "coherent"; I think it's worth picking one of these terms and sticking to it throughout the paper (it could be noted when defining coherence that the same concept is sometimes called "reconciled").
- ** Theorem statements in the main text should be linked to their formal statements and proofs in the appendix and vice versa; this makes it significantly easier to read the paper's theoretical results.
- ** Two comments on Theorem 1: (1) Some discussion of when strict inequality / equality will hold would be nice. Based on reading the proof of Appendix D, we should expect equality if  the predicted $\hat\mu(x)$ is coherent. This again goes back to needing a discussion of when we should / shouldn't expect this coherence to hold. (2) I think there is sort of an assumption on the data, which is that it is hierarchical with structure $H$. True, $H$ could just be the identity matrix, but I think Theorem 1 should still state this as a needed constraint/input.
- Algorithm 3 is pretty redundant with Algorithm 4. If the paper needs to save some space, I think Algorithm 3 could be cut.
- "We justify in detail why this additional assumption may not be considered unnatural nor too restrictive." -- I don't agree with the non-restrictiveness of this assumption. I think the following paragraph does a good job of justifying that such assumptions are extremely common in the literature, and thus there is reason to believe that a similar assumption is *needed* to get any kind of theoretical efficiency result. So I agree that this type of assumption is very reasonable from a theoretical perspective. But, this is very different from saying that the elliptical assumption is non-restrictive. My a-priori assumption is that there are exactly zero applications in which the residuals are literally elliptical, and from that perspective, this is an extremely restrictive assumption. Maybe there are lots of empirical studies out there suggesting that, especially in hierarchical regression, the residuals are elliptical. But I think if a statement on the restrictiveness is going to be made here, such studies need to be cited.
- Theorem 3: Again, discussion of when strict inequality / equality should hold would be good here. Reading the proof sketch, it's not that intuitive when the key inequality will be tight or have slack in it.
- "it performs and ordinary" -- typo
- ** I don't think Section 5 (experiments) stated what the regression algorithm was. In particular, is it coherent or not?
- "It turns out that all algorithms do achieve the required coverage level" -- Would it be accurate to say "As expected, all algorithms do achieve..." just to make it clear that this is an expected result?
- "SCP for joint coverage based on ellipsoidal sets." -- The rows are differentiated here by being called Config "1" and "2" -- I wasn't clear what these two configurations are
- "Component-wise SCP." -- these standard errors are huge! I think more samples are needed to reduce the standard errors here. In particular, this table doesn't really show which method is best, as many of the confidence intervals overlap. Alternatively, another way to show which method is the best would be to report win percentages (i.e., what fraction of the time each algorithm has the smallest interval); I suggest this because I'm guessing that the times that WLS is large in configuration 5 might also be the same times that Combi is large, and so despite the fact that the WLS and Combi confidence intervals significantly overlap, it might be that Combi is never better than WLS. Another thing that I think should be discussed / explained: why are there huge differences in performance across the different configurations? The interval size is varying by two orders of magnitude! And some methods seem to be way better in some configurations than others. Why is this? Is it more about the depth $k$, the number of nodes $m$ or the number of leaves $n$?

---

> ### Author Response · Authors · 2026-02-12
> **Answers to the main comments**
>
> We thank the reviewer for the careful reading, constructive feedback, helpful suggestions (and overall positive appreciation)!
>
> Our answers come in late, as we waited for all 3 reviews to be available to provide an updated PDF version of the submission (with changes marked in purple).
>
> **Main change requested (are base forecasts coherent?)**
>
> The clarifications asked are particularly interesting (and we should have thought of including them straight away). In a nutshell, enforcing coherence during training typically requires joint learning across all nodes. This can be achieved, for instance, by adding constraints or penalties to a multivariate objective, or by inserting reconciliation steps within each iteration of an optimization procedure. Such approaches substantially increase computational cost, can introduce a bias, and restrict the choice of models. For this reason, practitioners often fit models separately between components and enforce coherence through forecast reconciliation. The latter is a post-hoc approach that is particularly compatible with conformal prediction, due to its model-agnostic nature.
>
> We included a synthetic summary of these considerations:
> - in the first two paragraphs of Section 1.1 (on page 1),
> - below Figure 1 (on page 4),
> - in new Remarks 2 and 3 (on pages 8 and 10).
>
> **Other most important comments (the ones with two asterisks)**
>
> *Component-wise guarantees*
>
> We agree that providing / explaining the proper motivation for this objective is vital. To us, these guarantees are actually more natural and more intuitive for practitioners, as each component is typically also considered individually. The most convincing examples we have (e.g., sales, by store or at a global level) are formed by time series, though, but they still explain why individual components are studied individually. We added some text at the end of page 2 and top of page 3 to reflect this, but are open to developing further the discussion if needed.
>
> *Linking theorem statements in the main text to their formal statements and proofs in the appendix*
>
> We agree on this and provided extra pointers and explanations around Theorem 1 (see end of page 6 and the theorem statement itself), which indeed, was problematic to relate to the statements in appendix. We checked the text around Theorems 2-3-4, the top of page 16 listing the content of the appendices, and the (sub)section titles in the appendices, and we believe that they should be sufficient to navigate easily the article. We of course would happily make any additional adjustment pointed out by the reviewer.
>
> *On the two comments on Theorem 1*
>
> As indicated above, we added a remark stating that equality holds when forecasts are already coherent. As for the assumption on data, yes, implicitly, here in Theorem 1 but actually in all theorems throughout the article, we assume that the data is $H$--linear, with $H$ being known. If this assumption is to be reminded in theorem statements, it should be in all of them, not just in Theorem 1. We would rather be inclined to think that the setting description makes it now clear enough that the hierarchical assumption is global to the article: see the new Assumption 1; but we welcome any suggestion to make this even clearer.
>
> Also, the case of $H$ being the identity, i.e., $n=m$ ('plain multivariate observations', as defined right before Section 2.1) is interesting to mention: in that case, data is actually not hierarchical, Algorithms 1 and 2 coincide, so do Algorithms 3-4-5, and we offer no improvement. Should we mention this somewhere, and if so, where? Again, we are ready to listen and follow the reviewer's suggestions on this.
>
> That all being said, we corrected something linked to the statement of Theorem 1: the piece of notation $\mathcal{E}$ for the procedure picking the matrix to compute norms was not introduced in the main body, only in Appendix D, which made the statement of Theorem 1 unreadable; we corrected this.
>
> *On the regression algorithm for experiments (is it coherent or not?)*
>
> We agree that it is critical to state in the main body how base forecasts were achieved; we now do so and added that 'we consider base forecasts given by generalized additive models learned independently on each component (details provided in Appendix F.1.4)'; in particular, base forecasts are built component by component, are therefore not coherent, and must be reconciled.

---

> > ### Author Response · Authors · 2026-02-12
> > **Other comments**
> >
> > - We added covariates to Example 1
> > - Beginning of Section 2.1 ('i.i.d. hierarchical data'): we agree that this formulation was ambiguous and now clearly state that observations are hierarchical (not features)
> > - 'Theorem 5 actually provides a stronger result of uniform domination...': we corrected the formulation (see beginning of Section 2.2)
> > - 'We however provide two separate statements to clarify the notation': as suggested by another reviewer, we actually put a reminder of these notation at the first place where they are critically used, i.e., right before Theorem 3; would this also address your comment? It is true that the $\mathring{\mu}$ of Section 2.3.1 and the $\tilde{\mu}$ of Section 2.3.2 coincide and are both given by projected forecasts; however, the notation have a wider use than just for the regression functions $\mu$ and we use them mainly to distinguish the prediction sets (the half circle, the ring, the hat, the tilde) depending on whether they are good for joint or component-wise prediction, and whether they are based on original or reconciled (projected) forecasts. Let us know whether we should provide a 2x2 table summarizing this, perhaps this would be useful? Finally, to us, reconciliation means adjusting forecasts by projections to make them coherent; we used reconciled forecasts in the sense of modified forecasts that became coherent. So, all reconciled forecasts are coherent but not all coherent forecasts are obtained by reconciliation (see the first two paragraphs of Section 1.1).
> > - We actually were happy to have an admittedly redundant statement of Algorithm 3 distinct from Algorithm 4 mostly to illustrate our notation 'hat versus tilde' (see the previous answer)
> > - We fully agree that the additional assumption is restrictive per se, but we actually meant that it not too restrictive compared to existing assumptions for efficiency results: we exactly added these critical words ('compared to existing assumptions for efficiency results') when starting the discussion of Assumption 3 and believe that this is enough to fully align our arguments with the similar arguments and grounds raised by the reviewer
> > - We added a similar comment after the proof sketch (Remark 3 top of page 10); we have little to no intuition regarding the tightness of the key inequality, which arises from a Pythagorean theorem applied to a matrix $M$ depending on the underlying elliptical distribution...
> > - Typo 'and' / 'an' corrected
> > - We implemented the 'As expected, ...' suggestion (see page 11)
> > - A forward pointer to Appendix F.1.2 was critically missing for the reader to understand what Configurations 1 and 2 were: we added it (see page 11)
> > - Reviewer uMXX is also concerned with identifying in which configurations the achievements are, or are not, statistically significant; we would be ready to add a column indicating when the best algorithm is significantly superior to the second best. However, we prefer to keep as single criterion the expected lengths, as this is exactly the criterion for the theoretical results (we do not provide theory for the win percentages). Yes, standard errors are rather large in the small configurations, and yes, the number $m$ of components is the main driver for different orders of magnitude (essentially, the root-empirical averages reported increase as $\sqrt{m}$). Should we incorporate some of these considerations (please help us finding which would be useful)? Note that the robustness of some methods to large configurations is already discussed in Appendix F.3.3.

---

### Review · Reviewer_xv8L · 2025-11-25

**Summary Of Contributions:**

The authors propose a method for improving the efficiency of multivariate conformal prediction given known linear constraints. Linear constraints can be written as the image of the data under a known linear map. This linear map H is then propagated through the mathematical machinery to produce algorithms and bounds.

**Audience:**

Yes

**Audience Explanation:**

I think that the result is probably straightforward, but could be a super useful one and could be valuable to the community at large.

**Claims And Evidence:**

No

**Claims Explanation:**

The contribution made by this paper is clear but I found the paper itself to obscure it unnecessarily. There isn't a single real-world motivating example. There is no high-level intuition for the logic and mathematical rigor which is introduced mercilessly.

I would suggest:

1. Introducing a real-world example. I think that downsampling / upsampling could be an excellent one. It is often (bi-)linear and provides an easy to understand example where the assumptions of the paper apply. Figure 1 provides a contrived example which is not particularly believable.

2. Improve notation. Putting the order statistic in the subscript is too verbose s_{\floor{T_{calib}+1,...etc}}. I would suggest introducing new notation here, e.g. "upper" and "lower" to help make the equations more readable.

**Requested Changes:**

Below are a mix of nitpicks and larger changes which would improve the flow of the paper.

1. When you write "i.i.d." put a slash after to escape the following space, e.g. do "i.i.d.\". This will just add a bit more professionalism to the paper.

2. The introduction does not need to be overly hierarchical. Introduce and motivate the topic, at the end of the introduction provide the contributions. Finally, create a separate top-level related-work section.

3. Think carefully about the notation that you define and do not define. For example, it's inconsistent to define `Tr` but not define `Im`.

4. Remark 1 is simply a prior. Calling it such might help ML-oriented readers understand the idea with fewer words necessary on your part.

5. $C: x\in\mathbb R^d \to C(x)$ is overly verbose notation and does not clarify the main question which is, what is the target space?

6. In the equation above 2.2, why is the probability $\approx 1-\alpha$ and not $= 1 - \alpha$? What does $\approx$ actually mean here?

7. Please give numbers to all equations.

8. Don't start mathematical notation on a newline, use a tilde (\~). E.g. regressor function\~ $\hat\mathbb\mu: \mathbb x\in \mathbb R^d...$

9. Use simpler language. Instead of "data-based definite positive matrix design to capture multivariate dependencies," why can't you just say covariance matrix? Is there any exception to the rule that this will be a simple covariance matrix?

9.1 It should be "positive definite data-based matrix"

10. "Synthetical data" should be "synthetic data."

11. Consider including plain English explanations of the theorems, as well as a story connecting them. This will make the paper accessible to a larger technical audience.

---

> ### Author Response · Authors · 2026-02-12
> **Answers to the main comments**
>
> We thank the reviewer for the detailed comments and feedback.
>
> Indeed, the submission is primarily theory-oriented, focusing on simple methodological developments supported by highly theoretical results. Some of the reviewer's suggestions appear to reflect expectations for a more applied submission. While we have clarified motivation and added references and illustrative examples where appropriate, the primary scope of the paper remains theoretical, as intended.
>
> Theoretical efficiency guarantees in conformal prediction are scarce. We believe that the set of concepts, formal results, proofs, and illustrative synthetic experiments constitutes a substantial and self-contained contribution. We provide an updated version of the submission with changes highlighted in purple.
>
> **About the lack of real-world motivating examples**
>
> We now provide motivating examples and references in Section 1.1, in particular, a link to a survey article. The situation of Figure 1 arises in setting where some sales or total consumption (e.g., electricity consumption) is the sum of sales or consumptions at more local level; e.g., to match Figure 1, three stores in one city and two stores in another city. Figure 1 is indeed stylized as typically there would be more stores and more cities but it illustrates the concept of the hierarchical nature of the observations. The vast literature of hierarchical forecasting, to which we point, provides lots of examples and real-world applications.
>
> As detailed in the answer to Reviewer uMMX, we could get access to hierarchical time series data but not to hierarchical i.i.d. data, which exists like household spending surveys, but cannot be shared due to confidentiality.
>
> Now, the application to image treatment (through downsampling / upsampling) looks interesting and could be an excellent case as well but we are not specialists in image treatment (and are even ignorant about the basics of the domain): may we leave it to motivated TMLR readers to apply the theory we developed on such a use case?
>
> **There is no high-level intuition for the logic and mathematical rigor which is introduced mercilessly. Improve notation. [...]**
>
> The paper combines concepts from hierarchical forecasting and conformal prediction, and therefore, some (much?) technical content is necessary; nevertheless, we aimed to make the exposition as clear and self-contained as possible. We acknowledge that despite our best efforts, reading this submission will require some preliminary familiarity to conformal prediction and multivariate statistics, though. (Note that the piece of notation for order statistics is fully standard.)

---

> > ### Author Response · Authors · 2026-02-12
> > **Other comments**
> >
> > 1. It turns out that we were already writing 'i.i.d.\' in the LaTeX code!
> > 2. Our introduction addresses the three sets of elements mentioned (motivation, which we extended; related work; contributions and outline) and we would rather discuss related work and literature before stating our contributions
> > 3. We agree that we should have defined the notation Im and now do so; we also added here the notation for the Lebesgue measure and for the $A$--norms
> > 4. We agree that the clear message of Remark 1 should be that $H$ is known, which is why we rewrote the remark; however, we would keep the name of 'prior' to prior distributions in Bayesian analysis, which is why we did not use this word here
> > 5. We had written (and still write): '$C : x \in \mathbb{R}^d \to C(x)$ is an application taking subsets of $\mathbb{R}^m$ as values, i.e., $C(x)$ is a subset of $\mathbb{R}^m$; we want to avoid introducing a notation like $\mathcal{P}(\mathbb{R}^m)$ for the set of subsets of $\mathbb{R}^m$ and say that $C$ takes values in $\mathcal{P}(\mathbb{R}^m)$ but would happily do so if the reviewer feels that it would help clarifying the introduction of $C$
> > 6. The $\approx$ present in Section 2.1 and in Theorem 1 refer to the upper and lower bounds stated in Theorem 5 (see page 27); these bounds are super classic in conformal prediction, which is why, for the sake of brevity, we rather wrote $\approx$ instead of devoting space to formally stating the upper and lower bounds
> > 7. In our subcommunity of machine learning, articles typically only number equations that are referred to at some other place in the article (simply because there are so many) and we would rather stick to this convention
> > 8. We deliberately chose not to systematically start inline formulas with a $\sim$, as the current formatting improves readability and maintains a clear flow of mathematical expressions.
> > 9. Yes, while this is a specific and natural choice, other ones are considered (namely, by the WLS and Combi methods, see page 11) // 9.1. We corrected the word order into 'positive definite data-based matrix' (page 5)
> > 10. We corrected the single occurrence of 'synthetical' into 'synthetic'
> > 11. Our intentions, instead of commenting theorems right before or right after their statements, which would introduce repetitions was to provide such general comments in Sections 2.1 and 2.2 (and all theorems should be read with these sections in mind) and to guide the reader into the logic of the article at the beginning of each subsection stating a theorem (e.g., beginnings of Sections 4, 4.2, and 4.3)

---

### Review · Reviewer_uMXX · 2026-02-04

**Summary Of Contributions:**

This paper proposes algorithms to construct prediction regions when fitting models to hierarchical data. The keystone of the proposed algorithms is to use a projection step, and the paper explore several choices of projection matrices, some of which can be estimated from the data. The paper makes the following assumptions: the non-conformity scores are i.i.d, have an elliptical distribution and have a finite second moment. Under these conditions, the paper derives several guarantees for joint coverage and component-wise coverage, establishing that using a projection reduces the volume of the prediction region. Finally, the paper provides numerical experiments on simulated data: the experimental results illustrate the theory and provide insight into which projection matrix to use.

**Additional Comments:**

The paper is clearly written.

**Audience:**

Yes

**Audience Explanation:**

The TMLR community is interested in conformal prediction and more generally uncertainty quantification. On the other hand, I'm not familiar with hierarchical data and I don't know how often they arise.

**Broader Impact Concerns:**

None.

**Claims And Evidence:**

Yes

**Claims Explanation:**

The theoretical statements look sound to me, although I have only schemed the appendix. The setting for the numerical experiments is clearly explained. As far as I can tell, the paper does not make any speculative statements and the discussion highlights limitations.

**Requested Changes:**

I have a few questions and comments.

**Hierarchical data.** The paper should provide more context on hierarchical data and discuss applications, for example adding references at the end of the first paragraph. This would help motivate the paper. I also recommend extending the numerical experiments to include applications to real data. These additional experiments would provide illustrations of hierarchical data and strengthen the experiment.

I also encourage the authors to clearly indicate that hierarchical data do not imply a hierarchical model---I spent a good portion of my first read expecting the paper to be about hierarchical models. (But to be fair, that's one me!)

**Add real data to experiments.** As stated above, I believe the paper would be strengthened if the experiments were extended to real data. The authors allude to an example at the top of Section 5 and then explain they could not access a sufficient amount of data. In what sense is the data "insufficient"? Also, surely there must be other data sets that the author can use for their experiments.

**Clarification on distribution-free assumption." In line with the conformal prediction literature, the authors emphasize the distribution-free nature of their results. On the other hand, they make distribution assumptions about the nonconformity scores. Does this not imply assumptions about the distribution of the data in relationship to the regression algorithm being used?

**Minor comments.**
- Figure 1. In the tree, it could be worth stating explicitly that A = AA + AB + AC, etc. It took me sometime to parse the tree. The authors might also consider a concrete example to help clarify the concept of hierarchical data.
- First paragraph of Section 4, "PH = P" --> "PH = H".
- After definition 3: what is a "stability property"?
- In theorem 3, consider reminding the reader that $\tilde C_i$ is the prediction set for Algorithm 4 and $\widehat C_i$ for Algorithm 3.
- "The inequality above is a result of our own, though inspired by the literature of forecast reconciliation." Can the authors provide a reference or two (an "e.g.,")?
- Table under **practical implementation** $P_{WLS}'$ --> $P_{OLS}'$.
- In the equation defining $\bar L$, replace $10^3$ with $N$ to avoid magic numbers.
- I recommend replacing/complementing the tables in Section 5 with bar plots --- I believe this would make the outcomes of each method easier to compare and also it would be easier to see which performances are within each other's standard deviation.

---

> ### Author Response · Authors · 2026-02-12
> **Answers to the main comments**
>
> We thank the reviewer for the careful reading, constructive feedback, helpful suggestions (and overall positive appreciation)!
>
> We provide answers below but also submitted an updated PDF file (with changes marked in purple).
>
> **Clarification on distribution-free assumption**
>
> The coverage guarantees are fully distribution-free, and only the efficiency results rely on the additional assumption mentioned by the reviewer (as are all existing efficiency results in conformal prediction). We were already explaining and detailing both of this at the end of Section 4.1 but we agree that extra care and extra comments were needed, which we now provide also at the beginning of Section 4.1.
>
> **Hierarchical data**
>
> We added two references at the end of the first paragraph to better contextualize hierarchical data and their practical relevance. In particular, the recent review by Athanasopoulos et al. (2024) provides more context on hierarchical data and provides pointers to applications performed in the literature. Because of the existence of this recent review, we did not add too much text here. We also included a short comment on hierarchical modeling being a different concept at the end of the 'Terminology clarification' paragraph.
>
> **Application to real data**
>
> We wanted to perform applications to real data on top of synthetic data. We had access to (lots of) hierarchical data in the form of time series, but not to i.i.d. hierarchical data such as household expenditure surveys (total spendings + spendings per category)---this kind of data being most of the time private or confidential.
> - For example, we were able to access an i.i.d. dataset of consumer behavior in supermarkets, but it contained only a few hundred individuals, which is insufficient to meaningfully illustrate the efficiency results. (We by the way corrected the wording in page 11 from 'we could not get access to a sufficient amount of such data' into 'we could not get access to such data'.)
> - However, Reviewer xv8L seems to have an excellent example of i.i.d. hierarchical data, for which we are not competent. The hope is that some TMLR readers read this review thread and perform the said i.i.d. application.
> - As for hierarchical time series (for which we do have data available), we note in Section 6 that extending our framework and results to hierarchical time series requires additional methodological developments and therefore falls outside the scope of the present paper. We are currently pursuing this direction in a separate work, building on the results and achievements of this submission.

---

> > ### Author Response · Authors · 2026-02-12
> > **Answers to minor comments**
> >
> > - We added the indicated summation constraints under the suggested form
> > - We corrected the typo into $PH = H$
> > - We agree that our writing was cryptic: we meant stability under affine transformations as following from Lemma 3 in Appendix A.1, which the reader must read to understand what we mean; we therefore added an explicit pointer and now write: 'enjoying a stability property through affine transformations as expressed in Lemma 3 of Appendix 4.1'
> > - We implemented the great suggestion of reminding the reader of the notation right before stating Theorem 3
> > - We indeed added an example of source of inspiration (namely, Panagiotelis et al., 2021, Theorem 3.2)
> > - We clarified the notation and the fact that $\mathcal{P}'$-OLS is a constant function (which is why we avoid using this piece of notation in the table and rather report its constant value $P_{\boldsymbol{1}}$)
> > - We agree and replaced $10^3$ by N
> > - On adding barplots --- This is the only comment we did not implement (yet): we guess that the suggestion is about the two efficiency tables of pages 34 and 36; at any rate, we would consider only completing the tables, as the latter form to us the most effective format to report numbers that are sometimes of very different orders of magnitude (see, e.g., the two series of lengths for Configurations 1 and 2 in the table of page 36). However, even these bar plots complements are likely to be difficult to read. Yet, we agree that it is important to flag when superiority is achieved beyond standard errors. Would it be OK if we indicate, in one way or the other (e.g., through an additional column) whether this is the case? So, our suggestion would be to fully stick to tables but to complement them through an additional column addressing the important concern of significant difference achieved or not.

---

> > > ### Comment · Reviewer_uMXX · 2026-02-17
> > >
> > > Thank you for your detailed response. I think the clarifications you propose will improve the paper.
> > >
> > > > However, Reviewer xv8L seems to have an excellent example of i.i.d. hierarchical data, for which we are not competent. The hope is that some TMLR readers read this review thread and perform the said i.i.d. application.
> > >
> > > I wouldn't count on readers reading the review, so my suggestion is to have some summary of our discussion in the paper itself. In particular, you can highlight that i.i.d hierarchical data is difficult to come by, mention that a reviewer provides an example you didn't explore, and that more applications arise in a time-series rather than i.i.d context---which is a direction you are currently pursuing. This also helps define the scope of the paper's contribution.
> > >
> > >
> > > > On adding barplots...
> > >
> > > Ultimately, it's your call. Reporting tables with certain cells in bold is standard practice, but in my experience bar plots are always more informative and easier to read.

---

> > > > ### Author Response · Authors · 2026-02-17
> > > > **Will add a discussion summary**
> > > >
> > > > Thanks for your comments and suggestions---and we commit to add a summary of the discussion on real data and applications that took place on this page (that indeed, readers may not necessarily read).

---

### Author Response · Authors · 2026-05-18
**Revision comments**

We thank the reviewers and the associate editor for handling our submission.

To produce the final version, based on the review thread, we performed mainly two series of modifications, both mostly located in Section 5:
- We provided a summary of the discussion on real data with Reviewer uMXX (see Remark 4 as well as the last paragraph of Section 6)
- We added two elements in the exposition of numerical results (the reason for large standard errors, and a discussion of statistical significance) based on the discussions with Reviewer Hu5R and Reviewer uMXX

We also updated the bibliography (7 arXiv items were published meanwhile; only 3 arXiv preprints remain unpublished to date).

---

### Decision · Action_Editor_BGzn · 2026-05-03

**Recommendation:** Accept as is

**Audience:**

Yes

**Audience Explanation:**

The paper presents a nice combination of conform prediction and forecast reconciliation. Thus the current work might be interesting to researchers or practitioners working in either field.

**Claims And Evidence:**

Yes

**Claims Explanation:**

This paper presents conformal prediction methods for hierarchical multivariate data, where some response components are known linear combinations of others. Its main contribution is to combine split conformal prediction with forecast reconciliation, where they first project forecasts onto the coherent hierarchical subspace, then use conformal calibration to obtain valid prediction regions. The paper proves that this reconciliation step preserves standard conformal coverage and can produce smaller prediction sets than unreconciled conformal methods. The efficiency claims are supported by theoretical results showing that reconciliation projects residuals onto the coherent subspace and thereby reduces conformal ellipsoid volumes or weighted squared interval lengths, together with synthetic experiments demonstrating smaller prediction regions while maintaining target coverage. Unfortunately, experiments on real-world data were not provided because they had difficulty in finding i.i.d. hierarchical data.